# Coverage Improvement and Fast Convergence of On-policy Preference Learning

**Juno Kim** [1]  **Jihun Yun** [2]  **Jason D. Lee** [1]  **Kwang-Sung Jun** [3]

## Abstract

On-policy preference learning algorithms for language model alignment such as online direct policy optimization (DPO) can significantly outperform their offline counterparts. We provide a theoretical explanation for this phenomenon by analyzing how the sampling policy's coverage evolves throughout on-policy training. We propose and rigorously justify the *coverage improvement principle*: with sufficient batch size, each update moves into a region around the target where coverage is uniformly better, making subsequent data increasingly informative and enabling rapid convergence. In the contextual bandit setting with Bradley-Terry preferences and linear softmax policy class, we show that on-policy DPO converges exponentially in the number of iterations for batch size exceeding a generalized coverage threshold. In contrast, any learner restricted to offline samples from the initial policy suffers a slower minimax rate, leading to a sharp separation in total sample complexity. We further propose a hybrid sampler based on a novel *preferential* G-optimal design, which removes dependence on coverage and guarantees convergence in just two rounds. Finally, we develop principled on-policy schemes for reward distillation which achieve faster noiseless rates. Experimentally, we confirm that on-policy DPO and our proposed reward distillation algorithms outperform their off-policy counterparts and enjoy stable gains across iterations.

## 1. Introduction

Alignment is a crucial step in the training pipeline of large language models (LLMs) to steer them towards human-centric goals or values (Bai et al., 2022; Ouyang et al., 2022). Two widely used paradigms for alignment are reinforcement learning from human feedback (RLHF) (Christiano et al., 2017; Ziegler et al., 2019; Stiennon et al., 2020; OpenAI, 2024) and direct policy optimization (DPO) (Rafailov et al., 2023). RLHF first trains a reward model from human preferences and then learns a reward-maximizing policy using RL algorithms. However, the policy gradient updates in RLHF induce large gradient noise which slows or leads to unstable training, and treatments such as clipping (Schulman et al., 2017) introduce biases and additional sensitive hyperparameters. DPO bypasses the need for reward modeling by directly optimizing the policy from pairwise preference data. Due to its computational efficiency, stability and elegance, DPO and its variants have gained widespread adoption and achieved tremendous success over diverse domains (Liu et al., 2025; Xiao et al., 2025). However, compared to RLHF where the learned reward model can be freely queried, a major drawback of vanilla DPO is that the preference dataset typically must be collected beforehand and fixed throughout training due to the high costs and latency of gathering human feedback. This makes the data purely offline and off-policy, which can induce distribution shift or reinforce undesirable behaviors present in the initial model or sampling policy (Tajwar et al., 2024).

To remedy this issue, various *on-policy* direct preference learning methods[1] have been proposed. This includes DPO with online AI feedback (Guo et al., 2024) and iterative DPO (Xu et al., 2024a), whose methods we directly analyze; and variants such as iLR-DPO (Liu et al., 2024), fast-slow chasing DPO (Qi et al., 2024), online VPO (Cen et al., 2025), online IPO (Calandriello et al., 2024), IRPO (Pang et al., 2024), SPO (Swamy et al., 2024) and DNO (Rosset et al., 2024). Similar to online RLHF (Bai et al., 2022; Touvron et al., 2023; Lee et al., 2024; Dong et al., 2024), given access to a preference annotator, these methods iteratively generate and label a new batch of responses from the updated policy to apply a DPO-style update.

On-policy preference learning algorithms have been empirically shown to significantly outperform their offline human-feedback counterparts in various settings (Guo et al., 2024; Xu et al., 2024a; Calandriello et al., 2024; Rosset et al., 2024;

---

[1]UC Berkeley [2]KRAFTON [3]POSTECH GSAI/CSE. Correspondence to: Juno Kim <junokim@berkeley.edu>.

*Proceedings of the 43rd International Conference on Machine Learning*, Seoul, South Korea. PMLR 306, 2026. Copyright 2026 by the author(s).

---

[1]We focus on on-policy algorithms which are batched-online, i.e., data is collected in batches after each model update, but refer to the general approach simply as on-policy preference learning.

Pang et al., 2024; Liu et al., 2024; Dong et al., 2024; Tajwar et al., 2024), thus offering a promising way to enhance LLM alignment in an efficient and scalable manner. At the same time, iteratively training on signals from the same AI annotator can lead to overfitting or degeneracy (Casper et al., 2023; Zhu et al., 2024; Pal et al., 2024). Hence it is important to pinpoint the gains of on-policy preference learning to inform the design of more effective online algorithms.

In the RL theory literature, this is typically studied through the lens of the base model's *coverage* (Kakade & Langford, 2002; Antos et al., 2006; Munos & Szepesvári, 2008; Chen & Jiang, 2019; Xie et al., 2022; Song et al., 2024), which measures how well the base policy covers the target and is closely tied to statistical complexity. Existing theoretical works typically focus on designing *online exploration* algorithms augmenting RLHF or DPO which improve or circumvent this dependency via RL-inspired optimism, pessimism, or active learning modifications (Xiong et al., 2024; Song et al., 2024; Shi et al., 2025; Xie et al., 2025; Foster et al., 2025; Cen et al., 2025). However, these approaches necessitate optimizing complex, often impractical objectives, and do not explain the significant gains that can already be observed by simply making the switch to on-policy.

**Our contributions.** Unlike previous studies which treat coverage as a fixed object, we analyze the *dynamics* of the sampling policy's coverage throughout training. We argue that even without exploration, on-policy preference learning benefits from the **coverage improvement principle**: with sufficient batch size, each policy update will lie in a region around the target which has uniformly better coverage, which in turn guarantees a smaller error for the next update.

For the theoretical setting, we adopt the contextual bandit formalism with linear softmax policies and model preferences with the Bradley-Terry assumption. We consider the simplest iterative on-policy version of DPO, where each new batch is sampled purely on-policy and used to update the current policy with standard DPO (Guo et al., 2024; Xiong et al., 2024). Our contributions are summarized as follows:

- In Section 3, we show that on-policy DPO with batch size exceeding a generalized coverage threshold $C_*^p(R)$ achieves linear convergence in the number of rounds, resulting in sample complexity $(\ln \frac{1}{\varepsilon}) C_*^p(R)$. In contrast, in Section 2.2 we prove a $\frac{1}{\varepsilon^2} C_*^p(R)$ minimax lower bound for any offline preference learner sampling only from the base policy, hence an exponential separation in overhead.

- In Section 4, we construct a hybrid DPO sampler based on a novel *preferential* G-optimal design, which can be computed and sampled from before training. This algorithm avoids the dependence on both coverage and feature covariance and converges in just *two rounds*, demonstrating the effectiveness of carefully designed offline data.

- In Section 5, we develop principled on-policy schemes for *reward distillation* algorithms (Gao et al., 2024; Mao et al., 2024; Yun et al., 2025; Fisch et al., 2025), which learn from reward differences rather than preferences. Based on an alternative deviation-based notion of coverage for general function classes, we prove that these methods enjoy faster noiseless rates compared to DPO by avoiding the stochasticity of preference labeling.

- In Appendix B, we provide experiments supporting our theory on practical alignment tasks. We demonstrate that on-policy DPO as well as our proposed on-policy distillation algorithms outperform their off-policy counterparts and achieve stable and monotonic performance gains.

All proofs are deferred to the appendix. A discussion of related work is provided in Appendix A.

## 2. Formalization of Alignment

### 2.1. Problem Setup

**Notation.** We adopt a contextual bandit formalism for language modeling (Rafailov et al., 2023; Xiong et al., 2024; Cen et al., 2025; Foster et al., 2025). Let $\mathcal{X}, \mathcal{A}$ be finite prompt and response spaces. Let $\mathcal{D}$ be a fixed probability distribution over $\mathcal{X}$ and $\Delta(\mathcal{A})$ denote the space of distributions over $\mathcal{A}$. For any divergence measure $D$, we adopt the shorthand $D(\pi\|\pi') = \mathbb{E}_{x\sim\mathcal{D}}[D(\pi(x)\|\pi'(x))]$. We model the *policy* induced by a generative model as a function $\pi : \mathcal{X} \to \Delta(\mathcal{A})$ from prompts to a distribution over responses; we will also consider *joint* policies $\pi \in \Delta(\mathcal{X} \times \mathcal{A})$. We are given a *base* or reference policy $\pi_0$, typically an LLM obtained by supervised fine-tuning (SFT) to be aligned. We denote by $B_p(\theta_0, r) = \{\theta \in \mathbb{R}^d : \|\theta - \theta_0\|_p \leq r\}$ the closed $d$-dimensional $L^p$-ball of radius $r$ centered at $\theta_0$. The sigmoid function is $\sigma(z) = 1/(1 + e^{-z})$.

**Learning from preferences.** We study the problem of alignment from pairwise preference data. As is standard, we model preferences using the Bradley-Terry assumption (Bradley & Terry, 1952; Christiano et al., 2017; Rafailov et al., 2023).

**Assumption 1** (Bradley-Terry model). *There exists a reward function $r^* : \mathcal{X} \times \mathcal{A} \to \mathbb{R}$ such that preferences between $a^1, a^2 \in \mathcal{A}$ are determined according to*

$$\mathbb{P}_*(a^1 \succ a^2 \mid x) = \sigma(r^*(x, a^1) - r^*(x, a^2)). \quad (1)$$

In the basic formulation of reinforcement learning from human feedback (RLHF) (Christiano et al., 2017; Ziegler et al., 2019; Bai et al., 2022; Ouyang et al., 2022), we learn the reward function $r^*$ from human-labeled preference data using the negative log-likelihood loss and optimize the KL-regularized reward: $J_\gamma(\pi) := \mathbb{E}_{x\sim\mathcal{D}, a\sim\pi(x)}[r^*(x, a)] -$

$\gamma \text{KL}(\pi \| \pi_0)$, where $\gamma > 0$ is a regularization parameter. We define its solution as the *target policy* $\pi^*$:

$$\pi^*(a \mid x) \propto \pi_0(a \mid x) \exp(\gamma^{-1} r^*(x, a)).$$

We also define the *induced binary preference distribution* of a policy $\pi$ relative to $\pi'$ as

$$\mathbb{P}_{\pi,\pi'}(a^1 \succ a^2 | x) := \sigma\left( \gamma \ln \frac{\pi(a^1|x)}{\pi(a^2|x)} - \gamma \ln \frac{\pi'(a^1|x)}{\pi'(a^2|x)} \right).$$

Since $\mathbb{P}_{\pi^*,\pi_0} = \mathbb{P}_*$ under Assumption 1, learning the target policy $\pi^*$ is equivalent to learning the ground-truth preferences $\mathbb{P}_*$ (Rafailov et al., 2023). Taking this a step further, Yun et al. (2025) reframe the alignment problem as explicitly learning $\pi^*$, and we build upon this viewpoint, deriving our results in terms of convergence of the learned policy $\hat{\pi}$ to $\pi^*$. This *distribution learning* perspective offers a stricter but arguably more natural notion of alignment; simple reward maximization may hide undesirable behavior such as model collapse (Section 5.1). Nonetheless, our results can be straightforwardly converted into regret bounds for $J_\gamma$ as in Xiong et al. (2024); Xie et al. (2025).

**Direct preference optimization (DPO).** Given a dataset of prompts and preferred/dispreferred response pairs $D = \{(x, a^+, a^-)\}_{i=1}^n$, the DPO objective (Rafailov et al., 2023) is defined for policy $\pi$ as

$$\mathcal{L}_{\text{DPO}}(\pi; D) :=$$
$$\sum_{(x,a^+,a^-) \in D} - \ln \sigma \left( \gamma \ln \frac{\pi(a^+ \mid x)}{\pi(a^- \mid x)} - \gamma \ln \frac{\pi_0(a^+ \mid x)}{\pi_0(a^- \mid x)} \right).$$

Hence, the DPO objective can also be seen as the maximum likelihood objective w.r.t. the induced binary preference distribution $\mathbb{P}_{\pi,\pi_0}(\succ | x)$.

**Policy class.** To develop our theory in a concrete manner, we first focus on the linear softmax policy class (Xiong et al., 2024; Cen et al., 2025; Foster et al., 2025) in Sections 3 and 4. This will motivate the coverage conditions under which we study general function classes in Section 5.

Let $\phi : \mathcal{X} \times \mathcal{A} \to \mathbb{R}^d$ be a feature map such that $\|\phi(x, a)\|_2 \leq 1$ for all $x, a$ and fix a radius $R > 0$. We study parametrized policies in the class

$$\Pi := \{\pi_\theta : \mathcal{X} \to \Delta(\mathcal{A}) \mid \theta \in B_2(0, R),$$
$$\pi_\theta(a \mid x) \propto \pi_0(a \mid x) \exp(\theta^\top \phi(x, a))\}.$$

**Assumption 2** (Realizability). $\pi^* = \pi_{\theta^*}$ *for some* $\theta^* \in B_p(0, R)$ *and* $p \in [1, 2]$.

While $p = 2$ is most natural, our analysis can be applied to any geometry with the corresponding coverage, and the lower bound for offline learning is most naturally presented w.r.t. 1-norm, hence we allow for arbitrary $p$. Assumption 2 implies $r^*(x, \cdot) = \gamma(\theta^*)^\top \phi(x, \cdot) + c(x)$ for some function $c$ of $x$. If $\pi^*$ is not realizable, our results can still be modified to include an additive error term.

**Generalized coverage.** We require a notion of feature-level coverage, which is an instance of often-studied *generalized coverage* conditions (Xie et al., 2021; Uehara et al., 2022; Jiang & Xie, 2025; Agarwal et al., 2025). For a policy $\pi : \mathcal{X} \to \Delta(\mathcal{A})$, the feature covariance is

$$\mathbb{V}(\pi) := \mathbb{E}_{x \sim \mathcal{D}, a \sim \pi(x)} \left[ (\phi(x, a) - \mathbb{E}_{a \sim \pi(x)}[\phi(x, a)])^{\otimes 2} \right].$$

We define the (single) coverage of $\pi'$ by $\pi$, and the local $L^p$-coverage of $\pi^*$ with radius $r$, as

$$
\begin{aligned}
C_{\pi \to \pi'} &:= \inf\{C > 0 \mid \mathbb{V}(\pi') \preceq C \cdot \mathbb{V}(\pi)\}, \\
C_*^p(r) &:= \sup\{C_{\pi_\theta \to \pi^*} \mid \theta \in B_p(\theta^*, r)\}.
\end{aligned}
\tag{2}
$$

A naive bound shows that $C_{\pi \to \pi'} \leq \|\pi'/\pi\|_\infty$ and $C_*^p(r) \leq e^{2r}$ (Lemma C.8), but $C_*^p(r)$ may be much smaller than the density ratio and possibly polynomial in $r$ (Agarwal et al., 2020; Jun et al., 2021; Agarwal et al., 2025; Jiang & Xie, 2025). Our results hold regardless of the magnitude of $C_*^p$ as we show a separation between on-policy and offline DPO in the total sample complexity *overhead* $n/C_*^p(R)$.

**Assumption 3.** $\mathbb{V}(\pi^*) \succeq \lambda \mathbf{I}_d$ *for some* $\lambda > 0$.

For this condition to hold, we must have $\lambda \lesssim d^{-1}$. This may be weakened to $\mathbb{V}(\pi^*)|_S \succeq \lambda \mathbf{I}_S$ on the subspace $S = \{\mathbf{1}\}^\perp$ by always requiring the identifiability condition $\theta^\top \mathbf{1} = 0$; all subsequent analyses will still hold when restricting to $S$. We discuss how to avoid this assumption entirely using optimal design in Section 4.

In order to study the dynamics of coverage in Section 3, we assume a mild growth condition for $C_*^p$:

**Assumption 4.** *The map* $r \mapsto \sqrt{C_*^p(r)}$ *is convex on* $[0, R]$. *More generally, for some* $\kappa \geq 1$, *the set* $\{r \in [0, R] : C_*^p(r)/r^2 \leq \kappa \cdot C_*^p(R)/R^2\}$ *is a connected interval.*

This assumption is weak in the sense that if it does not hold, we may simply replace $C_*^p$ by any upper bound $\bar{C}_*^p \geq C_*^p$ with $\bar{C}_*^p(0) = 1$ which is square root-convex, and our results will still be valid. For instance, the naive bound $\bar{C}_*^p(r) = e^{2r}$ always suffices.

## 2.2. Minimax Lower Bound for Offline Preference Learning

In this section, we show a information-theoretic lower bound for general offline preference learning methods, including vanilla DPO. Under the Bradley-Terry model, Foster et al. (2025, Theorem 2.1) as well as Xie et al. (2025, Proposition 2.1) give a worst-case proof that to learn a reward-maximizing policy, any (online or offline) learner starting

from $\pi_0$ requires an exponentially large (w.r.t. inverse temperature) number $n_0$ of samples, roughly equal to the target coverage $C_{\pi_0 \to \pi^*}$. Their argument considers a skewed two-armed bandit policy $\pi_0$ from which it is unlikely to sample the optimal action even once. However, these results do not quantify the *convergence* of error once $n > n_0$, and hence cannot capture the difference between online and offline DPO in the practical regime when DPO is actually effective. Here, we obtain a worst-case convergence rate in $n$ which applies to any purely offline method, and clarify the dependency on coverage and dimension.

Consider the bandit setting where $\mathcal{X} = \{\bot\}$, $\mathcal{A} = \{1, \cdots, d\}$ and features $\phi(i) = e_i \in \mathbb{R}^d$. We will construct a target hypothesis class $\Theta^* \subset B_p(0, R)$ consisting of near-uniform $\pi^*$ candidates, and an adversarial base policy $\pi_0$ from which worst-case coverage sample complexity is required to obtain information on $\pi^*$. For such $\pi^*$, we verify Assumptions 3,4 (with $\lambda \asymp 1/d$) and tightly bound the local $L^p$-coverage via the following result:

**Proposition 2.1.** *Let $\pi_\theta = \mathrm{softmax}(\theta)$ for $\theta \in \mathbb{R}^d$. Then if $\|\theta^*\|_\infty = O(1)$ and $d = O(\mathrm{poly}(R))$, it holds that $\mathbb{V}(\pi^*)|_S \gtrsim \frac{1}{d}\mathbf{I}_S$ and $C_*^p(R) \asymp e^R$ when $p = 1$, or $C_*^p(R) \asymp \frac{1}{d}e^{2^{1-1/p}R}$ when $p > 1$.*

We now show the lower bound, which applies to all estimators which rely on pairwise preferences sampled from the base policy $\pi_0$. If the sampling policy is different, for example if the responses in the dataset were collected ahead of time from a different LLM, the coverage term should be replaced by the corresponding coverage. The proof is an application of the Yang-Barron method (Yang & Barron, 1999) to the hypothesis class $\Theta^*$ equipped with the $L^p$-norm.

**Theorem 2.2** (Lower bound for offline preference learning)**.** *Let the dataset $D$ consist of $n$ i.i.d. pairs of actions $a^1, a^2 \sim \pi_0$ labeled by $\mathbb{P}_*$. Then for $p > 1$, over all estimators $\hat{\pi}(D) := \pi_{\hat{\theta}(D)}$ of $\pi^*$,*

$$\inf_{\hat{\theta}} \sup_{\theta^* \in \Theta^*} \mathbb{E}_{D \sim \pi^*}\left[\|\hat{\theta} - \theta^*\|_1\right] \gtrsim d\sqrt{\frac{C_*^1(R)}{n}},$$

$$\inf_{\hat{\pi}} \sup_{\theta^* \in \Theta^*} \mathbb{E}_{D \sim \pi^*}\left[\mathrm{KL}(\pi^* \| \hat{\pi}(D))\right] \gtrsim \frac{C_*^1(R)}{n} \wedge \frac{1}{d^2}.$$

*For $p > 1$, we instead have*

$$\inf_{\hat{\theta}} \sup_{\theta^* \in \Theta^*} \mathbb{E}_{D \sim \pi^*}\left[\|\hat{\theta} - \theta^*\|_p\right] \gtrsim \sqrt{\frac{dC_*^p(R)}{n}}.$$

When $p = 1$, we recover the ordinary parametric $L^1$ minimax risk but multiplied with the corresponding coverage $C_*^1(R)$, reflecting the difficulty of receiving a positive signal when sampling offline from $\pi_0$. Here the KL bound is the 'correct' rate since an extra $d$ factor is incurred when converting from $L^2$ to $L^1$-norm, and another from KL curvature

(Lemma F.1). This result is tight: Yun et al. (2025) essentially show a KL upper bound of $\frac{d}{n} \times$(coverage) for their PMLE method, which is equivalent to offline DPO without the base policy $\pi_0$, and the $d$ factor (arising from their policy class size $\ln|\Pi|$) can be removed for near-uniform $\pi^*$ via Lemma F.1.

In contrast, when $p > 1$, the rate is 'suboptimal' in the sense that it is dimension-free. This is not an artifact of the proof, but rather a price that must be paid to manifest the worst-case dependence on $C_*^p(R)$. Intuitively, when $\pi^*$ is roughly uniform, the largest density ratio with $\pi_0 = \mathrm{softmax}(\varphi)$ such that $\theta^* \in B_p(\varphi, R)$ is achieved only when $\varphi \approx (2^{1/p}R, -2^{-1/p}R, 0, \cdots, 0)$ or a permutation thereof; however, only one coordinate needs to be resolved by paying coverage, so the dimensional dependency is lost. When $p = 1$, however, we can instead use $\varphi \approx (R, 0, \cdots, 0)$ so that $\pi_0$ has $d$ equally unlikely indices that need to be sampled from, and we recover a confusion factor of $d$.

## 3. Fast Convergence of On-policy DPO

By Theorem 2.2, any offline preference learner must suffer a slow convergence rate of $\sqrt{C_*^p(R)/n}$. In contrast, given access to real-time feedback such as AI feedback (Lee et al., 2024; Guo et al., 2024), an online learner can update its sampling policy based on the first batch of $n \approx C_*^p(R)$ preferences (which already contain some information on $\pi^*$) to one with more favorable coverage. This ensures the next on-policy batch is more informative, and iterating this process can achieve much faster convergence.

We argue that on-policy DPO and existing related methods (Guo et al., 2024; Xu et al., 2024a; Calandriello et al., 2024; Rosset et al., 2024; Swamy et al., 2024) already benefit from this *coverage improvement principle*, without requiring any exploratory modifications typically considered in the literature (Xie et al., 2022; Xiong et al., 2024; Xie et al., 2025; Shi et al., 2025; Foster et al., 2025; Cen et al., 2025). We focus on DPO for simplicity of presentation, but our techniques can be applied to any batched-online and on-policy preference learning method.

**On-policy DPO.** We study a basic batched-online version of DPO, described in Algorithm 1, where each new batch is sampled purely on-policy and used to iteratively update the current policy with the standard $\mathcal{L}_{\mathrm{DPO}}$ objective; this algorithm is also referred to as online DPO, iterative DPO, etc., in the literature (Guo et al., 2024; Xu et al., 2024a). We assume the feedback oracle is exact and assigns preferences according to $\mathbb{P}_*$, otherwise, the learned preferences will end up converging to that of the oracle. We remark that $\mathcal{L}_{\mathrm{DPO}}$ is strongly convex in $\theta$ in the constrained linear softmax setting, and thus can be efficiently optimized using standard methods due to the convexity of the domain.

**Algorithm 1** On-policy Direct Preference Optimization

**Require:** initial policy $\pi_0$, features $\phi$, radius $R$, norm $p$, iterations $K$, batch sizes $n_k$

1: **for** $k = 0, \cdots, K-1$ **do**
2:     Sample size $n_k$ preference dataset $D_k$ from $\hat{\pi}_k$:
3:     **for** $i = 1, \cdots, n_k$ **do**
4:         Sample $x \sim \mathcal{D}$, $a^1, a^2 \sim \hat{\pi}_k$
5:         Receive preferences from oracle; sets $(a^+, a^-) = (a^1, a^2)$ with probability $\mathbb{P}_*(a^1 \succ a^2 \mid x)$ and $(a^+, a^-) = (a^2, a^1)$ otherwise
6:     Update $\hat{\pi}_{k+1} = \arg\min_{\pi \in \Pi} \mathcal{L}_{\text{DPO}}(\pi; D_k)$
7: **Return** $\hat{\pi}_K$

### 3.1. Convergence Analysis

We first derive the error guarantee for a single update step depending on the coverage of the previous policy. The proof utilizes techniques from Foster & Krishnamurthy (2021); Yun et al. (2025) to bound the error of the learned preferences $\mathbb{P}_{\hat{\pi}_k, \pi_0}$, which is related back to the error of $\hat{\theta}_k$ via the feature covariance coverage in Eq. (2). We denote $\hat{\pi}_k = \pi_{\hat{\theta}_k}$ and $\mathcal{L}_{\text{DPO}}(\theta; D_k) = \mathcal{L}_{\text{DPO}}(\pi_\theta; D_k)$ by abuse of notation.

**Proposition 3.1** (One-step policy improvement). *Assume $\hat{\pi}_k \in \Pi$ satisfies $\hat{\theta}_k \in B_p(\theta^*, r_k)$ with radius $r_k \in [0, R]$. Suppose $D_k$ consists of $n_k$ i.i.d. samples of prompts $x \sim \mathcal{D}$ and pairs of responses $a^+, a^-$ sampled from $\hat{\pi}_k(x)$ and labeled using $\mathbb{P}_*$. Then it holds with probability at least $1 - \delta$ that the update $\hat{\theta}_{k+1} = \arg\min_{\theta \in B_p(0,R)} \mathcal{L}_{\text{DPO}}(\theta; D_k)$ satisfies $\hat{\theta}_{k+1} \in B_p(\theta^*, r_{k+1})$, where*

$$r_{k+1}^2 := \frac{52 e^{4\gamma R} d^{2/p}}{\lambda \gamma^2 n_k} C_*^p(r_k) \ln \frac{9\gamma R n_k}{\delta} \qquad (3)$$

*as long as $r_{k+1} \leq \frac{1}{2\gamma} \wedge R$.*

Next, we iteratively apply Proposition 3.1 to construct a sequence of shrinking confidence regions $B_p(\hat{\theta}_k, r_k)$ for $\theta^*$ and conduct a fixed-point analysis of the recursion $r_{k+1}^2 \asymp C_*^p(r_k)$ in Eq. (3) to conclude our main convergence result.

**Theorem 3.2** (Fast convergence of on-policy DPO). *Fix any $\eta, \delta \in (0,1)$ and $K \in \mathbb{N}$ and suppose $\sqrt{C_*^p(R)} \wedge R \gtrsim 1/\gamma$. Then with probability $1 - \delta$, for all $n$ such that*

$$\xi_n := \frac{52 e^{4\gamma R} d^{2/p}}{\lambda \eta^2 n} \ln \frac{9\gamma RKn}{\delta} \lesssim \frac{1}{C_*^p(R)}, \qquad (4)$$

*the output of running $K$ iterations of on-policy DPO (Algorithm 1) with $n_k \equiv n$ satisfies*

$$\text{KL}(\hat{\pi}_K \| \pi^*) \leq \|\hat{\theta}_K - \theta^*\|_p^2 \leq e^2 \gamma^{-2} \xi_n \vee R^2 \eta^{2K}.$$

In particular, the first term is $\widetilde{O}(R^2/n)$ for $n \gtrsim C_*^p(R)$ rather than $C_*^p(R)/n$ as in Theorem 2.2, while the second

term exhibits exponential convergence in $K$. We remark that the same bound holds for forward KL as well as $\chi^2$-divergence (Proposition C.9). Focusing on the dependence on $R$, we obtain the following corollary on total sample complexity of on-policy versus offline DPO:

**Corollary 3.3.** *Suppose $d = O(\text{poly}(R))$ and $\gamma \asymp 1/R$. Offline DPO needs $n_{\text{off}} \geq \Omega\left(\frac{1}{\varepsilon^2} C_*^p(R)\right)$ samples to achieve $\|\hat{\theta} - \theta^*\|_p \leq \varepsilon$. In contrast, on-policy DPO requires batch size $n = \widetilde{O}\left(\frac{R^2}{\varepsilon^2} \vee C_*^p(R)\right)$ and number of iterations $K = O\left(\ln \frac{R}{\varepsilon}\right)$, hence the total complexity $n_{\text{on}}$ satisfies*

$$n_{\text{on}} = \widetilde{O}\left(\frac{R^2}{\varepsilon^2} \vee (\ln \frac{R}{\varepsilon}) C_*^p(R)\right) \ll n_{\text{off}}.$$

In other words, while both methods require $n \gtrsim C_*^p(R)$ to enable learning (which is inevitable when sampling from $\pi_0$ as per Xie et al. (2025)), offline DPO requires an additional $1/\varepsilon^2$ multiplicative factor to bring the loss down to $\varepsilon$, while on-policy DPO requires only a $\ln(1/\varepsilon)$ factor in the number of iterations; an exponential separation in overhead!

**Remark 3.4.** The decay factor $\eta$ can be chosen to be arbitrarily small as long as Eq. (4) is satisfied. In particular, we may choose $\eta \sim 1/\ln K$ and achieve a *super-exponential* convergence rate in $K$ while only incurring an additional log factor in the required batch size $n$.

**Remark 3.5.** The recurring factor of $e^{2\gamma R} =: \kappa_\sigma$ is a crude bound for the curvature of the Bradley-Terry model, often appearing in analyses of logistic models (Faury et al., 2020; Russac et al., 2021; Chatterji et al., 2022). We choose to hide this dependency in the sequel by assuming $\gamma \asymp 1/R$, which essentially says $r^*$ is bounded. Indeed, $\gamma$ is typically set to be quite small, between $0.01$ and $0.1$ in standard implementations (Ouyang et al., 2022; Rafailov et al., 2023; von Werra et al., 2025). If this cannot be assumed, DPO requires an additional $\kappa_\sigma^2$ factor. This is still relatively mild compared to coverage, which can be $e^{\Theta(R)}$ in the worst case (Proposition 2.1). In this scenario, on-policy DPO still improves on offline when $\gamma < \frac{1}{4}$ and target loss $\varepsilon \lesssim \kappa_\sigma^{-2}$.

**Faster convergence under super-quadraticity.** Under a slightly stronger super-quadratic assumption on $C_*^p$,[2] we can use an exponential decay schedule for the batch size $n_k$ and achieve the same error with *constant* overhead (i.e., with total number of samples equal to the batch size of a single round of on-policy DPO in Theorem 3.2):

**Corollary 3.6.** *Suppose $\gamma \asymp 1/R$ and $C_*^p$ is super-quadratic, i.e., $C_*^p(r)/r^{2+\alpha}$ is nondecreasing for $r \geq r_0$ for some $\alpha > 0$. Let $n_i, n_f$ be batch sizes satisfying $\xi_{n_i} \lesssim 1/C_*^p(R)$ and $\xi_{n_f} \lesssim \varepsilon^2$ as required in Theorem 3.2.*

---

[2] The naive upper bound $C_*^p(R) \lesssim e^{2^{1-1/p}R}$ (Proposition 2.1) is super-quadratic. In fact, we can use doubly exponential decay if $C_*^p$ exhibits exponential growth; see the proof of Corollary 3.6.

*Then running Algorithm 1 with $n_k = \eta^{\alpha k} n_i$ for $k \lesssim \ln R$ iterations followed by a final batch of size $n_f$ guarantees $\|\hat{\theta} - \theta^*\|_p \lesssim \varepsilon$.*

Compared to Theorem 3.2, in order to achieve error $\varepsilon$, (1) the total number of iterations is reduced from $O(\ln \frac{R}{\varepsilon})$ to $O(\ln R)$, independent of $\varepsilon$; and (2) the total sample complexity is improved from $(\ln \frac{R}{\varepsilon}) \cdot O(n_i \vee n_f)$ to $O(n_i) + n_f$.

Intuitively, the batch size required to ensure improvement $r_{k+1} \leq \eta r_k$ is $n_k \gtrsim C_*^p(r_k)/r_k^2$, which is a U-shaped function of $r_k$. When $r_k \gg 1$, this is dictated by the coverage $C_*^p(r_k)$, which decreases exponentially with $k$ due to super-quadraticity. Once $r_k = O(1)$, the coverage becomes $\Theta(1)$ and one final round suffices to guarantee precision $1/\varepsilon^2$ independently of the initial coverage. Hence the total complexity is dominated by the initial and final batch sizes $n_i, n_f$. From a practical standpoint, this indicates that online preference learning may benefit from seeing more samples *early* in training when coverage is bad, and *late* in training when we want to converge with high precision.

## 4. Two-Step Convergence with Preferential Optimal Design

A substantial body of work has focused on developing online preference learning algorithms based on principles such as optimism or active learning to remove the dependence on coverage (Song et al., 2024; Xiong et al., 2024; Xie et al., 2022; 2024; Cen et al., 2025; Foster et al., 2025). While the analysis in Section 3 applies directly to on-policy methods without any exploratory modifications, it also motivates the design of more efficient coverage-free algorithms.

In this section, we present a new hybrid DPO variant which avoids the dependence on $\lambda$ (Assumption 3) as well as coverage, and achieves polynomial convergence in just two rounds. The algorithm itself, detailed in Algorithm 2, is exceedingly simple: replace sampling from $\hat{\pi}_k$ in on-policy DPO with a mixture $\frac{1}{2}\hat{\pi}_k + \frac{1}{2}\pi^g$ for a specific joint distribution $\pi^g \in \Delta(\mathcal{X} \times \mathcal{A})$. Since $\pi^g$ is fixed, the offline portion of the dataset can be constructed entire before training.

The core novelty is our construction of $\pi^g$ to uniformly minimize prediction variance for preference learning. As a consequence, sampling from $\pi^g$ will ensure good coverage in round one, which in turn guarantees small error in round two. We build upon the principle of G-optimal experimental design (Smith, 1918; Kiefer & Wolfowitz, 1960), in particular the Kiefer-Wolfowitz equivalence (Theorem D.1).

**Constructing the preferential design.** For a conditional law $\pi \in \Delta(\mathcal{A})$, denote $\phi(x, \pi) := \mathbb{E}_{a \sim \pi}[\phi(x, a)]$ and $\mathbb{V}(x, \pi) := \mathbb{E}_\pi[(\phi(x, a) - \phi(x, \pi))^{\otimes 2}]$. For a *joint* law $\pi \in \Delta(\mathcal{X} \times \mathcal{A})$ with $x$-marginal $\pi_\mathcal{X}$, we abuse notation and write $\mathbb{V}(\pi) = \mathbb{E}_{x \sim \pi_\mathcal{X}, a \sim \pi(x)}[(\phi(x, a) - \phi(x, \pi))^{\otimes 2}]$ as

---

**Algorithm 2** DPO with Preferential Optimal Design

**Require:** base policy $\pi_0$, features $\phi$, batch size $n$

1: **for** $x \in \mathcal{X}$ **do**
2:     Compute G-optimal design $\pi^g(x) \in \Delta(\mathcal{A})$: solve $\arg\min_\pi \max_a \|\phi(x, a) - \phi(x, \pi)\|_{\mathbb{V}(x, \pi)^{-1}}$
3:     Set $\phi^g(x, \cdot) = \phi(x, \cdot) - \phi(x, \pi^g(x))$
4: Compute G-optimal design $\mu^g \in \Delta(\mathcal{X} \times \mathcal{A})$: solve $\arg\min_\mu \sup_{x,a} \|\phi^g(x, a)\|_{\mathbb{E}_\mu[\phi^g(x,a)^{\otimes 2}]^{-1}}$
5: Set $\pi^g(x, a) = \mu^g_\mathcal{X}(x) \pi^g(a \mid x)$
6: **for** $k = 0, 1$ **do**
7:     Sample size $n$ preference dataset $D'_k$ from $\hat{\pi}'_k(x, a) = \frac{1}{2}\mathcal{D}(x)\hat{\pi}_k(a \mid x) + \frac{1}{2}\pi^g(x, a)$
8:     Update $\hat{\pi}_{k+1} = \arg\min_{\pi \in \Pi} \mathcal{L}_{\text{DPO}}(\pi; D'_k)$
9: **Return** $\hat{\pi}_2$

---

well as $\phi(\pi) := \mathbb{E}_{(x,a) \sim \pi}[\phi(x, a)]$; policies $\pi : \mathcal{X} \to \Delta(\mathcal{A})$ can be seen as having $x$-marginal $\mathcal{D}$ by default.

We first show a Kiefer-Wolfowitz theorem *with centering* for the features $\phi(x, a)$ with $x$ fixed.

**Lemma 4.1.** *For any $x \in \mathcal{X}$ it holds that*

$$\min_{\pi \in \Delta(\mathcal{A})} \max_{a \in \mathcal{A}} \|\phi(x, a) - \phi(x, \pi)\|^2_{\mathbb{V}(x, \pi)^{-1}} = d.$$

We denote this *conditional* G-optimal design as $\pi^{cg}(x) = \pi^{cg}(\cdot \mid x)$ for each $x \in \mathcal{X}$. The (rather unintuitive) idea to now construct the desired $\pi^g \in \Delta(\mathcal{X} \times \mathcal{A})$ is to extract the $x$-marginal from the *joint* or *global* G-optimal design of the $\pi^{cg}(x)$-centered features $\phi^g(x, a) := \phi(x, a) - \phi(x, \pi^{cg}(x))$:

**Proposition 4.2** (Preferential G-optimal design). *Let $\mu^g \in \Delta(\mathcal{X} \times \mathcal{A})$ be a global design satisfying*

$$\sup_{x \in \mathcal{X}, a \in \mathcal{A}} \phi^g(x, a)^\top M(\mu)^{-1} \phi^g(x, a) = d,$$
$$M(\mu) := \mathbb{E}_{(x,a) \sim \mu}\left[\phi^g(x, a)\phi^g(x, a)^\top\right].$$

*The preferential G-optimal design $\pi^g \in \Delta(\mathcal{X} \times \mathcal{A})$ is defined as $\pi^g(x, a) := \mu^g_\mathcal{X}(x)\pi^{cg}(a \mid x)$ where $\mu^g_\mathcal{X}$ is the $x$-marginal of $\mu^g$. Then $\pi^g$ satisfies*

$$\sup_{x \in \mathcal{X}, a \in \mathcal{A}} \phi^g(x, a)^\top \mathbb{V}(\pi^g)^{-1} \phi^g(x, a) \leq d^2.$$

**Remark 4.3.** More precisely, from a design perspective, we actually want to choose an $x$-marginal $\mu_\mathcal{X}$ which optimizes

$$\mu_\mathcal{X} \in \arg\min_{\mu \in \Delta(\mathcal{X})} \sup_{x,a} \|\phi^g(x, a)\|^2_{\mathbb{V}(\mu \times \pi^{cg})^{-1}}, \qquad (5)$$

and Proposition 4.2 guarantees that the optimal value is at most $d^2$, achieved by the $x$-marginal from the global G-optimal design with centering $\mu^g$. We leave open the

question of whether Eq. (5) can be improved to $O(d)$ (hence removing the $d$ factor in Theorem 4.4). We also remark that both the global approach and Eq. (5) can be efficiently optimized using the Frank-Wolfe or other convex optimization algorithms, which yield an approximate solution (i.e., up to a constant factor) in $\widetilde{O}(d)$ time (Todd, 2016; Lattimore & Szepesvári, 2020). However, our current approach also requires solving for $\pi^{cg}(x)$ for all $x \in \mathcal{X}$; whether we can avoid the full dependency on $|\mathcal{X}|$, perhaps under additional structural conditions on $\phi$, is left to future work.

With these results in place, we now state the error guarantee (under $\mathbb{V}(\pi^*)$-norm) for Algorithm 2.

**Theorem 4.4.** *Suppose $\gamma \asymp 1/R$ and $\varepsilon \lesssim 1/d$. With probability at least $1 - \delta$, the output of Algorithm 2 satisfies $\|\hat{\theta}_2 - \theta^*\|_{\mathbb{V}(\pi^*)} \leq \varepsilon$ when $n \gtrsim \widetilde{O}\left(\frac{dR^2}{\varepsilon^2} \ln \frac{1}{\delta}\right)$.*

Intuitively, the proof is as follows: sampling from $\hat{\pi}'_{k-1}$ ensures that both $\|\hat{\theta}_k - \theta^*\|_{\mathbb{V}(\pi^g)}$ and $\|\hat{\theta}_k - \theta^*\|_{\mathbb{V}(\hat{\pi}_{k-1})}$ are small by applying the techniques from Proposition 3.1. Due to the $\phi^g$-variance minimization property of $\pi^g$, the former with $k = 1$ can be used to show the policy $\hat{\pi}_1$ constructed from $\hat{\theta}_1$ has good coverage. This turns the latter with $k = 2$ into a $\mathbb{V}(\pi^*)$-norm guarantee. Therefore, we are able to avoid dependency on both coverage and $\lambda$ (which can be much worse than $d^{-1}$) and achieve polynomial complexity.

Together, Proposition 4.2 and Theorem 4.4 establish the existence of an offline sampler $\pi^g$ that guarantees near-optimal coverage for all future rounds and which can be fully computed before training, demonstrating the efficiency of online preference learning with well-designed offline data (cf. Ball et al. (2023)). We mention that this also improves upon the GSHF algorithm due to Xiong et al. (2024), which achieves $\varepsilon$-regret with $O(d/\varepsilon^2)$ batch size but requires $\widetilde{\Omega}(d)$ rounds of training, and needs to re-compute the exploration policy after each round.

# 5. Fast Convergence of On-policy Reward Distillation

While direct preference learning methods are attractive due to their simplicity and stability, they still suffer from out-of-distribution or degeneracy issues (Xu et al., 2024b). A recent line of work has instead proposed a class of methods termed *reward model distillation* (Gao et al., 2024; Fisch et al., 2025; Yun et al., 2025), which trains the implicit reward model of the policy to fit reward differences in paired responses. These retain much of the simplicity of DPO as they only need an already-trained reward model $\tilde{r}$ and do not require policy gradient estimation, and match or outperform DPO baselines (Gao et al., 2024; Fisch et al., 2025).

In this section, we extend our coverage analysis to understand the benefits of on-policy reward distillation. Our key insight is that distillation is fundamentally noiseless since the learner accesses relative rewards directly instead of sampling stochastic preferences from Eq. (1), and thus should achieve fast rates in the sense of van Erven et al. (2015), that is, $O(n^{-1})$ instead of $O(n^{-1/2})$.

## 5.1. Distilling Rewards: REBEL, VCB, PD

**REBEL and VCB.** The REBEL algorithm (Gao et al., 2024) aims to iteratively solve the KL-constrained reward maximization problem $\arg\max_\pi \mathbb{E}_{x\sim\mathcal{D},a\sim\pi(x)}[\tilde{r}(x,a)] - \gamma\text{KL}(\pi\|\hat{\pi}_k)$, where $\tilde{r}$ is a queryable reward model and $\hat{\pi}_k$ is the previous round's learned policy. Given access to paired samples $(x, a^1, a^2)$, REBEL regresses the log-likelihood ratio of $a^1, a^2$ (regularized with the previous round) on the difference in rewards by minimizing the squared loss

$$\mathbb{E}_{x\sim\mathcal{D},\, a^1,a^2\sim\hat{\pi}_k(x)}$$
$$\left[\left(\gamma\ln\frac{\pi(a^1\mid x)}{\pi(a^2\mid x)}\frac{\hat{\pi}_k(a^2\mid x)}{\hat{\pi}_k(a^1\mid x)} - \tilde{r}(x,a^1) + \tilde{r}(x,a^2)\right)^2\right].$$

Mao et al. (2024) study VCB, the offline version of REBEL where $\hat{\pi}_k$ is replaced with the SFT policy; a pessimistic variant has also been developed by Fisch et al. (2025). REBEL can be interpreted as an adaptive policy gradient method approximating mirror descent, and the expected reward $J(\hat{\pi}_k) = \mathbb{E}_{x\sim\mathcal{D},a\sim\hat{\pi}_k(x)}[r^*(x,a)]$ is shown to converge at a $O(K^{-1/2})$ rate to its maximum (Gao et al., 2024).

However, we claim that this iteration is flawed from a distribution learning perspective. Assuming each step is solved perfectly, the model learns the RLHF solution $\hat{\pi}_{k+1} \propto \hat{\pi}_t \exp(\gamma^{-1}\tilde{r})$; running this update $K$ times will compound the tilt to yield $\hat{\pi}_K \propto \pi_0 \exp(K\gamma^{-1}\tilde{r})$, collapsing onto a one-hot distribution supported only on the maximum reward response as $K \to \infty$. Each preference distribution $\mathbb{P}_{\pi,\pi_0}(a^1 \succ a^2 \mid x)$ will also collapse on whichever of $a^1, a^2$ has higher reward. Even though this indeed maximizes expected reward, the model fails to learn the target *distributions* $\pi^*$ and $\mathbb{P}_*$, resulting in undesirable behavior.[3] We also should not regularize on-policy DPO with the previous policy instead of the base policy, as done in Gao et al. (2024), for the same reason.

**Preference distillation (PD).** Yun et al. (2025) propose an alternative method of *preference distillation* (PD), motivated by the observation that $\tilde{r}$ can be used to simulate preferences as $\tilde{\mathbb{P}}(a^1 \succ a^2 \mid x) = \sigma(\tilde{r}(x,a^1) - \tilde{r}(x,a^2))$. Instead of sampling preferences from $\text{Bern}(\tilde{\mathbb{P}})$ and running DPO, however, they use the expectation of the DPO objective over preference labeling, which can be viewed as directly using

---

[3]This is different from the degeneracy issue described in Fisch et al. (2025) for offline DPO, which applies only to the tabular and unconstrained policy class setting.

$\tilde{\mathbb{P}}$ as soft preference labels:

$$\mathbb{E}_{x,a^1,a^2 \sim \pi_0(x)} \left[ -\tilde{\mathbb{P}}(a^1 \succ a^2 \mid x) \ln \sigma\left(\gamma \ln \frac{\pi(a^1 \mid x)}{\pi(a^2 \mid x)}\right) \right.$$
$$\left. -\tilde{\mathbb{P}}(a^2 \succ a^1 \mid x) \ln \sigma\left(\gamma \ln \frac{\pi(a^2 \mid x)}{\pi(a^1 \mid x)}\right) \right].$$

However, they perform off-policy optimization without considering iterative updates, and use KL regularization rather than the DPO regularization scheme. Moreover, their obtained convergence rate for KL implies a slow $O(n^{-1/2})$ rate in our setting via Jensen's inequality, which does not take advantage of the noiseless distillation step.

**Designing on-policy reward distillation methods.** Combining the above works, we can give a general recipe for designing (pairwise relative) reward distillation algorithms. Let $D$ be a dataset consisting of queries and pair of responses $(x, a^1, a^2)$ and $\ell : \mathbb{R} \times \mathbb{R} \to \mathbb{R}_{\geq 0}$ be any loss function such that $\ell(y, z) = 0$ iff $y = z$. We define the reward distillation objective as

$$\mathcal{L}_{\mathrm{RD}}(\pi; \pi', r, D) :=$$
$$\sum_D \ell\left(\gamma \ln \frac{\pi(a^1 \mid x)}{\pi(a^2 \mid x)} \frac{\pi'(a^2 \mid x)}{\pi'(a^1 \mid x)}, \, r(x, a^1) - r(x, a^2)\right).$$

For instance, the REBEL update can be written as $\hat{\pi}_{k+1} = \arg\min_{\pi \in \Pi} \mathcal{L}_{\mathrm{RD}}(\pi; \hat{\pi}_k, \tilde{r}, D_k)$ where $\ell$ is squared loss and $D_k$ is sampled on-policy from the most recent policy $\hat{\pi}_k$. One can also see that preference distillation can also be cast in the above form with the binary KL loss $\ell(y, z) = \mathrm{KL}(\mathrm{Bern}(\sigma(y)) \| \mathrm{Bern}(\sigma(z)))$. As discussed before, the REBEL scheme induces degeneracy in the limit; fortunately, there are two straightforward ways to remedy this issue. One is to simply use $\pi_0$ for regularization:

$$\hat{\pi}_{k+1} = \arg\min_{\pi \in \Pi} \mathcal{L}_{\mathrm{RD}}(\pi; \pi_0, \tilde{r}, D_k) \quad \text{(fixed-regularization)}$$

similar to on-policy DPO. However, one may still desire to use the improved policy $\hat{\pi}_k$ for regularization. In this scenario, $\tilde{r}$ can be replaced with a *modified* reward $\hat{r}_k$ obtained by subtracting the implicit reward estimate from the last step to extract the 'effective' training signal:

$$\hat{\pi}_{k+1} = \arg\min_{\pi \in \Pi} \mathcal{L}_{\mathrm{RD}}(\pi; \hat{\pi}_k, \hat{r}_k, D_k), \quad \hat{r}_k := \tilde{r} - \gamma_c \ln \frac{\hat{\pi}_k}{\pi_0}$$
$$\text{(reward-calibration)}$$

where $\gamma_c$ is the *calibration level*. When $\gamma_c = \gamma$, this update is equivalent to the fixed-regularization update if $\ell$ is shift-invariant as for REBEL, or in the realizable setting where zero loss is achieved. In practice, we implement $\gamma_c$ as a tunable hyperparameter. Empirically, this helps stability by controlling rewards in a flexible manner and reducing signal variance as the target difference $\hat{r}_k(x, a^1) - \hat{r}_k(x, a^2)$ grows

---

**Algorithm 3** On-policy Reward Distillation

**Require:** initial policy $\pi_0$, policy class $\Pi$, reward model $\tilde{r}$, iterations $K$, batch size $n$, loss $\ell$

1: **for** $k = 0, \cdots, K - 1$ **do**
2:      Sample size $n$ dataset $D_k$ on-policy from $\hat{\pi}_k$: $x \sim \mathcal{D}$, $a^1, a^2 \sim \hat{\pi}_k$
3:      **if** fixed-regularization **then**
4:          Update $\hat{\pi}_{k+1} = \arg\min_{\pi \in \Pi} \mathcal{L}_{\mathrm{RD}}(\pi; \pi_0, \tilde{r}, D_k)$
5:      **if** reward-calibration **then**
6:          Update $\hat{\pi}_{k+1} = \arg\min_{\pi \in \Pi} \mathcal{L}_{\mathrm{RD}}(\pi; \hat{\pi}_k, \hat{r}_k, D_k)$ where $\hat{r}_k = \tilde{r} - \gamma_c \ln(\hat{\pi}_k/\pi_0)$
7: **Return** $\hat{\pi}_K$

---

smaller. Both updates are summarized in Algorithm 3. We confirm experimentally in Appendix B that on-policy DPO and our proposed reward distillation algorithms outperform their off-policy counterparts and enjoy stable, monotonic performance gains across iterations.

### 5.2. Theoretical Setup and Convergence Analysis

For this section, we develop our results for general function classes. The reasons are twofold: first, in the finite-dimensional linear setting, $\theta^*$ can be exactly recovered by regressing against rewards with $n \gtrsim \lambda d$ samples, so iterative learning is not necessary; second, we aim to isolate the 'correct' notion of coverage – w.r.t. mean absolute deviation – that allows for fast rates, which is fundamentally different from the variance-based coverage considered in Sections 3,4 and in the literature (Agarwal et al., 2020; Uehara et al., 2022; Agarwal et al., 2025).[4]

For a function $f : \mathcal{X} \times \mathcal{A} \to \mathbb{R}$, define $\Delta f : \mathcal{X} \times \mathcal{A}^2 \to \mathbb{R}$ as $\Delta f(x, a^1, a^2) := f(x, a^1) - f(x, a^2)$. Let $\mathcal{F}$ be a class of functions such that $\|f\|_\infty \leq R$ for all $f \in \mathcal{F}$ and the class of pairwise differences $\Delta\mathcal{F} := \{\Delta f : f \in \mathcal{F}\}$ has bounded pseudodimension $\mathrm{Pdim}(\Delta\mathcal{F})$; see Appendix E.2 for definitions. We consider the associated policy class

$$\Pi_\mathcal{F} := \{\pi_f : \mathcal{X} \to \Delta(\mathcal{A}) \mid f \in \mathcal{F},$$
$$\pi_f(a \mid x) \propto \pi_0(a \mid x) \exp f(x, a)\}.$$

We assume both the target policy and reward model are realizable. The analysis can be adapted straightforwardly to non-realizable $\tilde{r}$ but must incur a slower $1/\sqrt{n}$ rate.

**Assumption 5.** $\gamma^{-1}\tilde{r} \in \mathcal{F}$ and $\pi^* = \pi_{f^*}$ for some $f^* \in \mathcal{F}$.

We also require a different notion of coverage. Let

$$\mathrm{MAD}_\pi(f) := \mathbb{E}_{x, a \sim \pi(x)}\left[\left|f(x, a) - \mathbb{E}_{a \sim \pi(x)}[f(x, a)]\right|\right]$$

---

[4]Indeed, it is straightforward to similarly generalize Section 3 to general function classes by defining coverage in terms of the variance $\mathbb{E}_{x \sim \mathcal{D}}[\mathrm{Var}_{a \sim \pi(x)} f(x, a)]$ instead of $\mathrm{MAD}_\pi(f)$, and obtain an analogous $1/\sqrt{n}$ rate.

denote the mean absolute deviation, and the single and local MAD-based coverages as

$$C_{\pi \to \pi'} := \inf\{C > 0 \mid \mathrm{MAD}_{\pi'}(f) \le C \cdot \mathrm{MAD}_\pi(f),$$
$$\forall f \in \mathcal{F} - \mathcal{F}\},$$
$$C_\mathcal{F}(r) := \sup\{C_{\pi_f \to \pi^*} \mid \mathrm{MAD}_{\pi^*}(f - f^*) \le r\}.$$

Intuitively, $C_\mathcal{F}$ measures an $L^1$-based coverage over a local $L^1$-ball, compared to $C_*^p$ in Section 2 which measures an $L^2$-based coverage over a local $L^p$-ball. $C_\mathcal{F}$ is also uniformly upper bounded by $e^{2R}$ over the class $\Pi_\mathcal{F}$ similarly to Lemma C.8. However, in the general function class setting, there is no guaranteed *convex* upper bound for $C_\mathcal{F}$ (respecting $C_\mathcal{F}(0) = 1$) since we may have large density ratio even with small absolute deviation, unlike Section 2 where we could just invoke $C_*^p(r) \le e^{2r}$. Hence we postulate:

**Assumption 6.** $C_\mathcal{F}$ admits a convex upper bound $\bar{C}_\mathcal{F}$ on $[0, R]$ such that $\bar{C}_\mathcal{F}(0) = \Theta(1)$.

As with Assumption 4, Assumption 6 still holds in the linear softmax setting (Lemma E.3). More broadly, a sufficient condition is for functions in $\mathcal{F}$ to satisfy a bound of the form $\|f\|_\infty \lesssim \|f\|_1^\alpha$ for some $\alpha \in (0, 1]$, so that $\mathrm{MAD}_{\pi^*}(f - f^*) \le r$ implies an $O(r^\alpha)$ density ratio bound, and $C_\mathcal{F}(r) = \exp(O(r^\alpha))$ indeed admits a convex upper bound by truncating at the unique inflection point. Such inequalities hold in broad generality given a metric-measure structure on the domain for function classes of generalized smoothness such as Lipschitz functions and fall under the umbrella of Gagliardo-Nirenberg inequalities (Saloff-Coste, 2001; Bakry et al., 2013; Ranjbar-Motlagh, 2023).

With these ingredients in place, we now state the analogous result to Theorem 3.2 for on-policy reward distillation.

**Theorem 5.1** (Noiseless rates for on-policy reward distillation). *Fix any* $\eta, \delta \in (0, 1)$ *and* $K \in \mathbb{N}$. *Let* $\tilde{r} \in \gamma\mathcal{F}$ *be a reward model with error* $\varepsilon_{\mathrm{RM}} := \mathrm{MAD}_{\pi^*}(\gamma^{-1}\tilde{r} - f^*)$. *Then with probability at least* $1 - \delta$, *for all* $n$ *such that*

$$\xi_n := \frac{R \, \mathrm{Pdim}(\Delta\mathcal{F})}{\eta \cdot n} \ln \frac{Kn}{\delta} \lesssim \frac{R}{C_\mathcal{F}(R)} \wedge \Theta(1),$$

*the output of* $K$ *iterations of Algorithm 3 satisfies*

$$\mathrm{MAD}_{\pi^*}(\hat{f}_k - f^*) \lesssim \xi_n \vee R\eta^K \vee \varepsilon_{\mathrm{RM}}.$$

The left-hand side is a linear measure of error, proportional to $\|\theta - \theta^*\|_2$ in the linear softmax setting (rather than quadratic as for KL; see Lemma E.3). Hence reward distillation achieves a faster $n^{-1} \vee e^{-\Omega(K)}$ rate than the $n^{-1/2} \vee e^{-\Omega(K)}$ rate of on-policy DPO, but with an irreducible error $\varepsilon_{\mathrm{RM}}$ measuring the quality of the reward model. Of course, this issue is also present for on-policy DPO as the feedback oracle may also not be well-aligned with $\mathbb{P}_*$ or even compatible with the Bradley-Terry assumption, e.g., when using LLM-as-a-Judge (Guo et al., 2024).

## 6. Conclusion

Our work provides a theoretical explanation for the strong performance of on-policy preference learning based on the coverage improvement principle: on-policy updates can rapidly improve their own data quality through better coverage. We show linear convergence of on-policy DPO in the number of iterations and establish a sharp separation in sample complexity compared to offline DPO. We also describe a simple hybrid sampler, based on a novel preferential G-optimal design, which eliminates dependence on coverage and enables two-step convergence. Finally, we develop principled on-policy schemes for reward distillation methods, and clarify how on-policy distillation can outperform learning directly from preferences by viewing each round as noiseless relative regression.

## Acknowledgments

This work was partly supported by Institute of Information & communications Technology Planning & Evaluation (IITP) grant funded by the Korea government (MSIT) (No.RS-2019-II191906, Artificial Intelligence Graduate School Program (POSTECH)). Kwang-Sung Jun was supported in part by the National Science Foundation under grant CCF-2327013. JDL acknowledges support of a Google Research Award, NSF IIS 2107304, NSF CCF 2539753, NSF CAREER Award 2540142, and NSF CCF 2019844. This research was supported by a grant from KRAFTON AI.

## Impact Statement

This paper presents work whose goal is to advance the field of Machine Learning. There are many potential societal consequences of our work, none which we feel must be specifically highlighted here.

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

# A. Related Work

## A.1. On-policy preference learning methods

We first provide a brief and non-exhaustive overview of on-policy DPO-style preference learning methods for language model alignment. One of the simplest and most widely used versions of on-policy DPO is introduced by Guo et al. (2024); Xu et al. (2024a), where each batch is sampled purely on-policy from the previous policy and used to iteratively update the current policy, with online preference feedback obtained from either an LLM annotator (Guo et al., 2024) or a classifier model (Xu et al., 2024a). Liu et al. (2024) add length regularization to address verbosity and find that on-policy DPO can improve a 7B model to perform on par with GPT-4. Calandriello et al. (2024) propose an online preference learning algorithm which combines identity preference optimization (Azar et al., 2024) and Nash mirror descent (Munos et al., 2024).

An alternative approach to on-policy DPO is proposed by Qi et al. (2024), which simulates intraspecific competition between fast and slow chasing DPO-updated models to drive learning. Another line of work by Swamy et al. (2024); Rosset et al. (2024) reformulates preference optimization as two-player zero-sum games with the preference function as the payoff, and develops iterative single-play algorithms to solve for the Nash equilibrium; these works guarantee average-iterate convergence to the minimax winner policy. Pang et al. (2024) extend iterative DPO from instruction fine-tuning to chain-of-thought reasoning tasks (Wei et al., 2023).

## A.2. Theory of on-policy preference learning

We now describe existing theoretical studies of on-policy preference learning most relevant to our work.

Xie et al. (2025) introduce XPO, an optimistic variant of DPO with online exploration, and show a regret bound in the MDP setting with sample complexity dependent on the trajectory-level coverability coefficient (best-case coverage) rather than the potentially poor coverage of the fixed reference policy. This is similar to online RL, where the existence of a distribution with good concentrability is sufficient for sample-efficient exploration (Xie et al., 2022). The XPO algorithm is further studied by Huang et al. (2024) from the viewpoint of self-improvement via distribution sharpening.

Shi et al. (2025) study convergence of gradient descent for DPO and propose an online sampler which achieves quadratic convergence. However, their analysis crucially relies on a tabular softmax parametrization, which is unrealistic for large prompt/response spaces, as well as access to exact population gradients. In contrast, our approach studies the *statistical* convergence or sample complexity of on-policy DPO in the finite-sample regime, and under a log-linear feature model (Section 3) or general function class (Section 5) setting.

Xiong et al. (2024) obtain statistical guarantees for offline RLHF/DPO with pessimism as well as online exploration methods in the contextual bandit setting with linear softmax policies. Their iterative GSHF algorithm utilizes active exploration to guarantee $\varepsilon$ suboptimality with $O(d/\varepsilon^2)$ batch size independent of data coverage (Theorem 3). However, their algorithm requires $\widetilde{\Omega}(d)$ iterations and needs to re-compute the exploration policy $\hat{\pi}_k^2$ based on the collected prompts and current (exploitation) policy $\hat{\pi}_k^1$. In contrast, our preferential G-optimal design (Section 4) achieves convergence with a comparable batch size but only two iterations, and only requires computing a fixed design $\pi^g$ beforehand, resulting in a much simpler algorithm.

Also in the linear contextual bandit setting, Foster et al. (2025) study the computational-statistical tradeoff for online alignment methods by distinguishing data efficiency, or reward model queries, and computational efficiency, or sampling oracle queries; the complexity of the latter is shown to be lower bounded by coverage. They propose an inference-time exploration algorithm based on active learning principles which achieves near-optimal (i.e., up to polynomial dependence) query complexity for both oracles. Under the same setting, Cen et al. (2025) propose value-incentivized preference optimization, the online version of which is an optimistic variant of per-sample online DPO, and obtain regret bounds. The algorithm encourages exploration by effectively adding a negative KL regularizer to the DPO loss. Earlier works by Zhan et al. (2024); Wu & Sun (2024) also utilize experimental design or active learning for preference-based RL.

Focusing specifically on the effects of coverage, Song et al. (2024) show that a global coverage condition is necessary to convert small loss under the reference policy to a performance guarantee for offline DPO, while only local coverage is needed for online RLHF. Moreover, they propose a hybrid preference learning algorithm which combines offline DPO with reverse KL regularization from online samples. Our results also leverage a notion of local coverage, however we provide end-to-end statistical learning guarantees.

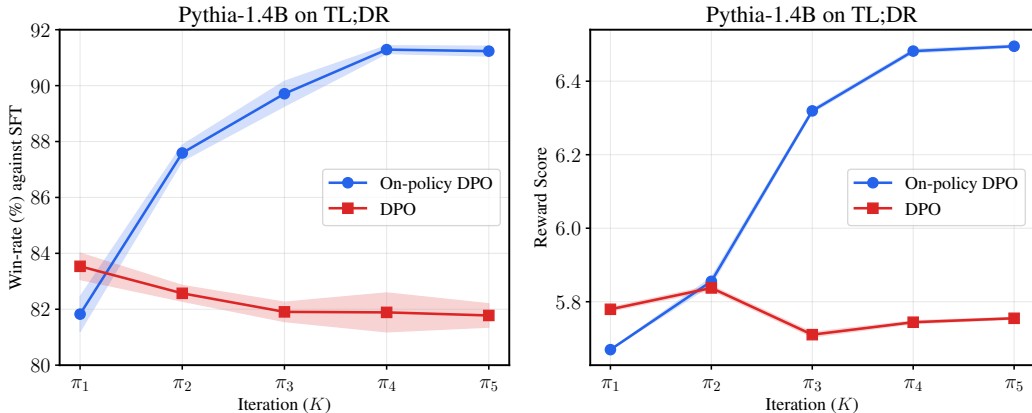

*Figure 1.* Win-rate (left) and reward score (right) of on-policy versus off-policy DPO over 5 iterations. Curves represent the mean performance over 5 runs with standard deviations indicated.

Besides direct alignment from preferences, there is also a wide literature on on-policy or online preference-based reinforcement learning (PbRL), which we mention here without going into detail (Xu et al., 2020; Novoseller et al., 2020; Chen et al., 2022; Pacchiano et al., 2023; Wang et al., 2023; Du et al., 2024; Zhan et al., 2024; Wu & Sun, 2024; Das et al., 2024).

### A.3. Reward distillation methods

Given access to a (typically frozen) reward model and a dataset consisting of unlabeled response pairs, reward model distillation aims to match the pairwise differences of the 'rewards' implicit in the policy (Rafailov et al., 2023) with those of the explicit rewards. The value-based calibration algorithm proposed by Mao et al. (2024) and REBEL algorithm independently proposed by Gao et al. (2024) implement this distillation process with squared loss in an offline and online manner, respectively, and demonstrate similar or stronger performance as PPO and DPO baselines. Fisch et al. (2025) study a pessimistic version of this objective, leading to improved robustness. Moreover, Yun et al. (2025) propose the preference distillation algorithm which acts as a soft version of DPO with synthetic preference labels computed from the relative rewards using the Bradley-Terry model. This can also be seen as reward distillation with the binary KL loss as we show in Section 5.1. Finally, the $A^*$-PO algorithm by Brantley et al. (2025) bypasses the need for pairwise responses by directly estimating the optimal value function to use in the distillation objective.

## B. Experiments

In this section, we empirically corroborate our theoretical results. In Section B.1, we show that on-policy DPO exhibits fast convergence on TL;DR summarization and achieves significant gains over off-policy DPO. In Section B.2, we verify the degradation issue of REBEL on general chat (Gao et al., 2024) and show that our proposed on-policy versions of preference distillation and REBEL succeed in monotonically improving over iterations.

### B.1. TL;DR Summarization

As a warmup, we first evaluate on- and off-policy DPO on the TL;DR summarization task.[5] Our experimental setup largely follows that of Guo et al. (2024). However, as the PaLM-2 family used in their work is not publicly available at present, we adopt Pythia-1.4B[6] (Biderman et al., 2023) as the SFT policy and use the Pythia-6.9B reward model[7] to provide online response feedback.

For on-policy DPO (Algorithm 1), at each round $k$, we generate two candidate responses for each prompt in the TL;DR dataset from $\hat{\pi}_k$ with sampling temperature $\tau = 0.9$ and label them based on the reward scores. We then update the policy

---

[5]https://huggingface.co/datasets/trl-lib/tldr
[6]https://huggingface.co/trl-lib/pythia-1b-deduped-tldr-sft
[7]https://huggingface.co/trl-lib/pythia-6.9b-deduped-tldr-rm

*Table 1.* Results on general chat for off- and on-policy DPO and preference distillation with fixed-regularization (PDFR) across standard benchmarks. The best-performing value for each benchmark is highlighted in **bold**.

| Policy | Method | AlpacaEval 2.0 | | MMLU (5-shot) | GSM8K (5-shot) | Arc (25-shot) | Winogrande (5-shot) | TruthfulQA (0-shot) | HellaSwag (10-shot) |
|---|---|---|---|---|---|---|---|---|---|
| | | LC Win-rate | Win-rate | | | | | | |
| Off-policy | Base | 31.97 | 31.37 | 66.76 | 75.89 | 62.54 | 75.53 | 52.46 | 78.91 |
| | DPO (epoch 3) | 41.26 | 41.18 | **67.14** | 76.50 | **65.61** | 75.77 | **57.16** | 80.18 |
| | PDFR (epoch 3) | 36.45 | 36.97 | 66.95 | **78.01** | 64.51 | 76.56 | 56.33 | 79.76 |
| On-policy | DPO (iter 1) | 39.83 | 37.64 | 66.84 | 76.80 | 63.48 | 75.77 | 53.62 | 79.71 |
| | DPO (iter 2) | 43.16 | 40.99 | 66.92 | 77.41 | 64.16 | 76.40 | 54.87 | 80.21 |
| | DPO (iter 3) | **45.44** | 43.54 | 67.13 | 77.33 | 64.51 | 76.32 | 55.54 | **80.47** |
| | PDFR (iter 1) | 37.89 | 37.08 | 66.91 | 77.56 | 63.57 | 75.85 | 53.76 | 79.54 |
| | PDFR (iter 2) | 41.38 | 40.62 | 67.01 | 77.26 | 64.16 | 76.72 | 55.13 | 80.04 |
| | PDFR (iter 3) | 43.99 | **44.09** | 67.13 | 77.03 | 64.51 | **76.72** | 56.08 | 80.45 |

*Table 2.* Results on general chat for on-policy REBEL and reward distillation with reward-calibration (RDRC). The best-performing value for each method is highlighted in **bold**.

| Policy | Method | AlpacaEval 2.0 | | MMLU (5-shot) | GSM8K (5-shot) | Arc (25-shot) | Winogrande (5-shot) | TruthfulQA (0-shot) | HellaSwag (10-shot) |
|---|---|---|---|---|---|---|---|---|---|
| | | LC Win-rate | Win-rate | | | | | | |
| Off-policy | Base | 31.97 | 31.37 | 66.76 | 75.89 | 62.54 | 75.53 | 52.46 | 78.91 |
| | REBEL (epoch 5) | 33.85 | 33.13 | 66.47 | 77.33 | 63.40 | 76.01 | 53.71 | 79.20 |
| On-policy | REBEL (iter 1) | 40.84 | 39.32 | 66.96 | 77.10 | 63.65 | 76.01 | 53.55 | 79.45 |
| | REBEL (iter 2) | 43.02 | 41.61 | 67.06 | 76.80 | 63.82 | 75.93 | 55.03 | 79.89 |
| | REBEL (iter 3) | **43.81** | **42.42** | 67.07 | **77.33** | 64.33 | 76.32 | 55.02 | 79.87 |
| | REBEL (iter 4) | 43.63 | 42.30 | 67.07 | 77.33 | 64.25 | **76.48** | **55.44** | **79.90** |
| | REBEL (iter 5) | 42.90 | 41.74 | **67.14** | 76.88 | **64.51** | 76.01 | 55.42 | 79.74 |
| | RDRC (iter 1) | 39.02 | 37.39 | 67.12 | 76.80 | 63.65 | 76.09 | 53.53 | 79.49 |
| | RDRC (iter 2) | 42.16 | 40.93 | 67.09 | 76.95 | 63.91 | **76.40** | 54.87 | 79.70 |
| | RDRC (iter 3) | 42.67 | 41.30 | 67.12 | 76.12 | 64.16 | 76.32 | 54.93 | 79.73 |
| | RDRC (iter 4) | 43.90 | 42.30 | 67.18 | 77.48 | 64.08 | 76.24 | 55.27 | 79.66 |
| | RDRC (iter 5) | **44.07** | **42.61** | **67.18** | **77.63** | **64.33** | 75.77 | **55.54** | **79.84** |

for 5 total iterations, each using a peak learning rate $\eta = 5 \times 10^{-7}$ with a 10% warmup phase followed by cosine scheduling. As a controlled baseline, we compare with off-policy DPO trained for 5 epochs on the dataset generated in the first round of on-policy DPO. We set $\gamma = 0.1$ and use a fixed batch size of 128 for all runs. Models are evaluated using win-rate against the SFT baseline and reward score; win-rate is computed using an LLM-based judge with preferences determined by `gpt-5-mini`.

The results, averaged over 5 runs, are plotted in Figure 1. On-policy DPO demonstrates significant gains in both win-rate and rewards and exhibits the fast convergence predicted by our analysis in Section 3. In contrast, vanilla DPO suffers early saturation and even degradation after the first epoch, confirming the results of Guo et al. (2024); Xu et al. (2024a) under a more controlled setting.

## B.2. General Chat

We conduct comprehensive experiments on the general chat task in Gao et al. (2024). We compare on-policy DPO (Guo et al., 2024), REBEL (Gao et al., 2024), and two of our proposed distillation schemes in Algorithm 3: reward distillation with reward-calibration (RDRC) modifying REBEL, and preference distillation (Yun et al., 2025) made on-policy with fixed-regularization (PDFR). We use LLaMA-3-8B-Instruct[8] (AI@Meta, 2024) as the base policy and train on the UltraFeedback dataset[9] (Cui et al., 2023) with ArmoRM-Llama3-8B-v0.1[10] (Wang et al., 2024) serving as the reward model. We follow the

---

[8] https://huggingface.co/meta-llama/Meta-Llama-3-8B-Instruct
[9] https://huggingface.co/datasets/allenai/ultrafeedback_binarized_cleaned_train
[10] https://huggingface.co/RLHFlow/ArmoRM-Llama3-8B-v0.1

training setup described in Appendix H.2 of Gao et al. (2024).[11] For each prompt, we generate two candidate responses with a maximum response length of 2048 tokens. In all runs, we use a batch size of 128 and peak learning rate $\eta = 3 \times 10^{-7}$ with a 10% warmup ratio followed by cosine scheduling. Below, we provide implementation details specific to each method.

**DPO/PDFR.** We fix $\gamma = 0.05$ and train for 3 iterations (on-policy) or 3 epochs (off-policy). Since the reward scale is not controlled, we implement PDFR with a tunable scaling parameter $\beta$ when computing the distillation 'soft labels': $\tilde{\mathbb{P}}(a^1 \succ a^2 \mid x) = \sigma(\beta(\tilde{r}(x, a^1) - \tilde{r}(x, a^2)))$. This allows us to control the sharpness of the induced preference distribution. In our experiments, we use $\beta = 100$.

**REBEL/RDRC.** We train for 5 iterations/epochs to test our prediction that REBEL-style distillation can become increasingly degenerate over training (Section 5). We follow the recommended 3-iteration schedule for REBEL, $\gamma \in \{10^{-6}, 10^{-4}, 10^{-2}\}$, then fix $\gamma = 10^{-2}$ for the remaining two rounds. For RDRC, we use a relatively small fixed calibration level $\gamma_c = 3 \times 10^{-4}$ for stability.

**Evaluation.** We evaluate response quality with AlpacaEval 2.0 (Dubois et al., 2024), which uses `GPT-4 turbo` as the reference model. We also measure alignment tax, i.e., the extent to which alignment degrades general capabilities, by evaluating on OpenLLM leaderboard benchmarks (Beeching et al., 2023), including MMLU (Hendrycks et al., 2021), GSM8K (Cobbe et al., 2021), ARC Challenge (Clark et al., 2018), Winogrande (Sakaguchi et al., 2021), TruthfulQA (Lin et al., 2022), and HellaSwag (Zellers et al., 2019).

**Results.** The evaluation results for DPO/PDFR and REBEL/RDRC and shown in Tables 1,2, respectively. Similar to on-policy DPO, our proposed distillation schemes PDFR and RDRC yield clear monotonic improvements over iterations and beat their off-policy counterparts by a wide margin, while incurring similar alignment tax as validated on the OpenLLM leaderboard. This is consistent with our theoretical results in Sections 3 and 5. Moreover, REBEL indeed degrades in performance after the third iteration as predicted in Section 5.1. In contrast, reward calibration enables RDRC to continually improve beyond this threshold and achieve best performance at the final iteration.

## C. Analysis of On-policy DPO

### C.1. Proof of Proposition 3.1

We will prove the following extended proposition for completeness. The tighter bound (2) comes from constraining the update to $\Theta_k := B_p(\hat{\theta}_k, r_k)$ assuming knowledge of $r_k$. This allows us to slightly improve the fixed-point analysis of Theorem 3.2, however we omit this for brevity. In addition, we can also allow choosing any user-defined radius $\bar{R} \geq R$ for the policy class if $R$ is unknown; the dependence on $\bar{R}$ (as opposed to $R$) will be only logarithmic.

**Proposition C.1** (Full version of Proposition 3.1). *Assume $\hat{\pi}_k \in \Pi$ satisfies $\hat{\theta}_k \in B_p(\theta^*, r_k)$ with radius $r_k \in [0, \bar{R}]$. Suppose $D_k$ consists of $n_k$ i.i.d. samples of prompts $x \sim \mathcal{D}$ and pairs of responses $a^+, a^-$ sampled from $\hat{\pi}_k(x)$ and labeled using $\mathbb{P}_*$. Let $\Theta_k := B_p(\hat{\theta}_k, r_k)$ and $\delta \in (0, 1)$. Then it holds with probability at least $1 - \delta$ that:*

(1) *The update $\hat{\theta}_{k+1} = \arg\min_{\theta \in B_p(0, \bar{R})} \mathcal{L}_{\text{DPO}}(\theta; D_k)$ satisfies $\hat{\theta}_{k+1} \in B_p(\theta^*, r_{k+1})$ where*

$$r_{k+1}^2 := \frac{52 e^{4\gamma R} d^{2/p}}{\lambda \gamma^2 n_k} C_*^p(r_k) \ln \frac{9\gamma \bar{R} n_k}{\delta}, \quad \text{as long as} \quad r_{k+1} \leq \frac{1}{2\gamma} \wedge \bar{R}.$$

(2) *The update $\hat{\theta}_{k+1} = \arg\min_{\theta \in \Theta_k} \mathcal{L}_{\text{DPO}}(\theta; D_k)$ satisfies $\hat{\theta}_{k+1} \in B_p(\theta^*, r_{k+1})$ where*

$$r_{k+1}^2 := \frac{52 e^{4\gamma R} d^{2/p}}{\lambda \gamma^2 n_k} C_*^p(r_k) \ln \frac{9\gamma r_k n_k}{\delta}, \quad \text{as long as} \quad r_{k+1} \leq \frac{1}{2\gamma} \wedge 2r_k.$$

The proof proceeds in several steps. We will give the proof for Proposition C.1(2) and provide the necessary modifications to obtain Proposition C.1(1) at the end.

---

[11] https://github.com/ZhaolinGao/REBEL

**Step 1: Bounding learned preferences.** For each preference pair $(a^+, a^-)$, denote the corresponding pair of unlabeled responses as $(a^1, a^2)$ and the indicator $y = 1\{a^+ = a^1\}$ so that $\mathbb{P}(y = 1) = \mathbb{P}_*(a^1 \succ a^2 \mid x)$ by Assumption 1. We adopt the following shorthand:

$$\mathbb{P}_{\pi,\pi'}(a^1 \succ a^2 \mid x) := \sigma\left(\gamma \ln \frac{\pi(a^1 \mid x)}{\pi(a^2 \mid x)} - \gamma \ln \frac{\pi'(a^1 \mid x)}{\pi'(a^2 \mid x)}\right), \tag{6}$$

so that $\mathbb{P}_* = \mathbb{P}_{\pi^*,\pi_0}$. Then the objective at the $k$th step can be rewritten as

$$\mathcal{L}_{\mathrm{DPO}}(\pi; D_k) = -\sum_{(x,a^1,a^2) \in D_k} y \ln \mathbb{P}_{\pi,\pi_0}(a^1 \succ a^2 \mid x) + (1 - y) \ln \mathbb{P}_{\pi,\pi_0}(a^2 \succ a^1 \mid x).$$

Construct an $\varepsilon$-cover $\mathcal{M}$ of the $L^p$-ball $B_p(\hat{\theta}_k, r_k)$ so that $|\mathcal{M}| \leq (3r_k/\varepsilon)^d$. Define the policy estimator $\check{\theta}_{k+1}$ as the closest element to $\hat{\theta}_{k+1}$ in $\mathcal{M}$, with ties broken arbitrarily, and set $\check{\pi}_{k+1} := \pi_{\check{\theta}_{k+1}}$. Let $\tilde{D}_k$ be an independent copy of $D_k$. From the symmetrization inequality (Lemma C.4), it holds with probability $1 - \delta$,

$$-\ln \mathbb{E}_{\tilde{D}_k}\left[\exp\left(\frac{1}{2}(\mathcal{L}_{\mathrm{DPO}}(\pi^*; \tilde{D}_k) - \mathcal{L}_{\mathrm{DPO}}(\check{\pi}_{k+1}; \tilde{D}_k))\right)\right]$$

$$\leq -\frac{1}{2}(\mathcal{L}_{\mathrm{DPO}}(\pi^*; D_k) - \mathcal{L}_{\mathrm{DPO}}(\check{\pi}_{k+1}; D_k)) + \ln \frac{|\mathcal{M}|}{\delta}.$$

The left-hand side further satisfies

$$-\ln \mathbb{E}_{\tilde{D}_k}\left[\exp\left(\frac{1}{2}(\mathcal{L}_{\mathrm{DPO}}(\pi^*; \tilde{D}_k) - \mathcal{L}_{\mathrm{DPO}}(\check{\pi}_{k+1}; \tilde{D}_k))\right)\right]$$

$$= -\ln \mathbb{E}_{\tilde{D}_k}\left[\exp\left(\frac{1}{2}\sum_{(x,a^1,a^2) \in \tilde{D}_k} y \ln \frac{\mathbb{P}_{\check{\pi}_{k+1},\pi_0}(a^1 \succ a^2 \mid x)}{\mathbb{P}_*(a^1 \succ a^2 \mid x)} + (1 - y) \ln \frac{\mathbb{P}_{\check{\pi}_{k+1},\pi_0}(a^2 \succ a^1 \mid x)}{\mathbb{P}_*(a^2 \succ a^1 \mid x)}\right)\right]$$

$$= -n_k \ln \mathbb{E}_{x \sim \mathcal{D}, a^1, a^2 \sim \hat{\pi}_k(x)}$$

$$\left[\mathbb{P}_*(a^1 \succ a^2 \mid x)\left(\frac{\mathbb{P}_{\check{\pi}_{k+1},\pi_0}(a^1 \succ a^2 \mid x)}{\mathbb{P}_*(a^1 \succ a^2 \mid x)}\right)^{1/2} + \mathbb{P}_*(a^2 \succ a^1 \mid x)\left(\frac{\mathbb{P}_{\check{\pi}_{k+1},\pi_0}(a^2 \succ a^1 \mid x)}{\mathbb{P}_*(a^2 \succ a^1 \mid x)}\right)^{1/2}\right]$$

$$= -n_k \ln \mathbb{E}_{x \sim \mathcal{D}, a^1, a^2 \sim \hat{\pi}_k(x)}$$

$$\left[\sqrt{\mathbb{P}_{\check{\pi}_{k+1},\pi_0}(a^1 \succ a^2 \mid x)\mathbb{P}_*(a^1 \succ a^2 \mid x)} + \sqrt{\mathbb{P}_{\check{\pi}_{k+1},\pi_0}(a^2 \succ a^1 \mid x)\mathbb{P}_*(a^2 \succ a^1 \mid x)}\right]$$

$$\geq \frac{n_k}{4}\mathbb{E}_{x \sim \mathcal{D}, a^1, a^2 \sim \hat{\pi}_k(x)}\left[\left(\mathbb{P}_{\check{\pi}_{k+1},\pi_0}(a^1 \succ a^2 \mid x) - \mathbb{P}_*(a^1 \succ a^2 \mid x)\right)^2\right],$$

by Lemma C.5.

For the right-hand side, since $\|\hat{\theta}_{k+1} - \check{\theta}_{k+1}\|_2 \leq \|\hat{\theta}_{k+1} - \check{\theta}_{k+1}\|_p \leq \varepsilon$, by Lemma C.7,

$$\mathcal{L}_{\mathrm{DPO}}(\check{\pi}_{k+1}; D_k) \leq \mathcal{L}_{\mathrm{DPO}}(\hat{\pi}_{k+1}; D_k) + 2\gamma\varepsilon n_k \leq \mathcal{L}_{\mathrm{DPO}}(\pi^*; D_k) + 2\gamma\varepsilon n_k$$

and so

$$-\frac{1}{2}(\mathcal{L}_{\mathrm{DPO}}(\pi^*; D_k) - \mathcal{L}_{\mathrm{DPO}}(\check{\pi}_{k+1}; D_k)) + \ln \frac{|\mathcal{M}|}{\delta} \leq \gamma\varepsilon n_k + d \ln \frac{3r_k}{\varepsilon} + \ln \frac{1}{\delta}.$$

It follows that

$$\mathbb{E}_{x \sim \mathcal{D}, a^1, a^2 \sim \hat{\pi}_k(x)}\left[\left(\mathbb{P}_{\hat{\pi}_{k+1},\pi_0}(a^1 \succ a^2 \mid x) - \mathbb{P}_*(a^1 \succ a^2 \mid x)\right)^2\right]$$

$$\leq \mathbb{E}_{x \sim \mathcal{D}, a^1, a^2 \sim \hat{\pi}_k(x)}\left[\left(\mathbb{P}_{\hat{\pi}_{k+1},\pi_0}(a^1 \succ a^2 \mid x) - \mathbb{P}_{\check{\pi}_{k+1},\pi_0}(a^1 \succ a^2 \mid x)\right)^2\right]$$

$$+ \mathbb{E}_{x \sim \mathcal{D}, a^1, a^2 \sim \hat{\pi}_k(x)}\left[\left(\mathbb{P}_{\check{\pi}_{k+1},\pi_0}(a^1 \succ a^2 \mid x) - \mathbb{P}_*(a^1 \succ a^2 \mid x)\right)^2\right]$$

$$\leq \frac{\gamma^2 \varepsilon^2}{4} + \frac{4}{n_k} \left( \gamma \varepsilon n_k + d \ln \frac{3r_k}{\varepsilon} + \ln \frac{1}{\delta} \right),$$

again applying Lemma C.7. Choosing $\varepsilon = 1/\gamma n_k$ and noting $1 + \frac{1}{16n_k} \leq \frac{17}{16} < \ln 3$, we conclude:

$$\mathbb{E}_{x \sim \mathcal{D}, a^1, a^2 \sim \hat{\pi}_k(x)} \left[ \left( \mathbb{P}_{\hat{\pi}_{k+1}, \pi_0}(a^1 \succ a^2 \mid x) - \mathbb{P}_*(a^1 \succ a^2 \mid x) \right)^2 \right] \leq \frac{4}{n_k} \left( d \ln(3\gamma r_k n_k) + \ln \frac{3}{\delta} \right). \tag{7}$$

**Step 2: Initial improvement of radius.** Recall that

$$\mathbb{P}_{\hat{\pi}_{k+1}, \pi_0}(a^1 \succ a^2 \mid x) = \sigma \left( \gamma \ln \frac{\hat{\pi}_{k+1}(a^1 \mid x)}{\hat{\pi}_{k+1}(a^2 \mid x)} - \gamma \ln \frac{\pi_0(a^1 \mid x)}{\pi_0(a^2 \mid x)} \right)$$

where

$$\left| \gamma \ln \frac{\hat{\pi}_{k+1}(a^1 \mid x)}{\hat{\pi}_{k+1}(a^2 \mid x)} - \gamma \ln \frac{\pi_0(a^1 \mid x)}{\pi_0(a^2 \mid x)} \right| = \gamma \left| \hat{\theta}_{k+1}^\top (\phi(x, a^1) - \phi(x, a^2)) \right| \leq 2\gamma \|\hat{\theta}_{k+1}\|_2.$$

Since $\hat{\theta}_k \in B_p(\theta^*, r_k)$ and $\hat{\theta}_{k+1} \in B_p(\hat{\theta}_k, r_k)$, we have

$$\|\hat{\theta}_{k+1}\|_2 \leq \|\hat{\theta}_{k+1}\|_p \leq \|\hat{\theta}_k\|_p + r_k \leq \|\theta^*\|_p + 2r_k \leq R + 2r_k. \tag{8}$$

Moreover by Assumption 1, $\mathbb{P}_*(a^1 \succ a^2 \mid x) = \sigma(r^*(x, a^1) - r^*(x, a^2))$ where

$$\left| r^*(x, a^1) - r^*(x, a^2) \right| = \gamma |\theta^{*\top}(\phi(x, a^1) - \phi(x, a^2))| \leq 2\gamma R$$

since $\|\theta^*\|_2 \leq \|\theta^*\|_p \leq R$. Hence by Lemma C.3,

$$\mathbb{E}_{x \sim \mathcal{D}, a^1, a^2 \sim \hat{\pi}_k(x)} \left[ \left( \mathbb{P}_{\hat{\pi}_{k+1}, \pi_0}(a^1 \succ a^2 \mid x) - \mathbb{P}_*(a^1 \succ a^2 \mid x) \right)^2 \right]$$

$$\geq \left( \frac{1}{5.09(4\gamma r_k \vee 1)e^{2\gamma R}} \right)^2$$

$$\times \mathbb{E}_{x \sim \mathcal{D}, a^1, a^2 \sim \hat{\pi}_k(x)} \left[ \left( \gamma \ln \frac{\hat{\pi}_{k+1}(a^1 \mid x)}{\hat{\pi}_{k+1}(a^2 \mid x)} - \gamma \ln \frac{\pi_0(a^1 \mid x)}{\pi_0(a^2 \mid x)} - r^*(x, a^1) + r^*(x, a^2) \right)^2 \right].$$

Now, owing to the fact that $\mathbb{E}[(X - X')^2] = 2 \operatorname{Var}(X)$ for i.i.d. $X, X'$,

$$\mathbb{E}_{x \sim \mathcal{D}, a^1, a^2 \sim \hat{\pi}_k(x)} \left[ \left( \gamma \ln \frac{\hat{\pi}_{k+1}(a^1 \mid x)}{\hat{\pi}_{k+1}(a^2 \mid x)} - \gamma \ln \frac{\pi_0(a^1 \mid x)}{\pi_0(a^2 \mid x)} - r^*(x, a^1) + r^*(x, a^2) \right)^2 \right]$$

$$= 2\mathbb{E}_{x \sim \mathcal{D}} \left[ \operatorname{Var}_{a \sim \hat{\pi}_k(x)} \left( \gamma \ln \frac{\hat{\pi}_{k+1}(a \mid x)}{\pi_0(a \mid x)} - r^*(x, a) \right) \right]$$

$$= 2\gamma^2 \mathbb{E}_{x \sim \mathcal{D}} \left[ \operatorname{Var}_{a \sim \hat{\pi}_k(x)} \left( (\hat{\theta}_{k+1} - \theta^*)^\top \phi(x, a) \right) \right]$$

$$= 2\gamma^2 (\hat{\theta}_{k+1} - \theta^*)^\top \mathbb{V}(\hat{\pi}_k)(\hat{\theta}_{k+1} - \theta^*).$$

Moreover,

$$(\hat{\theta}_{k+1} - \theta^*)^\top \mathbb{V}(\hat{\pi}_k)(\hat{\theta}_{k+1} - \theta^*) \geq \frac{1}{C_{\hat{\pi}_k \to \pi^*}}(\hat{\theta}_{k+1} - \theta^*)^\top \mathbb{V}(\pi^*)(\hat{\theta}_{k+1} - \theta^*) \geq \frac{\lambda}{C_{\hat{\pi}_k \to \pi^*}} \|\hat{\theta}_{k+1} - \theta^*\|_2^2$$

by Assumption 3. Since $\|\hat{\theta}_k - \theta^*\|_p \leq r_k$, we also have $C_{\hat{\pi}_k \to \pi^*} \leq C_*^p(r_k)$ by definition. Putting everything together, we have in conjunction with Eq. (7):

$$\|\hat{\theta}_{k+1} - \theta^*\|_p^2$$

$$\leq \frac{d^{2/p-1}}{\lambda} C_{\hat{\pi}_k \to \pi^*} (\hat{\theta}_{k+1} - \theta^*)^\top \mathbb{V}(\hat{\pi}_k)(\hat{\theta}_{k+1} - \theta^*)$$

$$\leq \frac{d^{2/p-1}}{2\lambda\gamma^2} C_{\hat{\pi}_k \to \pi^*} 5.09^2 (4\gamma r_k \vee 1)^2 e^{4\gamma R} \mathbb{E}_{x \sim \mathcal{D}, a^1, a^2 \sim \hat{\pi}_k(x)} \left[ \left( \mathbb{P}_{\hat{\pi}_{k+1}, \pi_0}(a^1 \succ a^2 \mid x) - \mathbb{P}_*(a^1 \succ a^2 \mid x) \right)^2 \right]$$

$$\leq \underbrace{\frac{52 e^{4\gamma R} d^{2/p}}{\lambda \gamma^2 n_k} C_*^p(r_k) \ln \frac{9\gamma r_k n_k}{\delta}}_{=: \Phi_k} (16\gamma^2 r_k^2 \vee 1).$$

Denote this initial upper bound as $\tilde{r}_{k+1,1}^2$. This is an improvement over the trivial bound

$$\|\hat{\theta}_{k+1} - \theta^*\|_p \leq \|\hat{\theta}_{k+1} - \hat{\theta}_k\|_p + \|\hat{\theta}_k - \theta^*\|_p \leq 2r_k,$$

if and only if

$$\tilde{r}_{k+1,1}^2 = \Phi_k (16\gamma^2 r_k^2 \vee 1) \leq 4r_k^2 \quad \Leftrightarrow \quad \Phi_k \leq \frac{1}{4\gamma^2} \wedge 4r_k^2. \tag{9}$$

**Step 3: Iterative tightening.** The following trick allows us to shave off a factor of at most $O(\bar{R})$ from the initial bound $\tilde{r}_{k+1,1}$ when $n_k$ is sufficiently large. With the newfound knowledge that $\|\hat{\theta}_{k+1} - \theta^*\|_p \leq \tilde{r}_{k+1,1}$, we can replace Eq. (8) with $\|\hat{\theta}_{k+1}\|_p \leq R + \tilde{r}_{k+1,1}$ and repeat the following steps to show that

$$\|\hat{\theta}_{k+1} - \theta^*\|_p^2 \leq \Phi_k (4\gamma^2 \tilde{r}_{k+1,1}^2 \vee 1) =: \tilde{r}_{k+1,2}^2.$$

Assuming $\tilde{r}_{k+1,1} \leq 2r_k$ as in Eq. (9), this gives an improved bound $\tilde{r}_{k+1,2} \leq \tilde{r}_{k+1,1}$. Repeating this argument yields a sequence of increasingly tighter radii $(\tilde{r}_{k+1,i})_{i=1}^\infty$ defined recursively as

$$\tilde{r}_{k+1,i+1}^2 = \Phi_k (4\gamma^2 \tilde{r}_{k+1,i}^2 \vee 1). \tag{10}$$

By the monotone convergence theorem, we may define the limiting radius $r_{k+1} := \lim_{i \to \infty} \tilde{r}_{k+1,i}$ so that $\|\hat{\theta}_{k+1} - \theta^*\|_2 \leq r_{k+1}$. Taking $i \to \infty$ on both sides of Eq. (10) yields

$$r_{k+1}^2 = \Phi_k (4\gamma^2 r_{k+1}^2 \vee 1) \quad \Rightarrow \quad r_{k+1}^2 = \Phi_k$$

since $r_{k+1}^2 = \Phi_k \cdot 4\gamma^2 r_{k+1}^2$ cannot be true under Eq. (9). Hence we have proven Proposition C.1(2).

Finally, for the radius-agnostic update described in Proposition C.1(1), we must replace the policy class radius $r_k$ in the covering number analysis of Step 1 with the worst-case radius $\bar{R}$, as well as Eq. (8) in Step 2 with the trivial bound $\|\hat{\theta}_{k+1}\|_2 \leq \bar{R}$. This yields the initial bound

$$\|\hat{\theta}_{k+1} - \theta^*\|_2^2 \leq \underbrace{\frac{52 e^{4\gamma R} d^{2/p}}{\lambda \gamma^2 n_k} C_*^p(r_k) \ln \frac{9\gamma \bar{R} n_k}{\delta}}_{=: \Phi_k} (4\gamma^2 \bar{R}^2 \vee 1) =: \tilde{r}_{k+1,1}^2.$$

Then as long as

$$\Phi_k (4\gamma^2 \bar{R}^2 \vee 1) \leq \bar{R}^2 \quad \Leftrightarrow \quad \Phi_k \leq \frac{1}{4\gamma^2} \wedge \bar{R}^2,$$

we can again construct a tightening sequence as $\tilde{r}_{k+1,i+1}^2 = \Phi_k (4\gamma^2 \tilde{r}_{k+1,i}^2 \vee 1)$ and repeat the limiting argument to get rid of the $4\gamma^2 \bar{R}^2$ factor and obtain the desired guarantee. $\square$

**Remark C.2.** The use of the symmetrization inequality and Lemma C.5 to bound the squared differences of the preference distributions borrow from Theorem 6 of Foster & Krishnamurthy (2021) and Theorem 4 of Yun et al. (2025). Moreover, using the tighter sigmoid difference bound Lemma C.3 rather than Lemma 8 of Yun et al. (2025) allows us to avoid exponential dependency on $\bar{R}$, which would otherwise result in an $e^{2\gamma \bar{R}}$ factor; after the tightening argument, the final dependence on $\bar{R}$ is only logarithmic (arising from the log-entropy of the policy class).

## C.2. Proof of Theorem 3.2

Define the quantity

$$\xi := \frac{\eta^2}{\gamma^2}\xi_n = \frac{52 e^{4\gamma R} d^{2/p}}{\lambda\gamma^2 n} \ln \frac{9\gamma RKn}{\delta}.$$

From the assumed condition $\xi_n \lesssim 1/C_*^p(R)$, we can guarantee

$$\xi \le \frac{\eta^2 R^2}{C_*^p(R)} \wedge \frac{1}{4\gamma^2 C_*^p(R)} \wedge \frac{\eta^2}{e^2} \tag{11}$$

where the second comparison follows from $\sqrt{C_*^p(R)} \wedge R \gtrsim 1/\gamma$.

By Proposition 3.1 and Assumption 2, union bounding over all $K$ iterations, we have with probability at least $1 - \delta$ that $\|\hat\theta_k - \theta^*\|_p \le r_k$ for all $k \le K$ for the sequence of radii $(r_k)_{k\ge 0}$ defined recursively as

$$r_{k+1}^2 = \xi C_*^p(r_k), \quad r_0 = R, \tag{12}$$

as long as $r_k \le (1/2\gamma) \wedge R$ for each $k \ge 1$ (we verify this below).

We now analyze the convergence of this iteration. For $z > 0$, define the set

$$S_z := \{r > 0 \mid \xi C_*^p(r) \le z^2 r^2\}.$$

If $C_*^p$ is square root-convex, the mapping $r \mapsto \sqrt{\xi C_*^p(r)} - zr$ is convex, so its level set $S_z$ is a closed convex subset of $\mathbb{R}_{>0}$. More generally, under Assumption 4, we may choose $\eta' \le \eta$ such that

$$\frac{(\eta')^2}{\xi} = \kappa \frac{C_*^p(R)}{R^2}$$

satisfying Eq. (11), so $S_{\eta'}$ is a closed interval, and proceed with the remainder of the proof with $\eta$ replaced by $\eta'$. Now under the first condition of Eq. (11), it holds that $R \in S_\eta$, so $S_\eta, S_1$ are nonempty and we may write

$$S_1 = [\alpha_1, \beta_1], \quad S_\eta = [\alpha_\eta, \beta_\eta], \quad \text{where} \quad \alpha_1 \le \alpha_\eta \le R \le \beta_\eta \le \beta_1.$$

Note that $\eta^2 \alpha_\eta^2 \ge \xi C_*^p(\alpha_\eta) \ge \xi C_*^p(0) = \xi$ so that $\alpha_\eta \ge \sqrt{\xi}/\eta$. This bound is nearly tight: since

$$\xi C_*^p\left(\frac{e\sqrt\xi}{\eta}\right) \le \xi \exp\left(\frac{2e\sqrt\xi}{\eta}\right) \le e^2\xi = \eta^2\left(\frac{e\sqrt\xi}{\eta}\right)^2$$

due to Lemma C.8 and the third condition of Eq. (11), it follows that $\alpha_\eta \le e\sqrt\xi/\eta$. Moreover, the set $S_1$ is invariant under the mapping $r \mapsto \sqrt{\xi C_*^p(r)}$, so the sequence $(r_k)_{k\ge 0}$ is monotone decreasing and $r_1 = \sqrt{\xi C_*^p(R)} \le (1/2\gamma) \wedge R$ under the second condition of Eq. (11).

We now claim that

$$r_K \le \alpha_\eta \vee \eta^K r_0 \le \frac{e\sqrt\xi}{\eta} \vee R\eta^K.$$

Indeed, if $r_K > \alpha_\eta$, then $r_k \in S_\eta$ for all $k \le K$ and

$$r_{k+1}^2 = \xi C_*^p(r_k) \le \eta^2 r_k^2 \quad \Rightarrow \quad r_K \le \eta r_{K-1} \le \cdots \le \eta^K r_0.$$

Therefore,

$$\|\hat\theta_K - \theta_*\|_2^2 \le \|\hat\theta_K - \theta_*\|_p^2 \le \frac{e^2\xi}{\eta^2} \vee R^2\eta^{2K} = \frac{e^2\xi_n}{\gamma^2} \vee R^2\eta^{2K}$$

and $\|\hat\theta_K - \theta_*\|_2^2$ upper bounds both $\mathrm{KL}(\hat\pi_K\|\pi^*), \mathrm{KL}(\pi^*\|\hat\pi_K)$ by Proposition C.9. We also have

$$\chi^2(\hat\pi_K, \pi^*) \lesssim \ln\left(1 + \chi^2(\hat\pi_K, \pi^*)\right) \le \frac{e^2\xi_n}{\gamma^2} \vee R^2\eta^{2K}$$

due to the inequality $z \lesssim \ln(1 + z)$, valid for positive $z = O(1)$. $\qquad\square$

**Proof of Corollary 3.3.** The lower bound for $n_{\text{off}}$ follows immediately from Theorem 2.2. For the upper bound, we require $n \geq \widetilde{O}(e^{4\gamma R}C_*^p(R))$ to satisfy Eq. (4), then Theorem 3.2 implies

$$\|\hat{\theta}_K - \theta^*\|_p \leq \widetilde{O}\left(\frac{e^{2\gamma R}}{\gamma\sqrt{n}}\right) \vee R\eta^K.$$

Setting each term to $\varepsilon$ gives $n = e^{4\gamma R} \cdot \widetilde{O}\left(\frac{R^2}{\varepsilon^2} \vee C_*^p(R)\right)$ and $K = O\left(\ln\frac{R}{\varepsilon}\right)$. When $\gamma R = \Theta(1)$, this yields Corollary 3.3. In the general case, we still have $n_{\text{on}} \ll n_{\text{off}}$ if

$$e^{4\gamma R}\frac{R^2}{\varepsilon^2} \ll \frac{1}{\varepsilon^2}e^{2^{1-1/p}R}, \quad e^{4\gamma R}C_*^p(R) \ll \frac{1}{\varepsilon^2}C_*^p(R) \quad \Leftrightarrow \quad \gamma < 2^{-1-1/p}, \quad \varepsilon \ll e^{-2\gamma R},$$

verifying Remark 3.5. $\qquad \square$

**Proof of Corollary 3.6.** Let $K$ be the smallest integer such that $\eta^K R < r_0$. We inductively show that $\|\hat{\theta}_k - \theta^*\|_p \leq \eta^k R$ when $0 \leq k < K$. By Proposition 3.1 with $\eta^k R$ in place of $r_k$, it holds with probability $1 - \delta/K$ that

$$\|\hat{\theta}_{k+1} - \theta^*\|_p \leq r_{k+1}, \quad r_{k+1}^2 \asymp \frac{d^{2/p}}{\lambda\gamma^2 n_k}C_*^p(\eta^k R)\ln\frac{\gamma RKn_k}{\delta}$$

as long as $r_{k+1} \leq \frac{1}{2\gamma} \wedge 2\eta^k R$ is satisfied. In particular, $n_0 = n_i$ suffices when $k = 0$ from the proof of Theorem 3.2. For $k \geq 1$, we choose $n_k$ so that $r_{k+1} \leq r_k \wedge \eta^{k+1}R$; by Lemma C.6, this is satisfied if $n_k \asymp m_k\ln(m_k/\delta)$ where

$$m_k \asymp \frac{d^{2/p}}{\lambda\gamma^2\eta^{2k+2}R^2}C_*^p(\eta^k R).$$

Since

$$\frac{m_k}{m_{k-1}} = \frac{1}{\eta^2} \cdot \frac{C_*^p(\eta^k R)}{C_*^p(\eta^{k-1}R)} \leq \frac{1}{\eta^2} \cdot \frac{(\eta^k R)^{2+\alpha}}{(\eta^{k-1}R)^{2+\alpha}} = \eta^\alpha < 1$$

by the super-quadratic assumption, we have that $n_k/n_{k-1} \leq m_k/m_{k-1} \leq \eta^\alpha$ also converges exponentially, so the sum $n_0 + \cdots + n_{K-1}$ is dominated by $n_0 = n_i$. Then at step $K$, it holds $r_K < r_0$ so that $C_*^p(r_K) \leq e^{2r_0}$ is bounded by a constant, and we can ensure by setting $n_K = n_f$ that

$$\|\hat{\theta}_{K+1} - \theta^*\|_p^2 \lesssim \frac{e^{2r_0}d^{2/p}}{\lambda\gamma^2 n_f}\ln\frac{n_f}{\delta} \leq \varepsilon^2.$$

We remark that if $C_*^p(r) \sim e^{\Theta(r)}$ exhibits exponential growth as in Proposition 2.1; Lemma C.8, $r_k \leq \eta^k R$ implies that the required $n_k \asymp e^{\Theta(r_k)}/r_k^2$ can even decay doubly exponentially. $\qquad \square$

### C.3. Auxiliary Results

**Lemma C.3** (Tight sigmoid differences). *Let $a, b > 0$ be fixed radii. For all $|z| \leq a$, $|w| \leq b$ it holds*

$$|\sigma(z) - \sigma(w)| \geq \frac{|z - w|}{5.09(|b - a| \vee 1)e^{a\wedge b}}.$$

This improves upon the $e^{-(a\vee b)}$ constant in Lemma 8 of Yun et al. (2025) by relaxing the exponential dependency on the larger radius $b$ to linear. A similar result has been shown in Lemma 9 of Faury et al. (2020), however our bound can be slightly tighter and will prove easier to manipulate.

*Proof.* By symmetry we may assume $a \leq b$ and $w > 0$. Fix an auxiliary constant $\tau > 0$. First suppose $w < a + \tau$ and note that $\sigma'$ is even and decreasing on $\mathbb{R}_{\geq 0}$. Then by the mean value theorem, there exists $t \in (-a, a + \tau)$ such that

$$\frac{\sigma(z) - \sigma(w)}{z - w} = \sigma'(t) \geq \sigma'(a + \tau) = \frac{1}{1 + e^{a+\tau}}\frac{1}{1 + e^{-(a+\tau)}} \geq \frac{1}{(1 + e^\tau)(1 + e^{-\tau})e^a}.$$

Now suppose $a + \tau \leq w \leq b$. Differentiating w.r.t. $z$ gives

$$\frac{\mathrm{d}}{\mathrm{d}z} \frac{\sigma(z) - \sigma(w)}{z - w} = \frac{1}{z - w} \left( \sigma'(z) - \frac{\sigma(z) - \sigma(w)}{z - w} \right),$$

so the value of the slope at any critical value $|z| < a$ (excluding the endpoints) is equal to $\sigma'(z)$. It follows that

$$\min_{|z| \leq a} \frac{\sigma(z) - \sigma(w)}{z - w} \geq \min \left\{ \frac{\sigma(w) - \sigma(a)}{w - a}, \frac{\sigma(w) - \sigma(-a)}{w + a}, \sigma'(a) \right\}.$$

The first and third terms are bounded below as

$$\sigma'(a) \geq \frac{\sigma(w) - \sigma(a)}{w - a} \geq \frac{\sigma(a + \tau) - \sigma(a)}{b - a} \geq \frac{\tau \sigma'(a + \tau)}{b - a}$$

again by the mean value theorem, and we can reuse the above lower bound for $\sigma'(a + \tau)$. Moreover, the second term is larger than the first since

$$\frac{\sigma(w) - \sigma(a)}{w - a} \leq \frac{\sigma(w) - 1 + \sigma(a)}{w + a} \quad \Leftrightarrow \quad \frac{1}{w} \left( \sigma(w) - \frac{1}{2} \right) \leq \frac{1}{a} \left( \sigma(a) - \frac{1}{2} \right)$$

and the function $w \mapsto (\sigma(w) - 1/2)/w$ is decreasing. Hence, we have shown

$$\sup_{|z| \leq a, |w| \leq b} \left| \frac{\sigma(z) - \sigma(w)}{z - w} \right| \geq \frac{1}{|b - a| \vee \tau} \cdot \frac{\tau}{(2 + e^{\tau} + e^{-\tau}) e^{a \wedge b}}$$

for all $a, b, \tau > 0$; the statement follows by taking $\tau = 1$. $\qquad \square$

**Lemma C.4** (Symmetrization). *Denote by $D, \tilde{D}$ two independent datasets of i.i.d. samples and let $C(\pi, D)$ be any functional of policy $\pi$ and dataset $D$. Let $\hat{\pi} := \hat{\pi}(D)$ be any policy estimator determined by $D$ which takes values in a finite class $\Pi$. Then with probability $1 - \delta$, it holds that*

$$-\ln \mathbb{E}_{\tilde{D}}[\exp(C(\hat{\pi}, \tilde{D}))] \leq -C(\hat{\pi}, D) + \ln(|\Pi|/\delta).$$

*Proof.* See the proof of Theorem 6 of Foster & Krishnamurthy (2021). $\qquad \square$

The following technique was used in Theorem 3 of Yun et al. (2025).

**Lemma C.5.** *For $[0, 1]$-valued random variables $Z, W$,*

$$-\ln \mathbb{E} \left[ \sqrt{ZW} + \sqrt{(1 - Z)(1 - W)} \right] \geq \frac{1}{4} \mathbb{E} \left[ (Z - W)^2 \right].$$

*Proof.* It holds that

$$
\begin{aligned}
-\ln \mathbb{E} \left[ \sqrt{ZW} + \sqrt{(1 - Z)(1 - W)} \right] &\geq 1 - \mathbb{E} \left[ \sqrt{ZW} + \sqrt{(1 - Z)(1 - W)} \right] \\
&= \frac{1}{2} \mathbb{E} \left[ \left( \sqrt{Z} - \sqrt{W} \right)^2 \right] + \frac{1}{2} \mathbb{E} \left[ \left( \sqrt{1 - Z} - \sqrt{1 - W} \right)^2 \right] \\
&= \frac{1}{2} \mathbb{E} \left[ \left( \frac{Z - W}{\sqrt{Z} + \sqrt{W}} \right)^2 \right] + \frac{1}{2} \mathbb{E} \left[ \left( \frac{Z - W}{\sqrt{1 - Z} + \sqrt{1 - W}} \right)^2 \right] \\
&\geq \frac{1}{4} \mathbb{E} \left[ (Z - W)^2 \right].
\end{aligned}
$$

$\square$

**Lemma C.6.** *Let $\alpha \geq 3$, $0 < \beta < 1$. Then*

$$z \geq \frac{2}{\beta} \ln \frac{\alpha}{\beta} \quad \Rightarrow \quad \frac{\ln \alpha z}{z} \leq \beta.$$

*Proof.* Define $f(z) = (\ln \alpha z)/z$. It is easily checked that $f$ is decreasing for $z \geq 2 \ln 3$ and

$$f\left(\frac{2}{\beta} \ln \frac{\alpha}{\beta}\right) = \frac{\beta}{2}\left(\ln \frac{\alpha}{\beta}\right)^{-1}\left(\ln 2 + \ln \frac{\alpha}{\beta} + \ln \ln \frac{\alpha}{\beta}\right) \leq \frac{\beta}{2}\left(\frac{\ln 2}{\ln 3} + 1 + \frac{1}{e}\right) \approx 0.9994\beta < \beta,$$

where we have used that $\sup_{z>0} \ln z/z = 1/e$. $\qquad\square$

Recall the definition of the preference distribution $\mathbb{P}_{\pi,\pi'}$ given in Eq. (6).

**Lemma C.7.** *Given $\theta, \theta' \in \mathbb{R}^d$ and an arbitrary policy $\pi$, it holds that*

$$\|\mathbb{P}_{\pi_\theta,\pi} - \mathbb{P}_{\pi_{\theta'},\pi}\|_\infty \leq \frac{\gamma}{2}\|\theta - \theta'\|_2 \quad and \quad \|\ln \mathbb{P}_{\pi_\theta,\pi} - \ln \mathbb{P}_{\pi_{\theta'},\pi}\|_\infty \leq 2\gamma\|\theta - \theta'\|_2.$$

*Proof.* Noting that $\sigma$ is $\frac{1}{4}$-Lipschitz,

$$\left|\mathbb{P}_{\pi_\theta,\pi}(a^1 \succ a^2 \mid x) - \mathbb{P}_{\pi_{\theta'},\pi}(a^1 \succ a^2 \mid x)\right|$$
$$= \left|\sigma\left(\gamma \ln \frac{\pi_\theta(a^1 \mid x)}{\pi_\theta(a^2 \mid x)} - \gamma \ln \frac{\pi(a^1 \mid x)}{\pi(a^2 \mid x)}\right) - \sigma\left(\gamma \ln \frac{\pi_{\theta'}(a^1 \mid x)}{\pi_{\theta'}(a^2 \mid x)} - \gamma \ln \frac{\pi(a^1 \mid x)}{\pi(a^2 \mid x)}\right)\right|$$
$$\leq \frac{\gamma}{4}\left|\ln \frac{\pi_\theta(a^1 \mid x)}{\pi_\theta(a^2 \mid x)} - \ln \frac{\pi_{\theta'}(a^1 \mid x)}{\pi_{\theta'}(a^2 \mid x)}\right|$$
$$= \frac{\gamma}{4}\left|(\theta - \theta')^\top(\phi(x, a^1) - \phi(x, a^2))\right| \leq \frac{\gamma}{2}\|\theta - \theta'\|_2.$$

Moreover, $(\ln \sigma(z))' = \frac{\sigma'(z)}{\sigma(z)} = 1 - \sigma(z)$ so that $\ln \sigma$ is 1-Lipschitz, hence we similarly have

$$\left|\ln \mathbb{P}_{\pi_\theta,\pi}(a^1 \succ a^2 \mid x) - \ln \mathbb{P}_{\pi_{\theta'},\pi}(a^1 \succ a^2 \mid x)\right|$$
$$= \left|\ln \sigma\left(\gamma \ln \frac{\pi_\theta(a^1 \mid x)}{\pi_\theta(a^2 \mid x)} - \gamma \ln \frac{\pi(a^1 \mid x)}{\pi(a^2 \mid x)}\right) - \ln \sigma\left(\gamma \ln \frac{\pi_{\theta'}(a^1 \mid x)}{\pi_{\theta'}(a^2 \mid x)} - \gamma \ln \frac{\pi(a^1 \mid x)}{\pi(a^2 \mid x)}\right)\right|$$
$$\leq 2\gamma\|\theta - \theta'\|_2.$$

$\qquad\square$

**Lemma C.8** (Worst-case coverage). *Given $\theta, \theta' \in \mathbb{R}^d$, it holds that*

$$C_{\pi_\theta \to \pi_{\theta'}} \leq \sup_{x,a} \frac{\pi_{\theta'}(a \mid x)}{\pi_\theta(a \mid x)} \leq \exp(2\|\theta - \theta'\|_2).$$

*Proof.* Write

$$\pi_\theta(a \mid x) = \frac{1}{Z_x(\theta)}\pi_0(a \mid x)\exp(\theta^\top \phi(x, a)), \quad \text{where} \quad Z_x(\theta) := \mathbb{E}_{a \sim \pi_0(x)}\left[\exp(\theta^\top \phi(x, a))\right].$$

Then

$$Z_x(\theta) \leq \mathbb{E}_{a \sim \pi_0(x)}\left[\exp((\theta')^\top \phi(x, a))\right] \sup_{a \in \mathcal{A}} \exp((\theta - \theta')^\top \phi(x, a)) \leq Z_x(\theta')\exp(\|\theta - \theta'\|_2),$$

$$\frac{\pi_{\theta'}(a \mid x)}{\pi_\theta(a \mid x)} = \frac{Z_x(\theta)}{Z_x(\theta')}\exp((\theta' - \theta)^\top \phi(x, a)) \leq \exp(2\|\theta - \theta'\|_2).$$

Moreover for arbitrary $u \in \mathbb{R}^d$, we have that

$$u^\top \mathbb{V}(\pi_{\theta'})u = \mathbb{E}_{x \sim \mathcal{D}, a \sim \pi_{\theta'}(x)}\left[(u^\top \phi(x, a) - \mathbb{E}_{a \sim \pi_{\theta'}(x)}[u^\top \phi(x, a)])^2\right]$$
$$\leq \mathbb{E}_{x \sim \mathcal{D}, a \sim \pi_{\theta'}(x)}\left[(u^\top \phi(x, a) - \mathbb{E}_{a \sim \pi_\theta(x)}[u^\top \phi(x, a)])^2\right]$$
$$\leq \sup_{x,a} \frac{\pi_{\theta'}(a \mid x)}{\pi_\theta(a \mid x)} \cdot \mathbb{E}_{x \sim \mathcal{D}, a \sim \pi_\theta(x)}\left[(u^\top \phi(x, a) - \mathbb{E}_{a \sim \pi_\theta(x)}[u^\top \phi(x, a)])^2\right]$$
$$= \sup_{x,a} \frac{\pi_{\theta'}(a \mid x)}{\pi_\theta(a \mid x)} \cdot u^\top \mathbb{V}(\pi_\theta)u.$$

$\qquad\square$

The following proposition gives a tight upper bound on the KL and Rényi divergence between linear softmax policies. This is tighter than the naive density ratio approach in Lemma C.8, which only yields $2\|\theta - \theta'\|_2$.

**Proposition C.9.** *Given* $\theta, \theta' \in \mathbb{R}^d$*, it holds that*

$$\mathrm{KL}(\pi_\theta \| \pi_{\theta'}) \leq \ln\left(1 + \chi^2(\pi_\theta, \pi_{\theta'})\right) \leq \|\theta - \theta'\|_2^2.$$

*Proof.* The first inequality follows from Jensen's inequality (applied to the hidden expectation over $x \sim \mathcal{D}$) and monotonicity of Rényi divergence (van Erven & Harremoes, 2014). We proceed to show the second inequality.

Define the partition and log-partition functions

$$Z_x(\theta) := \mathbb{E}_{a \sim \pi_0(x)}\left[\exp(\theta^\top \phi(x, a))\right], \quad A_x(\theta) := \ln Z_x(\theta).$$

It holds that

$$
\begin{aligned}
\nabla^2 A_x(\theta) &= \frac{\nabla^2 Z_x(\theta)}{Z_x(\theta)} - \frac{\nabla Z_x(\theta) \nabla Z_x(\theta)^\top}{Z_x(\theta)^2} \\
&= \mathbb{E}_{a \sim \pi_\theta(x)}\left[\phi(x, a)\phi(x, a)^\top\right] - \mathbb{E}_{a \sim \pi_\theta(x)}[\phi(x, a)]\mathbb{E}_{a \sim \pi_\theta(x)}[\phi(x, a)]^\top \\
&= \mathrm{Var}_{a \sim \pi_\theta(x)}(\phi(x, a))
\end{aligned}
$$

and $\|\phi(x, a)\|_2 \leq 1$, hence $\nabla^2 A_x(\theta) \preceq \mathbf{I}_d$ for all $x, \theta$. It follows for all $\nu \in \mathbb{R}^d$ that the second order finite difference

$$
\begin{aligned}
A_x(\theta + \nu) + A_x(\theta - \nu) - 2A_x(\theta) &= \int_{-1}^{0} \int_{t}^{t+1} \frac{\mathrm{d}^2}{\mathrm{d}s^2}\bigg|_{s=t'} A_x(\theta + s\nu) \, \mathrm{d}t' \, \mathrm{d}t \\
&= \int_{-1}^{0} \int_{t}^{t+1} \nu^\top \nabla^2 A_x(\theta + t'\nu)\nu \, \mathrm{d}t' \, \mathrm{d}t \\
&\leq \|\nu\|_2^2.
\end{aligned}
$$

Now set $\nu = \theta' - \theta$. The Rényi divergence of order two can be expressed as

$$
\begin{aligned}
\ln\left(1 + \chi^2(\pi_\theta, \pi_{\theta'})\right) &= \ln \mathbb{E}_{x \sim \mathcal{D}, a \sim \pi_{\theta'}(x)}\left[\left(\frac{\pi_\theta(a \mid x)}{\pi_{\theta'}(a \mid x)}\right)^2\right] \\
&= \ln \mathbb{E}_{x \sim \mathcal{D}, a \sim \pi_{\theta'}(x)}\left[\frac{Z_x(\theta')^2}{Z_x(\theta)^2}\exp(-2\nu^\top \phi(x, a))\right] \\
&= \ln \mathbb{E}_{x \sim \mathcal{D}, a \sim \pi_0(x)}\left[\frac{Z_x(\theta')}{Z_x(\theta)^2}\exp((\theta' - 2\nu)^\top \phi(x, a))\right] \\
&= \ln \mathbb{E}_{x \sim \mathcal{D}}\left[\frac{Z_x(\theta + \nu)Z_x(\theta - \nu)}{Z_x(\theta)^2}\right] \\
&= \ln \mathbb{E}_{x \sim \mathcal{D}}\left[\exp\left(A_x(\theta + \nu) + A_x(\theta - \nu) - 2A_x(\theta)\right)\right],
\end{aligned}
$$

which is upper bounded by $\|\nu\|_2^2$ as shown. $\qquad \square$

# D. Analysis of Preferential Optimal Design

## D.1. Proof of Proposition 4.2

We first provide some auxiliary results and the proof of Lemma 4.1.

**Theorem D.1** (Kiefer-Wolfowitz equivalence theorem (Kiefer & Wolfowitz, 1960)). *Let* $\psi : \mathcal{X} \to \mathbb{R}^d$ *be such that* $\psi_1, \cdots, \psi_d$ *are linearly independent and the range of* $\psi$ *is compact. Let* $C$ *be any class of Borel probability measures on* $\mathcal{X}$ *which includes all probability measures with finite support. Denote* $M(\mu) := \mathbb{E}_{x \sim \mu}[\psi(x)\psi(x)^\top]$ *for* $\mu \in C$*. Then the following are equivalent:*

(1) $\mu$ *maximizes* $\det M(\mu)$ *(D-optimal design);*

(2) $\mu$ *minimizes* $\max_{x\in\mathcal{X}}\psi(x)^\top M(\mu)^{-1}\psi(x)$ (G-optimal design);

(3) $\mu$ *satisfies* $\max_{x\in\mathcal{X}}\psi(x)^\top M(\mu)^{-1}\psi(x) = d$.

**Lemma D.2.** *The map $\pi \mapsto \mathbb{V}(\pi)$ is concave in Loewner order.*

*Proof.* Let $\pi^1, \pi^2 \in \Delta(\mathcal{X} \times \mathcal{A})$ and $\lambda \in (0,1)$. It holds for $\pi := \lambda\pi^1 + (1-\lambda)\pi^2$ that

$$
\begin{aligned}
\mathbb{V}(\pi) &= \lambda\mathbb{E}_{x,a\sim\pi^1}\left[(\phi(x,a) - \phi(x,\pi(x)))^{\otimes 2}\right] + (1-\lambda)\mathbb{E}_{x,a\sim\pi^2}\left[(\phi(x,a) - \phi(x,\pi(x)))^{\otimes 2}\right] \\
&\succeq \lambda\mathbb{E}_{x,a\sim\pi^1}\left[(\phi(x,a) - \phi(x,\pi^1(x)))^{\otimes 2}\right] + (1-\lambda)\mathbb{E}_{x,a\sim\pi^2}\left[(\phi(x,a) - \phi(x,\pi^2(x)))^{\otimes 2}\right] \\
&= \lambda\mathbb{V}(\pi^1) + (1-\lambda)\mathbb{V}(\pi^2).
\end{aligned}
$$

$\square$

**Proof of Lemma 4.1.** Define the augmented feature map $\bar{\phi}(x,a) := (1 \ \ \phi(x,a)^\top)^\top$ and let

$$
M(x,\pi) := \mathbb{E}_{a\sim\pi}\left[\bar{\phi}\bar{\phi}^\top\right] = \begin{pmatrix} 1 & \phi(x,\pi)^\top \\ \phi(x,\pi) & \mathbb{E}_{a\sim\pi}[\phi(x,a)\phi(x,a)^\top] \end{pmatrix}
$$

be its correlation matrix. Note that $\mathbb{V}(x,\pi) = \mathbb{E}_{a\sim\pi}[\phi(x,a)\phi(x,a)^\top] - \phi(x,\pi)\phi(x,\pi)^\top$ is the Schur complement of the $(1,1)$-block of $M(x,\pi)$ and so

$$
M(x,\pi)^{-1} = \begin{pmatrix} 1 + \phi(x,\pi)^\top\mathbb{V}(x,\pi)^{-1}\phi(x,\pi) & -\phi(x,\pi)^\top\mathbb{V}(x,\pi)^{-1} \\ -\mathbb{V}(x,\pi)^{-1}\phi(x,\pi) & \mathbb{V}(x,\pi)^{-1} \end{pmatrix}.
$$

Moreover,

$$
\begin{aligned}
&\bar{\phi}(x,a)^\top M(x,\pi)^{-1}\bar{\phi}(x,a) \\
&= (1 \ \ \phi(x,a)^\top)\begin{pmatrix} 1 + \phi(x,\pi)^\top\mathbb{V}(x,\pi)^{-1}\phi(x,\pi) & -\phi(x,\pi)^\top\mathbb{V}(x,\pi)^{-1} \\ -\mathbb{V}(x,\pi)^{-1}\phi(x,\pi) & \mathbb{V}(x,\pi)^{-1} \end{pmatrix}\begin{pmatrix} 1 \\ \phi(x,a) \end{pmatrix} \\
&= 1 + (\phi(x,a) - \phi(x,\pi))^\top\mathbb{V}(x,\pi)^{-1}(\phi(x,a) - \phi(x,\pi)).
\end{aligned}
$$

Now by the Kiefer-Wolfowitz theorem (Theorem D.1), there exists[12] $\pi = \pi^{cg}(\cdot \mid x) \in \Delta(\mathcal{A})$ such that

$$
\max_{a\in\mathcal{A}}\bar{\phi}(x,a)^\top M(x,\pi)^{-1}\bar{\phi}(x,a) = \dim(\text{span}\,\bar{\phi}) = d + 1,
$$

and hence achieves $\max_{a\in\mathcal{A}}\|\phi(x,a) - \phi(x,\pi)\|^2_{\mathbb{V}(x,\pi)^{-1}} = d$. $\square$

**Proof of Proposition 4.2.** We first note that for a positive-definite matrix $A$, it holds that $x^\top A^{-1}x \leq z$ iff $zA \succeq xx^\top$. Indeed, substituting $y = (zA)^{-1/2}x$ reduces the statement to showing $\|y\| \leq 1$ iff $yy^\top \preceq \mathbf{I}_d$, which is clearly true. Hence by Lemma 4.1, we have

$$
\|\phi^{cg}(x,a)\|^2_{\mathbb{V}(x,\pi^{cg}(x))^{-1}} \leq d \quad \Rightarrow \quad d\mathbb{V}(x,\pi^{cg}(x)) \succeq \phi^g(x,a)\phi^g(x,a)^\top
$$

for all $x, a$. Since the left-hand side depends only on $x$, taking expectations on both sides w.r.t. $\mu$ yields

$$
d\mathbb{V}(\pi^g) = d\mathbb{E}_{x\sim\mu_\mathcal{X}}[\mathbb{V}(x,\pi^{cg}(x))] \succeq \mathbb{E}_{(x,a)\sim\mu}\left[\phi^g(x,a)\phi^g(x,a)^\top\right] = M(\mu).
$$

It follows that

$$
\sup_{x\in\mathcal{X},a\in\mathcal{A}}\phi^g(x,a)^\top\mathbb{V}(\pi^g)^{-1}\phi^g(x,a) \leq d \cdot \sup_{x\in\mathcal{X},a\in\mathcal{A}}\phi^g(x,a)^\top M(\mu)^{-1}\phi^g(x,a) = d^2
$$

since we took $\mu^g$ to be the (joint) G-optimal design of the features $\phi^g$. $\square$

---

[12]Existence is guaranteed since $\mathcal{X}$ is finite.

## D.2. Proof of Theorem 4.4

Similarly to Lemma C.8, the coverage of $\pi^*$ by $\hat{\pi}'_k$ is bounded as

$$C_{\hat{\pi}'_k \to \pi^*} \leq \sup_{x,a} \frac{\pi^*(x,a)}{\hat{\pi}'_k(x,a)} \leq 2 \sup_{x,a} \frac{\pi^*(a \mid x)}{\hat{\pi}_k(a \mid x)}.$$

Recall the definition $\phi^g(x,a) := \phi(x,a) - \phi(x, \pi^{cg}(x))$ where $\pi^{cg}(x)$ is the conditional G-optimal design with centering for $\phi$. We manipulate the log-likelihood ratio as follows:

$$\ln \frac{\pi^*(a \mid x)}{\hat{\pi}_k(a \mid x)} = (\theta^* - \hat{\theta}_k)^\top \phi^g(x,a) + (\theta^* - \hat{\theta}_k)^\top \phi(x, \pi^{cg}(x)) + \ln \frac{Z_x(\hat{\theta}_k)}{Z_x(\theta^*)}.$$

Note that via a change of measure,

$$\frac{Z_x(\hat{\theta}_k)}{Z_x(\theta^*)} = \mathbb{E}_{a \sim \pi_0(x)} \left[ \frac{1}{Z_x(\theta^*)} \exp(\hat{\theta}_k^\top \phi(x,a)) \right] = \mathbb{E}_{a \sim \pi^*(x)} \left[ \exp((\hat{\theta}_k - \theta^*)^\top \phi(x,a)) \right]$$

so that

$$\begin{aligned}
\ln \frac{\pi^*(a \mid x)}{\hat{\pi}_k(a \mid x)} &= (\theta^* - \hat{\theta}_k)^\top \phi^g(x,a) + \ln \mathbb{E}_{a \sim \pi^*(x)} \left[ \exp((\hat{\theta}_k - \theta^*)^\top \phi^g(x,a)) \right] \\
&\leq 2 \sup_a \left| (\hat{\theta}_k - \theta^*)^\top \phi^g(x,a) \right| \\
&\leq 2 \|\hat{\theta}_k - \theta^*\|_{\mathbb{V}(\pi^g)} \sup_a \|\phi^g(x,a)\|_{\mathbb{V}(\pi^g)^{-1}}.
\end{aligned}$$

By Proposition 4.2, it holds that $\|\phi^g(x,a)\|_{\mathbb{V}(\pi^g)^{-1}} \leq d$ for all $x, a$. Moreover for $k \geq 1$, repeating Step 1 of the proof of Proposition 3.1 at iteration $k - 1$ with the sampling distribution $\hat{\pi}_{k-1}$ replaced by $\hat{\pi}'_{k-1}$, we have that

$$\mathbb{E}_{x,a^1,a^2 \sim \hat{\pi}'_{k-1}} \left[ \left( \mathbb{P}_{\hat{\pi}_k, \pi_0}(a^1 \succ a^2 \mid x) - \mathbb{P}_*(a^1 \succ a^2 \mid x) \right)^2 \right] \leq \frac{4d}{n} \ln \frac{9 \gamma R n}{\delta}$$

and

$$\begin{aligned}
&\mathbb{E}_{x,a^1,a^2 \sim \hat{\pi}'_{k-1}} \left[ \left( \mathbb{P}_{\hat{\pi}_k, \pi_0}(a^1 \succ a^2 \mid x) - \mathbb{P}_*(a^1 \succ a^2 \mid x) \right)^2 \right] \\
&\gtrsim \frac{1}{(\gamma R e^{2\gamma R})^2} \mathbb{E}_{x,a^1,a^2 \sim \hat{\pi}'_{k-1}} \left[ \left( \gamma \ln \frac{\hat{\pi}_k(a^1 \mid x)}{\hat{\pi}_k(a^2 \mid x)} - \gamma \ln \frac{\pi_0(a^1 \mid x)}{\pi_0(a^2 \mid x)} - r^*(x,a^1) + r^*(x,a^2) \right)^2 \right] \\
&= \frac{1}{(\gamma R e^{2\gamma R})^2} \cdot 2\gamma^2 (\hat{\theta}_k - \theta^*)^\top \mathbb{V}(\hat{\pi}'_{k-1})(\hat{\theta}_k - \theta^*).
\end{aligned}$$

By Lemma D.2 it also holds that $\mathbb{V}(\hat{\pi}'_{k-1}) \succeq \frac{1}{2} \mathbb{V}(\pi^g) + \frac{1}{2} \mathbb{V}(\hat{\pi}_{k-1})$, which implies

$$\|\hat{\theta}_k - \theta^*\|_{\mathbb{V}(\pi^g)} + \|\hat{\theta}_k - \theta^*\|_{\mathbb{V}(\hat{\pi}_{k-1})} \lesssim R e^{2\gamma R} \sqrt{\frac{d}{n} \ln \frac{9 \gamma R n}{\delta}} \leq \varepsilon. \tag{13}$$

Hence taking $k = 1$, we have

$$\ln \frac{\pi^*(a \mid x)}{\hat{\pi}_1(a \mid x)} \leq 2 \|\hat{\theta}_k - \theta^*\|_{\mathbb{V}(\pi^g)} \sup_a \|\phi^g(x,a)\|_{\mathbb{V}(\pi^g)^{-1}} \lesssim d R e^{2\gamma R} \sqrt{\frac{d}{n} \ln \frac{9 \gamma R n}{\delta}} \lesssim d\varepsilon = O(1),$$

so that $C_{\hat{\pi}'_1 \to \pi^*} = \Theta(1)$, and taking $k = 2$, we conclude:

$$\|\hat{\theta}_2 - \theta^*\|_{\mathbb{V}(\pi^*)} \lesssim \|\hat{\theta}_2 - \theta^*\|_{\mathbb{V}(\hat{\pi}'_1)} \lesssim \varepsilon.$$

Note that while Eq. (13) is valid for $k \geq 1$, the initial sampling distribution $\hat{\pi}'_0$ is not guaranteed to have good coverage, and hence $k \geq 2$ is needed to ensure this can be translated into a useful bound in $\mathbb{V}(\pi^*)$-norm. $\square$

# E. Analysis of On-policy Reward Distillation

For completeness, we provide here the definition of pseudodimension of a class $\mathcal{F}$ of real-valued functions, the real-valued analogue of Vapnik-Chervonenkis dimension VCdim. A set $\{x_1, \cdots, x_m\}$ is said to be pseudo-shattered by $\mathcal{F}$ with witness vector $r = (r_1, \cdots, r_m)$, if for all $s \in \{-1, 1\}^m$ there exists $f_s \in \mathcal{F}$ such that $\mathrm{sgn}(f_s(x_i) - r_i) = s_i$ for all $i = 1, \cdots, m$. The pseudodimension $\mathrm{Pdim}(\mathcal{F})$ of $\mathcal{F}$ is defined as the cardinality of the largest set that can be pseudo-shattered by $\mathcal{F}$. Note that the VC dimension of the set of binary functions $b_f(x, r) = \mathrm{sgn}(f(x) - r)$ for $f \in \mathcal{F}$ is equal to $\mathrm{Pdim}(\mathcal{F})$. Moreover, if $\mathcal{F}$ is a linear space, $\mathrm{Pdim}(\mathcal{F}) = \dim(\mathcal{F})$.

If $\mathrm{Pdim}(\mathcal{F}) < \infty$, the fat-shattering dimension of $\Delta\mathcal{F}$ is $\widetilde{O}(\mathrm{Pdim}(\mathcal{F}))$ (Attias & Kontorovich, 2023), which can be used to bound $\mathrm{Pdim}(\Delta\mathcal{F})$ under mild structural assumptions on $\mathcal{F}$. Here, we directly use $\mathrm{Pdim}(\Delta\mathcal{F})$ for simplicity of presentation.

## E.1. Proof of Theorem 5.1

We start by bounding the error of the learned implicit rewards (Rafailov et al., 2023) against the reward model $\tilde{r}$. See Appendix E.2 for necessary definitions. Define the class of disagreement sets $\mathcal{A}_{\mathcal{F}}^{\pm} := \{A_f^{\pm} \mid f \in \mathcal{F}\}$ where

$$A_f^+ = \left\{ (x, a^1, a^2) \in \mathcal{X} \times \mathcal{A}^2 \mid \gamma \Delta f(x, a^1, a^2) > \Delta \tilde{r}(x, a^1, a^2) \right\},$$
$$A_f^- = \left\{ (x, a^1, a^2) \in \mathcal{X} \times \mathcal{A}^2 \mid \gamma \Delta f(x, a^1, a^2) < \Delta \tilde{r}(x, a^1, a^2) \right\},$$

and set $\mathcal{A}_{\mathcal{F}} := \{A_f \mid f \in \mathcal{F}\}$ where $A_f = A_f^+ \cup A_f^-$. Since any set shattered by $\mathcal{A}_{\mathcal{F}}^{\pm}$ can be pseudo-shattered by $\Delta(\mathcal{F})$ with witness $\gamma^{-1}\Delta\tilde{r}$, it holds that $\mathrm{VCdim}(\mathcal{A}_{\mathcal{F}}^{\pm}) \leq \mathrm{Pdim}(\Delta\mathcal{F})$ and hence

$$\mathrm{VCdim}(\mathcal{A}_{\mathcal{F}}) \leq \mathrm{VCdim}(\mathcal{A}_{\mathcal{F}}^+) + \mathrm{VCdim}(\mathcal{A}_{\mathcal{F}}^-) + 1 \leq 2\,\mathrm{Pdim}(\Delta\mathcal{F}) + 1.$$

Since $\tilde{r}$ is realizable in the sense of Assumption 5, $\hat{f}_{k+1}$ can attain zero loss. This implies that for all examples $(x, a^1, a^2) \in D_k$,

$$\gamma \ln \frac{\hat{\pi}_{k+1}(a^1 \mid x)}{\hat{\pi}_{k+1}(a^2 \mid x)} - \gamma \ln \frac{\pi_0(a^1 \mid x)}{\pi_0(a^2 \mid x)} = \gamma \Delta \hat{f}_{k+1}(x, a^1, a^2) = \Delta \tilde{r}(x, a^1, a^2)$$

must hold for the fixed-regularization update, and

$$\gamma \ln \frac{\hat{\pi}_{k+1}(a^1 \mid x)}{\hat{\pi}_{k+1}(a^2 \mid x)} - \gamma \ln \frac{\hat{\pi}_k(a^1 \mid x)}{\hat{\pi}_k(a^2 \mid x)} = \left( \tilde{r}_k(x, a^1) - \gamma \ln \frac{\hat{\pi}_k(a^1 \mid x)}{\hat{\pi}_0(a^1 \mid x)} \right) - \left( \tilde{r}_k(x, a^2) - \gamma \ln \frac{\hat{\pi}_k(a^2 \mid x)}{\hat{\pi}_0(a^2 \mid x)} \right)$$

for the variance-reduced update, which is clearly equivalent. Then $A_{\hat{f}_{k+1}} \cap D_k = \varnothing$, that is, $A_{\hat{f}_{k+1}}$ is consistent with $D_k$ w.r.t. the target hypothesis $A_{\gamma^{-1}\tilde{r}} = \varnothing$. By the $\varepsilon$-net theorem (see Theorem E.1), choosing

$$n \gtrsim \frac{\mathrm{Pdim}(\Delta\mathcal{F})}{\varepsilon} \ln \frac{1}{\varepsilon\delta}, \tag{14}$$

it holds with probability at least $1 - \delta$ that

$$\mathrm{Pr}_{x \sim \mathcal{D}, a^1, a^2 \sim \hat{\pi}_k(x)} \left( \gamma \Delta \hat{f}_{k+1}(x, a^1, a^2) \neq \Delta \tilde{r}(x, a^1, a^2) \right) = \mathrm{Pr}\left( A_{\hat{f}_{k+1}} \right) \leq \varepsilon.$$

We now recursively construct a sequence of bounds $(r_k)_{k \geq 0}$ such that $\mathrm{MAD}_{\pi^*}(\hat{f}_k - f^*) \leq r_k$ holds for all $k = 1, \cdots, K$ with probability at least $1 - \delta$, given initial condition $f_0 \equiv 0$ and $r_0 = 2R$. It follows from Lemma E.2 that

$$\mathrm{MAD}_{\hat{\pi}_k}(\gamma \hat{f}_{k+1} - \tilde{r})$$
$$= \mathbb{E}_{x \sim \mathcal{D}, a \sim \hat{\pi}_k(x)} \left[ \left| \gamma \hat{f}_{k+1}(x, a) - \tilde{r}(x, a) - \mathbb{E}_{a \sim \hat{\pi}_k(x)} \left[ \gamma \hat{f}_{k+1}(x, a) - \tilde{r}(x, a) \right] \right| \right]$$
$$\leq 8\gamma R \cdot \mathbb{E}_{x \sim \mathcal{D}} \left[ \mathrm{Pr}_{a^1, a^2 \sim \hat{\pi}_k(x)} \left( \gamma \hat{f}_{k+1}(x, a^1) - \tilde{r}(x, a^1) \neq \gamma \hat{f}_{k+1}(x, a^2) - \tilde{r}(x, a^2) \right) \mid x \right]$$
$$= 8\gamma R \cdot \mathrm{Pr}_{x \sim \mathcal{D}, a^1, a^2 \sim \hat{\pi}_k(x)} \left( \gamma \Delta \hat{f}_{k+1}(x, a^1, a^2) \neq \Delta \tilde{r}(x, a^1, a^2) \right)$$
$$\leq 8\gamma R\varepsilon.$$

Then the definition of $C_\mathcal{F}$ and the easily checked sublinearity of MAD,

$$
\begin{aligned}
\text{MAD}_{\pi^*}(\hat{f}_{k+1} - f^*) &\leq \text{MAD}_{\pi^*}(\hat{f}_{k+1} - \gamma^{-1}\tilde{r}) + \text{MAD}_{\pi^*}(\gamma^{-1}\tilde{r} - f^*) \\
&\leq C_{\hat{\pi}_k \to \pi^*} \text{MAD}_{\hat{\pi}_k}(\hat{f}_{k+1} - \gamma^{-1}\tilde{r}) + \text{MAD}_{\pi^*}(\gamma^{-1}\tilde{r} - f^*) \\
&\leq \frac{C_\mathcal{F}(r_k)}{\gamma} \text{MAD}_{\hat{\pi}_k}(\gamma\hat{f}_{k+1} - \tilde{r}) + \text{MAD}_{\pi^*}(\gamma^{-1}\tilde{r} - f^*) \\
&\leq 8RC_\mathcal{F}(r_k)\varepsilon + \varepsilon_{\text{RM}}.
\end{aligned}
$$

Therefore, we can set $r_{k+1}$ recursively as

$$
r_{k+1} := 2\varepsilon_{\text{RM}} \vee 16RC_\mathcal{F}(r_k)\varepsilon, \quad r_0 = 2R.
$$

The remainder of the analysis mimics that of Theorem 3.2. We may write the above relation as $r_{k+1} = 2\varepsilon_{\text{RM}} \vee \xi C_\mathcal{F}(r_k)$ where

$$
\xi = 16R\varepsilon \asymp \frac{R\,\text{Pdim}(\Delta\mathcal{F})}{n} \ln \frac{Kn}{\delta}
$$

satisfies Eq. (14) with $\delta$ replaced by $\delta/K$ due to Lemma C.6. Defining $S_z := \{r > 0 \mid \xi C_\mathcal{F}(r) \leq zr\}$, we ensure $R \in S_\eta$ and $2\xi/\eta \in S_\eta$ as long as $\xi \leq \eta R/C_\mathcal{F}(R)$ and

$$
\xi C_\mathcal{F}\left(\frac{2\xi}{\eta}\right) \leq 2\xi \quad \Leftrightarrow \quad \xi \leq \frac{C_\mathcal{F}^{-1}(2)\eta}{2} = \Theta(1)\eta.
$$

Then $(r_k)_{k \geq 0}$ is again monotone decreasing, regardless of the value of $\varepsilon_{\text{RM}}$, due to the invariance of $S_1$ under the mapping $r \mapsto \xi C_\mathcal{F}(r)$. Moreover, $r_k$ is exponentially decreasing while in $S_\eta$. As before, we conclude:

$$
r_K \leq \frac{2\xi}{\eta} \vee 2R\eta^K \vee 2\varepsilon_{\text{RM}}.
$$

The statement follows by setting $\xi_n := \xi/\eta$. $\qquad\square$

## E.2. Auxiliary Results

We require the following version of the $\varepsilon$-net theorem (Haussler & Welzl, 1987), due to Blumer et al. (1989).

**Theorem E.1** (Theorem A2.2, Blumer et al. (1989)). *Let $\mathcal{X}$ be a finite, countably infinite or Euclidean domain with probability measure $\mathcal{P}$. Let $\mathcal{H} \subseteq 2^\mathcal{X}$ be a well-behaved[13] hypothesis space of finite VC dimension $\text{VCdim}(\mathcal{H})$ and the target concept $h^*$ be any Borel set contained in $\mathcal{X}$. Then for any $\varepsilon, \delta \in (0, 1)$, given $m$ i.i.d. examples $(x_i, 1_{\{x_i \in c\}})_{i=1}^m$ of $h^*$ drawn according to $\mathcal{P}$ where*

$$
m \geq \frac{4}{\varepsilon} \ln \frac{2}{\delta} \vee \frac{8\,\text{VCdim}(\mathcal{H})}{\varepsilon} \log \frac{13}{\varepsilon},
$$

*with probability at least $1 - \delta$, every hypothesis $h \in \mathcal{H}$ that is consistent with all examples has error $\mathcal{P}(h\Delta h^*)$ at most $\varepsilon$.*

**Lemma E.2.** *Let $X$ be any random variable such that $|X| \leq a$ a.s. and $X'$ be an independent copy of $X$. Then it holds that*

$$
\mathbb{E}[|X - \mathbb{E}[X]|] \leq 4a \Pr(X \neq X').
$$

*Proof.* Let $(p_i)_{i=1}^\infty$ denote the probabilities of the atoms of $X$ in decreasing order. We have

$$
\Pr(X = X') = \sum_{i=1}^\infty p_i^2 \leq p_1,
$$

hence there exists a value $c \in [-a, a]$ such $\Pr(X = c) \geq \Pr(X = X')$. Then

$$
\mathbb{E}[|X - c|] \leq 2a \Pr(X \neq c) \leq 2a \Pr(X \neq X')
$$

---

[13]This is a benign measure-theoretic assumption which holds for virtually all cases of interest (Blumer et al., 1989).

and moreover

$$\mathbb{E}[|X - \mathbb{E}[X]|] \leq \mathbb{E}[|X - c|] + |c - \mathbb{E}[X]| \leq 2\mathbb{E}[|X - c|] \leq 4a \Pr(X \neq X').$$

$\square$

**Lemma E.3.** *In the linear softmax policy setting* $\mathcal{F} = \{f_\theta = \theta^\top \phi \mid \theta \in B_2(0, R)\}$, *it holds that* $\|\theta - \theta^*\|_2 \leq 2\lambda^{-1} \mathrm{MAD}_{\pi^*}(f_\theta - f^*)$ *and* $C_{\mathcal{F}}(r) \leq e^{4r/\lambda}$.

*Proof.* For any $\theta$ such that $\mathrm{MAD}_{\pi^*}(f_\theta - f^*) \leq r$ we have:

$$\begin{aligned}
r &\geq \mathbb{E}_{x \sim \mathcal{D}, a \sim \pi^*(x)} \left[ |(\theta - \theta^*)^\top (\phi(x,a) - \phi(x, \pi^*))| \right] \\
&\geq \frac{1}{2\|\theta - \theta^*\|_2} \mathbb{E}_{x \sim \mathcal{D}, a \sim \pi^*(x)} \left[ |(\theta - \theta^*)^\top (\phi(x,a) - \phi(x, \pi^*))|^2 \right] \\
&= \frac{1}{2\|\theta - \theta^*\|_2} (\theta - \theta^*)^\top \mathbb{V}(\pi^*)(\theta - \theta^*) \\
&\geq \frac{\lambda}{2} \|\theta - \theta^*\|_2,
\end{aligned}$$

and so bounding with density ratio as in Lemma C.8 gives $C_{\pi_\theta \to \pi^*} \leq \exp(2\|\theta - \theta^*\|_2) \leq e^{4r/\lambda}$. $\square$

# F. Proof of Lower Bounds

## F.1. Proof of Proposition 2.1

Suppose $\|\theta^*\|_\infty \leq \tau$ for some $\tau > 0$. Note that $\pi^*$ is near-uniform; writing $\pi^*(i) = e^{\theta_i^*}/Z^*$ for $Z^* = \sum_{i=1}^d e^{\theta_i^*}$, it follows that $de^{-\tau} \leq Z^* \leq de^\tau$ and hence $e^{-2\tau}/d \leq \pi^*(i) \leq e^{2\tau}/d$. Viewing $\pi^* = (\pi^*(1), \cdots, \pi^*(d))$ as a vector, from $\phi(a) = e_a$, it holds that $\mathbb{E}_{a \sim \pi^*}[e_a] = \pi^*$ and

$$\mathbb{V}(\pi^*) = \mathbb{E}_{a \sim \pi^*} \left[ (e_a - \pi^*)(e_a - \pi^*)^\top \right] = \mathrm{diag}\, \pi^* - \pi^* \pi^{*\top}.$$

It follows that for any $u \in S$,

$$\begin{aligned}
u^\top \mathbb{V}(\pi^*) u &= \sum_{i=0}^d \pi^*(i) u_i^2 - \left( \sum_{i=0}^d \pi^*(i) u_i \right)^2 = \frac{1}{2} \sum_{i,j=0}^d \pi^*(i) \pi^*(j) (u_i - u_j)^2 \\
&\geq \frac{e^{-4\tau}}{2d^2} \sum_{i,j=0}^d (u_i - u_j)^2 = \frac{e^{-4\tau}}{d} \|u\|_2^2,
\end{aligned}$$

and similarly $u^\top \mathbb{V}(\pi^*) u \leq \frac{1}{d} e^{4\tau} \|u\|_2^2$.

We now show the following claim: if $\|\theta\|_p \leq R$ then

$$\min_{1 \leq i \leq d} \mathrm{softmax}(\theta)_i \asymp \begin{cases} \frac{1}{d} e^{-R} & p = 1 \\ e^{-2^{1-1/p} R} & p > 1. \end{cases} \tag{15}$$

Since $\mathrm{softmax}(\theta + \theta^*)_i \geq e^{-2\tau} \mathrm{softmax}(\theta)_i$, this will imply by Lemma C.8 that

$$\begin{aligned}
C_*^p(R) &\leq \sup_{\theta \in B_p(\theta^*, R)} \sup_{x,a} \frac{\pi^*(a \mid x)}{\pi_\theta(a \mid x)} \\
&\asymp \frac{e^{2\tau}}{d} \sup_{\theta \in B_p(0, R)} \left( \min_{1 \leq i \leq d} \mathrm{softmax}(\theta + \theta^*)_i \right)^{-1} \\
&\asymp e^{4\tau} \begin{cases} e^R & p = 1 \\ \frac{1}{d} e^{2^{1-1/p} R} & p > 1 \end{cases}
\end{aligned}$$

as was to be shown; the first inequality is also an asymptotic equivalence as can be seen by taking $u = e_i$ above for the index achieving Eq. (15).

We proceed to prove Eq. (15). Without loss of generality, we can assume $\theta_1 \leq \cdots \leq \theta_d$ and $\|\theta\|_p = R$ by convexity; then

$$\min_{1 \leq i \leq d} \text{softmax}(\theta)_i = \frac{e^{\theta_1}}{e^{\theta_1} + \cdots + e^{\theta_d}}, \quad \sum_{i=1}^{d} |\theta_i|^p = R^p \tag{16}$$

is minimized when $\theta_1 \leq 0$ and $\theta_2, \cdots, \theta_d \geq 0$, otherwise any change in sign will decrease Eq. (16).

First suppose $p = 1$. If any two $\theta_j, \theta_k > 0$, replacing the logits with $\theta_j + \theta_k, 0$ will decrease Eq. (16) since $e^{\theta_j} + e^{\theta_k} < e^{\theta_j + \theta_k} + 1$. This implies the minimum must be attained when $\theta_2 = \cdots = \theta_{d-1} = 0$ and $\theta_d = R + \theta_1$, so that

$$\text{softmax}(\theta)_1 \geq \inf_{\theta_1 \in [-R, 0]} \frac{e^{\theta_1}}{e^{\theta_1} + d - 2 + e^{R + \theta_1}} \geq \frac{1}{(d-1)e^R + 1},$$

with equality when $\theta = (-R, 0, \cdots, 0)$.

Now consider the case $p > 1$. In Eq. (15), first fix $\theta_1$ and consider maximizing $e^{\theta_2} + \cdots + e^{\theta_d}$ under the norm constraint. We can check the function $z \mapsto e^z + e^{(c - z^p)^{1/p}}$ is strictly increasing near $z = 0$ when $p > 1$, so none of $\theta_2, \cdots, \theta_d$ can be zero in this case. Then by the method of Langrange multipliers, we can optimize

$$L(\theta) := \sum_{i=2}^{d} e^{\theta_i} - \lambda \left( \sum_{i=1}^{d} |\theta_i|^p - R^d \right) \quad \Rightarrow \quad \frac{dL}{d\theta_i} = e^{\theta_i} - \lambda p |\theta_i|^{p-1} = 0.$$

This implies $h(\theta_2) = \cdots = h(\theta_d) = \lambda p$ where $h(z) = e^z |z|^{1-p}$. Moreover the $h$ is decreasing on $(0, p-1)$ and increasing on $(p-1, \infty)$, so $h(z) = \lambda p$ has at most two roots $\alpha, \beta$ where the smaller root $\alpha \leq p - 1$. Thus we may write

$$\theta_1 = -\delta, \quad \theta_2 = \cdots = \theta_{d-k} = \alpha, \quad \theta_{d-k+1} = \cdots = \theta_d = \beta$$

for some integer $k < d$, subject to the condition

$$\delta, \alpha, \beta \geq 0, \quad \alpha \leq p - 1, \quad \delta^p + (d - k - 1)\alpha^p + k\beta^p = R^p. \tag{17}$$

Then Eq. (15) is bounded below as

$$\text{softmax}(\theta)_1 \geq \frac{e^{-\delta}}{e^{-\delta} + (d - k - 1)e^\alpha + ke^\beta} \geq \frac{1}{1 + de^{R + p - 1} + ke^{\beta + \delta}}, \tag{18}$$

so we instead look at maximizing $\beta + \delta + \ln k$ under the weaker condition $\delta^p + k\beta^p \leq R^p$. Fixing $k$ and differentiating w.r.t. $\beta$ yields the solution

$$\frac{d}{d\beta}(\beta + \delta) = \frac{d}{d\beta}\left( \beta + (R^p - k\beta^p)^{1/p} \right) = 1 - (R^p - k\beta^p)^{1/p - 1}k\beta^{p-1} = 0$$

$$\Rightarrow \quad \beta = \left( k + k^{\frac{p}{p-1}} \right)^{-1/p} R, \quad \delta = \left( k + k^{\frac{p}{p-1}} \right)^{-1/p} k^{\frac{1}{p-1}} R,$$

so that

$$\beta + \delta + \ln k = \left( k + k^{\frac{p}{p-1}} \right)^{-1/p} \left( 1 + k^{\frac{1}{p-1}} \right) R + \ln k = \left( 1 + k^{-\frac{1}{p-1}} \right)^{1 - 1/p} R + \ln k.$$

Since $k \leq d = O(\text{poly}(R))$, the second term is always dominated by the first, so the maximum occurs when $k = 1$ and $\beta = \delta = 2^{-1/p} R$. Substituting in Eq. (18), we conclude $\text{softmax}(\theta)_1 \gtrsim e^{-2^{1-1/p}R}$, and equality is indeed satisfied (up to constant order) when $\theta = (-2^{-1/p}R, 2^{-1/p}R, 0, \cdots, 0)$. $\quad \square$

### F.2. Proof of Theorem 2.2

We consider the cases $p = 1$ and $p > 1$ separately and give different constructions of the target hypothesis class $\Theta^*$.

**Case I: $p = 1$.** Let $\mathcal{X} = \varnothing$, $\mathcal{A} = \{0, \cdots, d\}$ and features $\phi(i) = e_i \in \mathbb{R}^{d+1}$ where we number the coordinates $0$ (extra coordinate) through $d$. We assume $d$ is even for simplicity of construction. For radius $R > \ln d + 1$, define $\varepsilon = de^{1-R} \in (0, 1)$ and define $\pi_0 \in \Delta(\mathcal{A})$ as

$$\pi_0(0) = 1 - \varepsilon, \quad \pi_0(1) = \cdots = \pi_0(d) = \frac{\varepsilon}{d}.$$

Note that $\pi_\theta = \text{softmax}(\theta - \varphi)$ for all $\theta$, and in particular $\pi_\varphi \overset{d}{=} \text{Unif}(\mathcal{A})$ for $\varphi = (\ln \frac{\varepsilon}{d(1-\varepsilon)}, 0, \cdots, 0)$. Set the target class $\Theta^* = B_1(\varphi, 1)$. Since $\|\varphi\|_1 = |\ln \frac{\varepsilon}{d(1-\varepsilon)}| \leq R - 1$, it holds that $\Theta^* \subset B_1(0, R)$ due to the triangle inequality.

Next, let $\mathcal{V} = \{v \in \{1, -1\}^d \mid v^\top \mathbf{1} = 0\}$ be the space of balanced binary codewords of length $d$ and denote the Hamming distance on $\mathcal{V}$ by $d_{\text{Ham}}$. Since the space of balanced codewords is exponentially large, by a straightforward modification of the Gilbert-Varshamov bound, we can show there exist $v_1, \cdots, v_M \in \mathcal{V}$ and universal constants $c_1, c_2 > 0$ such that

$$\min_{j \neq k} d_{\text{Ham}}(v_j, v_k) \geq c_1 d, \quad \ln M \geq c_2 d. \tag{19}$$

Fix $\tau \in (0, 1/d]$ to be defined later and define $\pi_j := \pi_{\theta_j}$ for $j = 1, \cdots, M$ where

$$\theta_j := \text{concat}\left(\ln \frac{\varepsilon}{d(1 - \varepsilon)}, \tau v_j\right).$$

We can verify that $\|\tau v_j\|_1 \leq \tau\sqrt{d} \cdot \|v_j\|_2 \leq d\tau \leq 1$ so that $\theta_j \in \Theta^* \subset B_1(0, R)$. Moreover, we have from Eq. (19) that for $j \neq k$,

$$2\tau\sqrt{c_1 d} \leq \|\theta_j - \theta_k\|_2 = \tau\|v_j - v_k\|_2 \leq 2\tau\sqrt{d}. \tag{20}$$

Suppose the ground truth $\theta^* = \theta_j$ for $1 \leq j \leq M$, so that $r^*(a) = \theta_j^\top \phi(a) = \theta_{j,a}$ for $a = 0, 1, \cdots, d$. We observe a preference dataset $D$ of $n$ i.i.d. samples $(a^1, a^2, y) \sim \mathcal{P}_j$, where the joint distribution $\mathcal{P}_j$ is defined as

$$(a^1, a^2) \sim \pi_0^{\otimes 2}, \quad y \sim \text{Bern}(\sigma(r^*(a^1) - r^*(a^2))) = \text{Bern}(\sigma(\theta_{j,a^1} - \theta_{j,a^2})). \tag{21}$$

By the chain rule for KL divergence and Lemma F.2,

$$\begin{aligned}
\text{KL}(\mathcal{P}_j \| \mathcal{P}_k) &= \mathbb{E}_{a^1, a^2 \sim \pi_0}[\text{KL}((\mathcal{P}_j)_y \| (\mathcal{P}_k)_y)] \\
&= \mathbb{E}_{a^1, a^2 \sim \pi_0}\left[\text{KL}(\text{Bern}(\sigma(\theta_{j,a^1} - \theta_{j,a^2})) \| \text{Bern}(\sigma(\theta_{k,a^1} - \theta_{k,a^2})))\right] \\
&\leq \frac{1}{8}\mathbb{E}_{a^1, a^2 \sim \pi_0}\left[(\theta_{j,a^1} - \theta_{j,a^2} - \theta_{k,a^1} + \theta_{k,a^2})^2\right] \\
&= \frac{1}{4}\mathbb{E}_{a \sim \pi_0}\left[(\theta_{j,a} - \theta_{k,a})^2\right] \\
&= \frac{\varepsilon}{4d}\|\theta_j - \theta_k\|_2^2 \leq \varepsilon\tau^2
\end{aligned}$$

due to Eq. (20). Now let $J \sim \text{Unif}(\{1, \cdots, M\})$ be a random index and suppose $D$ is sampled using $\theta^* = \theta_J$, so $D \sim \mathcal{P}_J^{\otimes n}$ conditioned on $J$. We have from the convexity of KL divergence that

$$I(D, J) \leq \frac{1}{M^2} \sum_{j,k=1}^M \text{KL}(\mathcal{P}_j^{\otimes n} \| \mathcal{P}_k^{\otimes n}) \leq n\varepsilon\tau^2.$$

Then for large enough $d$, by taking

$$\tau^2 = \frac{c_2 d}{4n\varepsilon} \wedge \frac{1}{d^2}, \tag{22}$$

we can ensure $I(D, J) \leq \frac{1}{4}\ln M$ and $\ln 2 \leq \frac{1}{4}\ln M$. Hence for any estimator $\hat{J} = \hat{J}(D)$ of $J$, the estimation error $\Pr(\hat{J} \neq J)$ must be at least $1/2$ by Fano's inequality.

Now denote the linear projection to $S = \{\mathbf{1}\}^\perp \subset \mathbb{R}^{d+1}$ by $\Pi_S = \mathbf{I}_d - \frac{1}{d+1}\mathbf{1}\mathbf{1}^\top$. The operator norm of $\Pi_S$ w.r.t. the $L^1$-norm is

$$\|\Pi_S\|_{1\to 1} = \max_j \sum_{i=0}^d |(\Pi_S)_{ij}| = \frac{2d}{d+1}. \tag{23}$$

We will apply the above error guarantee to the nearest-neighbor index estimator w.r.t. projected 1-norm,

$$\hat{J} := \underset{1\le j\le M}{\arg\min} \|\Pi_S(\theta_j - \hat{\theta})\|_1,$$

with ties broken arbitrarily. If $\hat{J} \ne J$, noting that $\theta_J - \theta_{\hat{j}} = \text{concat}(0, \tau(v_J - v_{\hat{j}})) \in S$, by the triangle inequality and Eq. (23),

$$
\begin{aligned}
\|\hat{\theta} - \theta_J\|_1 &\ge \frac{d+1}{2d}\|\Pi_S(\hat{\theta} - \theta_J)\|_1 \\
&\ge \frac{d+1}{2d}\left(\frac{1}{2}\|\Pi_S(\hat{\theta} - \theta_{\hat{j}})\|_1 + \frac{1}{2}\|\Pi_S(\hat{\theta} - \theta_J)\|_1\right) \\
&\ge \frac{d+1}{4d}\|\Pi_S(\theta_J - \theta_{\hat{j}})\|_1 = \frac{d+1}{4d}\|\theta_J - \theta_{\hat{j}}\|_1 \ge \frac{\tau c_1(d+1)}{2}.
\end{aligned} \tag{24}
$$

Therefore, the minimax rate is lower bounded as

$$
\begin{aligned}
\inf_{\hat{\theta}} \sup_{\theta^*\in\Theta^*} \mathbb{E}_{D\sim\pi^*}\left[\|\hat{\theta} - \theta^*\|_1\right] &\ge \inf_{\hat{\theta}} \mathbb{E}_J\left[\mathbb{E}_{D\sim\pi_J}\left[\mathbf{1}_{\{\hat{j}\ne J\}}\|\hat{\theta} - \theta_J\|_1\right]\right] \\
&\ge \Pr(\hat{J} \ne J) \cdot \frac{\tau c_1(d+1)}{2} \\
&\gtrsim d\sqrt{\frac{d}{n\varepsilon} \wedge \frac{1}{d^2}} \\
&\asymp d\sqrt{\frac{C_*^1(R)}{n} \wedge 1},
\end{aligned}
$$

since $C_*^1(R) \asymp d/\varepsilon$ by Proposition 2.1.

For the KL lower bound, we instead consider the estimator

$$\hat{J} := \underset{1\le j\le M}{\arg\min} \|\Pi_S(\theta_j - \hat{\theta})\|_2.$$

By Eq. (20),

$$
\begin{aligned}
\|\Pi_S(\hat{\theta} - \theta_J)\|_2 &\ge \frac{\|\Pi_S(\hat{\theta} - \theta_{\hat{j}})\|_2 + \|\Pi_S(\hat{\theta} - \theta_J)\|_2}{2} \\
&\ge \frac{\|\Pi_S(\theta_J - \theta_{\hat{j}})\|_2}{2} = \frac{\|\theta_J - \theta_{\hat{j}}\|_2}{2} \ge \tau\sqrt{c_1 d}.
\end{aligned} \tag{25}
$$

It also holds that $\|\theta_J - \varphi\|_\infty = \tau$ and $\theta_J - \varphi \in S$ since each $v_j$ is balanced, hence $\theta_J - \varphi \in B_\infty(0,\tau)$. Now define the parameter

$$\zeta := \frac{\Pi_S(\hat{\theta} - \theta_J)}{\|\Pi_S(\hat{\theta} - \theta_J)\|_\infty \vee 1} + \Pi_S\theta_J.$$

$\zeta$ lies on the line segment between $\Pi_S\theta_J$, $\Pi_S\hat{\theta}$ so that $\zeta \in S$ and

$$\|\zeta - \Pi_S\varphi\|_\infty \le 1 + \|\Pi_S(\theta_J - \varphi)\|_\infty = 1 + \|\theta_J - \varphi\|_\infty = 1 + \tau,$$

so $\zeta - \Pi_S\varphi \in B_\infty(0, 1+\tau)$. Therefore we may apply Lemma F.1 to $\theta_J - \varphi = \Pi_S\theta_J - \Pi_S\varphi$ and $\zeta - \Pi_S\varphi$ via the equality $\pi_\theta = \mathrm{softmax}(\theta - \varphi) = \mathrm{softmax}(\theta - \Pi_S\varphi)$, which yields:

$$
\begin{aligned}
\mathrm{KL}(\pi_J \| \pi_\zeta) &\geq \frac{e^{-4-4\tau}}{2(d+1)} \|\zeta - \Pi_S\theta_J\|_2^2 \\
&\geq \frac{e^{-4-4\tau}}{2(d+1)} \frac{\|\Pi_S(\hat\theta - \theta_J)\|_2^2}{\|\Pi_S(\hat\theta - \theta_J)\|_\infty^2 \vee 1} \\
&\geq \frac{e^{-4-4\tau}}{2(d+1)} \left( \|\Pi_S(\hat\theta - \theta_J)\|_2^2 \wedge 1 \right).
\end{aligned}
$$

Furthermore, we have $\mathrm{KL}(\pi_J\|\hat\pi) \geq \mathrm{KL}(\pi_J\|\pi_\zeta)$ due to convexity of the map $\theta \mapsto \mathrm{KL}(\pi_J\|\pi_\theta)$ along the segment between $\Pi_S\theta_J$, $\Pi_S\hat\theta$ (Lemma F.1). We conclude from Eq. (25):

$$
\begin{aligned}
\inf_{\hat\pi} \sup_{\theta^* \in \Theta^*} \mathbb{E}_{D\sim\pi^*} [\mathrm{KL}(\pi^*\|\hat\pi)] &\geq \inf_{\hat\pi} \mathbb{E}_J [\mathbb{E}_{D\sim\pi_J}[\mathrm{KL}(\pi_J\|\hat\pi)]] \\
&\geq \inf_{\hat\theta} \frac{e^{-4-4\tau}}{2(d+1)} \mathbb{E}_J \left[ 1_{\{\hat{J}\neq J\}} \left( \|\Pi_S(\hat\theta - \theta_J)\|_2^2 \wedge 1 \right) \right] \\
&\geq \frac{e^{-4-4\tau}}{2(d+1)} \Pr(\hat{J} \neq J)\tau^2 c_1 d \\
&\asymp \frac{C_*^1(R)}{n} \wedge \frac{1}{d^2}.
\end{aligned}
$$

$\square$

**Case II: $p > 1$.** Let $\mathcal{X} = \varnothing$, $\mathcal{A} = \{1, \cdots, d\}$ and features $\phi(i) = e_i \in \mathbb{R}^d$. Set

$$
\pi_0 = \mathrm{softmax}(-\varphi), \quad \varphi = (-2^{-1/p}(R-1), 2^{-1/p}(R-1), 0, \cdots, 0).
$$

Set the target class $\Theta^* = B_p(\varphi, 1) \subset B_p(0, R)$ and define $\theta_j := j\tau e_2 + \varphi$ for $j \in \{0, \pm 1, \cdots, \pm M\}$ for some sufficiently large $M = \Theta(1)$. It holds that all $\theta_j \in \Theta^*$ as long as $M\tau \leq 1$. Again defining the sample distribution $(a^1, a^2, y) \sim \mathcal{P}_j$ as Eq. (21),

$$
\mathrm{KL}(\mathcal{P}_j\|\mathcal{P}_k) \leq \frac{1}{4}\mathbb{E}_{a\sim\pi_0}\left[(\theta_{j,a} - \theta_{k,a})^2\right] \leq \pi_0(2)M^2\tau^2 \asymp e^{-2^{1-1/p}R}M^2\tau^2
$$

and so $I(D, J), \ln 2 \leq \frac{1}{4}\ln(2M+1)$ by taking $M$ sufficiently large and

$$
\tau^2 \asymp \frac{e^{2^{1-1/p}R}\ln(2M+1)}{M^2 n} \wedge \frac{1}{M}.
$$

Hence Fano's inequality applies and the estimation error of the following estimator is at least $1/2$:

$$
\hat{J} := \operatorname*{arg\,min}_{1 \leq j \leq M} \|\Pi_S(\theta_j - \hat\theta)\|_p.
$$

Note that for all $x \in \mathbb{R}^d$, $\|\Pi_S x\|_\infty = \max_{1 \leq i \leq d} |x_i - \bar{x}| \leq 2\|x\|_\infty$ so that $\|\Pi_S\|_{\infty\to\infty} \leq 2$, moreover $\|\Pi_S\|_{1\to1} < 2$ by Eq. (23). By the Riesz–Thorin interpolation theorem, it follows that $\|\Pi_S\|_{p\to p} \leq 2$ for all $p > 1$, so

$$
\|\hat\theta - \theta_J\|_p \geq \frac{1}{2}\|\Pi_S(\hat\theta - \theta_J)\|_p \geq \frac{1}{4}\|\theta_J - \theta_{\hat{j}}\|_p \gtrsim 1_{\{\hat{J}\neq J\}}\tau.
$$

Since $C_*^p(R) \asymp e^{2^{1-1/p}R}/d$ by Proposition 2.1, we conclude:

$$
\inf_{\hat\theta} \sup_{\theta^* \in \Theta^*} \mathbb{E}_{D\sim\pi^*}\left[\|\hat\theta - \theta^*\|_p\right] \gtrsim \frac{e^{2^{-1/p}R}}{\sqrt{n}} \wedge 1 \asymp \sqrt{\frac{dC_*^p(R)}{n}} \wedge 1.
$$

$\square$

## F.3. Auxiliary Results

**Lemma F.1** (Positive local curvature of KL). *Let $\tau > 0$ and consider the distribution $\pi_\theta := \mathrm{softmax}(\theta)$ defined over the domain*

$$S \cap B_\infty(0, \tau) = \left\{ \theta \in \mathbb{R}^d \mid \theta^\top \mathbf{1} = 0, \ \ \|\theta\|_\infty \leq \tau \right\}. \tag{26}$$

*Then for arbitrary $\theta, \theta' \in S \cap B_\infty(0, \tau)$, the map $\theta \mapsto \mathrm{KL}(\pi_{\theta'} \| \pi_\theta)$ is convex and*

$$\frac{e^{-4\tau}}{2d} \|\theta - \theta'\|_2^2 \leq \mathrm{KL}(\pi_{\theta'} \| \pi_\theta) \leq \frac{e^{4\tau}}{2d} \|\theta - \theta'\|_2^2.$$

*Proof.* The domain $S \cap B_\infty(0, \tau)$ is clearly convex. Identify $\pi_\theta$ with $(\pi_\theta(1), \cdots, \pi_\theta(d)) \in \mathbb{R}^d$ and define the map $h : S \cap B_\infty(0, \tau) \to \mathbb{R}_{\geq 0}$ as $h(\theta) := \mathrm{KL}(\pi_{\theta'} \| \pi_\theta)$. A standard computation shows

$$\nabla h(\theta) = \pi_\theta - \pi_{\theta'}, \quad \nabla^2 h(\theta) = \mathrm{diag}\, \pi_\theta - \pi_\theta \pi_\theta^\top$$

when restricted to $S$, and $h(\theta') = 0, \nabla h(\theta') = 0$. Then for any $u \in \mathbb{R}^d$,

$$u^\top \nabla^2 h(\theta) u = \sum_{i=1}^d \pi_\theta(i) u_i^2 - \left( \sum_{i=1}^d \pi_\theta(i) u_i \right)^2 = \frac{1}{2} \sum_{i,j=1}^d \pi_\theta(i) \pi_\theta(j) (u_i - u_j)^2.$$

Moreover as in Proposition 2.1, it follows from $\|\theta\|_\infty \leq \tau$ that

$$\frac{e^{-2\tau}}{d} \leq \pi_\theta(i) \leq \frac{e^{2\tau}}{d}, \quad i = 0, \cdots, d,$$

and so for any $u \in S$,

$$\frac{e^{-4\tau}}{d} \|u\|_2^2 = \frac{1}{2} \left( \frac{e^{-2\tau}}{d} \right)^2 \sum_{i,j=0}^d (u_i - u_j)^2 \leq u^\top \nabla^2 h(\theta) u \leq \frac{1}{2} \left( \frac{e^{2\tau}}{d} \right)^2 \sum_{i,j=0}^d (u_i - u_j)^2 = \frac{e^{4\tau}}{d} \|u\|_2^2.$$

The claim now follows from convexity of $\Theta_\infty(\tau)$ and the identity

$$h(\theta) = \int_0^1 (1-t)(\theta' - \theta)^\top \nabla^2 h(\theta + t(\theta' - \theta))(\theta' - \theta) \, \mathrm{d}t.$$

$\square$

**Lemma F.2.** *For $z, w \in (0, 1)$, it holds that $\mathrm{KL}(\mathrm{Bern}(\sigma(z)) \| \mathrm{Bern}(\sigma(w))) \leq \frac{1}{8} |z - w|^2$.*

*Proof.* Define $f_z(w) := \mathrm{KL}(\mathrm{Bern}(\sigma(z)) \| \mathrm{Bern}(\sigma(w)))$ so that

$$f_z(w) = \sigma(z) \ln \frac{\sigma(z)}{\sigma(w)} + (1 - \sigma(z)) \ln \frac{1 - \sigma(z)}{1 - \sigma(w)} = \ln \frac{\sigma(z)}{\sigma(w)} + (1 - \sigma(z))(w - z).$$

A simple computation gives $f_z(z) = f_z'(z) = 0$ and $f_z''(w) = \sigma(w)(1 - \sigma(w)) \in [0, \frac{1}{4}]$. It follows from Taylor's theorem that $f_z(w) \leq \frac{1}{8} |z - w|^2$.

$\square$

