# OpenReview forum: "Coverage Improvement and Fast Convergence of On-policy Preference Learning"
_ICML.cc/2026/Conference — ICML 2026 regular_

### Official Review · Reviewer_BFEx · 2026-03-05

**Soundness:** 2
**Presentation:** 3
**Significance:** 3
**Originality:** 3
**Overall Recommendation:** 5
**Confidence:** 3

**Summary:**

This paper establishes theoretical guarantees for on-policy Direct Policy Optimization (DPO) algorithms. It seeks to demonstrate the advantage of on-policy DPO by showing a strictly better sample complexity compared to its off-policy counterpart. Furthermore, by incorporating the G-optimal design technique into on-policy DPO, the paper proposes a novel algorithm that removes the dependence on both the coverage number and $\lambda$ in the upper bound. Finally, it explores on-policy reward distillation.

**Compliance With Llm Reviewing Policy:**

Affirmed.

**Final Justification:**

The rebuttal has resolved my primary concerns, leading to an increased score of 5 (Accept). The paper's strength lies in its rigorous theoretical analysis of DPO, which outweighs its minor limitations in restrictive assumptions and presentation. The insights provided are valuable and technically sound. I recommend acceptance and encourage the authors to carefully incorporate the reviewers' suggestions to further polish the final revision.

**Key Questions For Authors:**

1. Could you please clarify the derivation of Eq. (12)? Specifically, it is unclear where the last two terms, $\frac{1}{4\gamma^2 C_*^p(R)}$ and $\frac{\eta^2}{e^2}$, originate from.

2. I have some reservations regarding the soundness of Assumption 4, specifically concerning the following two points:
    - The existence of $\bar{C}\_{\*}^p$  is well-justified. However, it cannot be used to simply replace $C^{p}\_{\*}$; since $C\_{\pi\to \pi'}$ is defined as an infimum, a direct substitution would contradict this definition.
    - Substituting $C_*^p$ with $\bar{C}\_{\*}^p$ introduces a scaling factor of $\bar{C}\_{\*}^p / C\_{\*}^p$ into the final result. This ratio could be arbitrarily large (e.g., if $\bar{C}\_{\*}^p = (C_*^p)^2$), which might render the final upper bound vacuous and compromise the theoretical significance of the rate.

**Limitations:**

Please refer to the ‘Weaknesses’ part.

**Strengths And Weaknesses:**

**Strengths**
1. The paper establishes a non-asymptotic lower bound for off-policy DPO, demonstrating an $\Omega(1/\sqrt{n})$ convergence rate with respect to the dataset size $n$.

2. Under Assumption 4, the authors derive an $O(1/n)$ upper bound for on-policy DPO, where $n$ is the dataset size per round.

3. By integrating the G-optimal design technique into the on-policy DPO framework, the authors propose a novel algorithm that successfully eliminates the dependence on both the coverage number and $\lambda$ in the upper bound.

4. Overall, the mathematical derivations are rigorous and technically sound.




**Weaknesses**

1. About the soundness: The current comparison between Theorem 2.1 and Theorem 3.2 is insufficient to conclusively demonstrate the superiority of on-policy DPO.

     - Mismatched Assumptions: Theorem 3.2 relies on an additional assumption (Assumption 4), whereas Theorem 2.1 establishes a lower bound in a more general setting. The argument would be significantly more convincing if the authors could either provide a lower bound under this additional assumption, or derive an upper bound for the general case.
     - Inconsistent Sample Size Definitions: In Theorem 3.2, $n$ denotes the dataset size per round, rather than the total number of samples. To make a fair and rigorous comparison with the off-policy setting, the authors need to compare the total sample complexity required by the algorithm, as presented in Corollary 3.3.

2.	About the experiments: There appears to be a discrepancy between the theoretical results and the empirical evidence. The proposed theory suggests that on-policy methods require significantly lower sample complexity than off-policy methods to achieve a comparable error rate. However, the current experiments only demonstrate the superior performance of on-policy methods without specifying the size of data utilized. To fully bridge this gap, it is essential to provide a comparative analysis of sample efficiency (e.g., performance vs. number of samples) for both settings."

3.	Minor issue (Lines 979-981): A factor of $2$ appears to be missing when applying the inequality $(a+b)^2 \le 2a^2 + 2b^2$.

---

> ### Author Rebuttal · Authors · 2026-03-27
>
> Thank you for the thorough review! We first mention that the reviewer's main concerns (in Weaknesses) may stem from several points of confusion; please let us explain these points in more detail below.
>
> ### Weaknesses
> **Mismatched Assumptions: Thm 3.2 relies on an additional assumption (Assumption 4), whereas Thm 2.1 establishes a lower bound in a more general setting.**
> * In fact, this is exactly what we do in Appendix F! The minimax lower bound is proved in a constructed setting where we explicitly verify $C^p(R) \asymp e^{2^{1-1/p}R}$ (Prop F.1), which indeed satisfies Assp. 4. We remark that this is the standard way to show information-theoretic coverage lower bounds, for instance see Prop 2.1 of Xie et al. ('24) and Thm 2.1 of Foster et al. ('25) cited in our paper.
> * Furthermore, our analysis extends these results, which essentially only show $n\ge e^{\Theta(R)}$ is needed to begin learning. In contrast, we take a more sophisticated Yang-Barron approach and prove a **minimax** lower bound which holds for all $n$. This is crucial to show our main separation (Corollary 3.3).
>
> **Inconsistent Sample Size Definitions: To make a fair and rigorous comparison with the off-policy setting, the authors need to compare the total sample complexity required by the algorithm.**
> * Indeed, $n$ in Thm 3.2 denotes the batchsize per round, and is not meant to be directly compared to the lower bound (it is only a convergence result so there is no "unfairness") -- that is exactly what Corollary 3.3 is for. Corollary 3.3 states that the **total** sample complexity of on-policy DPO is computed by multiplying batchsize $n$ with number of rounds $K$, which yields $n\_{on} = \widetilde{O}(\frac{R^2}{\epsilon^2}\vee (\ln\frac{R}{\epsilon}) C^p(R))$. In contrast, the sample complexity of offline DPO is lower bounded by $n\_{off}\ge \Omega(\frac{1}{\epsilon^2}C^p(R))$, no matter how the $n_{off}$ samples are used. This is the rigorous separation that we claim.
>
> **The current experiments only demonstrate the superior performance of on-policy methods without specifying the size of data utilized. It is essential to provide a comparative analysis of sample efficiency (e.g., performance vs. number of samples) for both settings.**
> * The total number of samples trained on is **always equal** for the offline and on-policy runs: $n\_{on} = n\_{off} = K|D|$ where $D$ is the prompt dataset size, and $K=3/5$. For TL;DR, for on-policy DPO, we generate two candidate responses for each prompt in the TL;DR dataset for each update, over $5$ iterations. For offline DPO, we train for $5$ epochs on the dataset generated in the first round of on-policy DPO (we have also experimented with the original dataset, but this only results in worse performance). All other hyperparameters (LR schedule, $\gamma$, batchsize) are also set to be equal to ensure a fair comparison. The same is true for the General Chat task and distillation experiments. Hence, our results directly corroborate Corollary 3.3; on-policy methods substantially outperform their offline counterparts with the same sample budget.
>
> ### Questions
> **Could you please clarify the derivation of Eq. (12)?**
> For the 2nd term: since $\eta<1$ and $\xi_n \lesssim \frac{1}{C^p(R)}$, we have $\xi = \frac{\eta^2}{\gamma^2}\xi_n \lesssim \frac{1}{\gamma^2 C^p(R)}$. For the 3rd term: since $\xi_n \lesssim \frac{1}{C^p(R)} \lesssim \gamma^2$, we have $\xi = \frac{\eta^2}{\gamma^2}\xi_n \lesssim \eta^2$. In both cases, we may tune the proportionality constant in Eq. (5) to get the desired constant factors.
>
> **I have some reservations regarding the soundness of Assumption 4.**
> * Regarding substitution of $\bar{C}^p$: what we mean is that even if Assp. 4 does not hold for $C^p$, all theorems still hold when each instance of $C^p$ is replaced by $\bar{C}^p$ (which does satisfy the growth conditions *described* in Assp. 4). We will make this clearer in the main text.
> * Regarding the scaling factor $\bar{C}^p/C^p$: we concede that this factor may be suboptimal. Our analysis of coverage dynamics fundamentally relies on $C(r)$ improving as $r$ decreases, and the simplest sufficient condition is convexity. That said, we point out:
>
> (1) The naive upper bound $C^p(r) \lesssim e^{\Theta(r)}$ is tight in worst-case sense (we prove this in Prop F.1), so this is the concrete bound that will ordinarily be used unless one can prove a problem-specific tighter bound. In particular, this bound does satisfy Assp. 4.
>
> (2) This is only a sufficient but not necessary condition. Our framework is very general: $C^p$ is always monotone increasing, so by choosing $n_k$ large enough (depending on the current $C^p$), one can always obtain a tighter confidence region with improved coverage for the next batch. Assumption 4 is just one way to obtain concrete convergence rates w.r.t $k$.
>
> We hope that this answers most questions, and humbly ask the reviewer to consider raising their score if their concerns have been satisfactorily addressed.

---

> > ### Author Rebuttal · Reviewer_BFEx · 2026-04-02
> >
> > Thank you for the detailed responses. I appreciate the authors' efforts in addressing my concerns, and most of them have been resolved. Consequently, I will raise my score to 5 (Accept).
> >
> > The paper is mathematically sound and provides a thorough theoretical comparison between on-policy and off-policy DPO, which is a valuable contribution. However, there are a few remaining points that could further strengthen the manuscript:
> >
> > **Presentation**: To improve clarity, it would be beneficial to explicitly state the underlying assumptions for each theorem within the main text, even if in an informal version.
> >
> > **Assumptions**: Some of the theoretical assumptions feel slightly restrictive; providing more justification for these would help the reader better understand their necessity.
> >
> > **Novelty**: The proofs are technically correct; however,  the methodology appears to rely on well-established statistical tools. The paper could be further strengthened if the authors could further highlight the unique challenges or insights gained from applying these techniques to this specific DPO setting.
> >
> > Overall, I encourage the authors to carefully incorporate the suggestions from all reviewers into the final version.

---

> > > ### Author Response · Authors · 2026-04-03
> > >
> > > Thank you for following up on the review and giving helpful and detailed feedback! We will be sure to incorporate your suggestions into the manuscript.
> > >
> > > Regarding novelty, we acknowledge some tools are borrowed from existing results: the symmetrization inequality Lemma C.4 is used to prove Theorem 6 of Foster & Krishnamurthy (2021), Lemma C.5 is used to bound the squared differences of the preference distributions in Theorem 4 of Yun et al. (2025), and optimal experimental design dates back over a century. These are discussed in the paper (e.g., Remark C.2), and we will make these connections clearer. Nonetheless, the main idea of the paper to combine these techniques to study the *iterative improvement* of coverage, and the obtained fast convergence & exponential separation results, are new to the best of our knowledge. Also, the elegant proof of existence of the “preferential” G-optimal design was quite technically nontrivial and surprising to us, and highlight the potential benefits of experimental design principles in post-training.

---

### Official Review · Reviewer_VGdf · 2026-03-05

**Soundness:** 3
**Presentation:** 3
**Significance:** 2
**Originality:** 3
**Overall Recommendation:** 4
**Confidence:** 3

**Summary:**

This work studies the problem of on-policy preference learning under the linear contextual bandit setting with Bradley-Terry preferences. It proves the exponential convergence of on-policy DPO once the batch size exceeds a certain threshold. In addition, it introduces a hybrid DPO sampler based on a preferential G-optimal design to avoid the dependence on coverage. On-policy schemes for reward distillation have also been developed.

**Compliance With Llm Reviewing Policy:**

Affirmed.

**Final Justification:**

I thank the authors for their responses and maintain my positive assessment of the paper.

**Key Questions For Authors:**

See above.

**Limitations:**

Yes.

**Strengths And Weaknesses:**

Strength:
1. The paper establishes linear convergence for the on-policy DPO method, and the technical results appear to be sound and well-justified.

2. It proposes a hybrid DPO variant based on the G-optimal design principle. The construction of the distribution $\pi^g$ seems to be interesting and thoughtfully motivated.

Weakness:
1. In Algorithm 1, the feedback oracle is assumed to be exact, assigning preferences according to the true distribution $\mathbb{P}_*$. In practice, however, one may only have access to approximate preference scores. How would such inaccuracies influence the theoretical guarantees, and can the analysis be extended to accommodate oracle noise or misspecification?

2. I’m a little concerned that the stated “exponential separation” between offline and on-policy DPO may not be a fair comparison. Offline DPO’s $1/\epsilon^2$ factor is a statistical rate under fixed data, while on-policy DPO’s $\log(1/\epsilon)$ dependence is an iteration complexity that assumes continual data access.

---

> ### Author Rebuttal · Authors · 2026-03-27
>
> Thank you for the positive review of our contribution! Here are our responses.
>
> **In Algorithm 1, the feedback oracle is assumed to be exact, assigning preferences according to the true distribution. Can the analysis be extended to accommodate oracle noise or misspecification?**
>
> Actually, our analysis is **easily extended to handle non-realizability & inexact feedback**, as mentioned in lines 163, 210. We omitted these since they only result in additional additive error terms which do not add much to our understanding of coverage dynamics. We outline how these can be handled below.
> * Non-realizability: suppose $\pi^\star\notin\Pi$ but there exists $\tilde\pi\in\Pi$ close to $\pi^\star$, say with log density ratio bounded as $\epsilon^\star$. Then by Lemma C.7, $\lVert\ln\mathbb{P}\_{\pi^\star,\pi_0} - \ln\mathbb{P}\_{\tilde\pi,\pi_0}\rVert_\infty\le 2\gamma\epsilon^\star$. Plugging this into line 971 (the only place we use realizability) gives an additive $O(\epsilon^\star)$ error bound in the one-step guarantee Eq. (8), equivalently taking the max with $O(\epsilon^*)$. This results in an $O(\epsilon^\star)$ lower threshold on the improvement obtained by the iteration argument. Hence, we only incur an extra $O((\epsilon^\star)^2)$ term in the KL bound in Theorem 3.2.
> * Inexact feedback: suppose the oracle uses a preference distribution $\mathbb{P}'=\mathbb{P}\_{\pi_{\theta'}}\ne\mathbb{P}\_\star$. Then training on labels from $\mathbb{P}'$, we will converge to $\theta'$ instead of $\theta^\star$, exactly as in Theorem 3.2. Then we only incur an additive error $\lVert\theta'-\theta^\star\rVert_2$ in the final guarantee; the convergence analysis still applies.
>
> Finally, we mention that such realizability conditions are ubitiquous in RL theory or contextual bandit formulations of post-training, see for example [1,2,3].
>
> **Offline DPO’s $1/\epsilon^2$ factor is a statistical rate under fixed data, while on-policy DPO’s $\log(1/\epsilon)$ dependence is an iteration complexity that assumes continual data access.**
>
> * Comparing the total sample complexity is a standard approach to compare offline, online, and hybrid methods in sequential decision making problems. For example, see [4], Theorem 1.1; [5], Theorem 1; [6], Theorem 4; [7], Table 2; etc.
> * In the case that an online sample is much more costly than an offline sample, our theory can also be used to compare the relative sample requirements $n_{\mathrm{on}}/n_{\mathrm{off}}$ (Corollary 3.3) to see if using the on-policy algorithm is worth the cost, therefore yielding actionable predictions in such scenarios. Thus, we do not view this as a weakness of our theory.
>
> [1] Xiong et al. Iterative Preference Learning from Human Feedback: Bridging Theory and Practice for RLHF under KL-Constraint.
>
> [2] Xie et al. Exploratory Preference Optimization: Harnessing Implicit Q*-Approximation for Sample-Efficient RLHF.
>
> [3] Foster et al. Is a Good Foundation Necessary for Efficient Reinforcement Learning? The Computational Role of the Base Model in Exploration.
>
> [4] Foster et al. Offline Reinforcement Learning: Fundamental Barriers for Value Function Approximation.
>
> [5] Li et al. Reward-agnostic Fine-tuning: Provable Statistical Benefits of Hybrid Reinforcement Learning.
>
> [6] Zanette. Exponential Lower Bounds for Batch Reinforcement Learning: Batch RL can be Exponentially Harder than Online RL.
>
> [7] Tan et al. Hybrid Reinforcement Learning Breaks Sample Size Barriers in Linear MDPs.

---

> > ### Author Rebuttal · Reviewer_VGdf · 2026-04-02
> >
> > I thank the authors for their responses and maintain my positive assessment of the paper.

---

### Official Review · Reviewer_igfJ · 2026-03-11

**Soundness:** 3
**Presentation:** 3
**Significance:** 3
**Originality:** 3
**Overall Recommendation:** 4
**Confidence:** 3

**Summary:**

This paper studies preference learning, with a focus on the comparison between on-policy and off-policy algorithms. In particular, the authors show the theoretical advantage of on-policy learning in terms of convergence rate. Building on this insight, they then propose a sampling algorithm for preference learning that guarantees convergence in two rounds. Other settings including the reward distillation are also explored.

**Compliance With Llm Reviewing Policy:**

Affirmed.

**Key Questions For Authors:**

1. In the two-step algorithm (Algorithm 2), the same amount of data is collected in both stages. I wonder whether there is any (optimal) tradeoff between the first- and second-stage sample sizes under a fixed total budget.

2. The current simulation study does not seem to include the proposed G-optimal design sampling algorithm (Algorithm 2). It would be helpful if the simulations could demonstrate the effectiveness of it.

**Limitations:**

yes.

**Strengths And Weaknesses:**

**Strengths**

1. The paper is theoretically sound, with the theoretical results serving to explain empirical phenomena and providing insights for algorithmic design.
2. Methodologically, the authors present an on-policy preference learning algorithm based on the theoretical insights, which is efficient to implement (with only two rounds of interation) and enjoys desirable theoretical properties (fast convergence).

**Weaknesses**
1. In terms of presentation, the paper is relatively notation-heavy; it would be helpful to improve the presentation by, for example, periodically reminding the readers of the key notation to improve readability.
2. More simulation studies would help justify the advantage of the proposed methodology.

---

> ### Author Rebuttal · Authors · 2026-03-27
>
> Thank you for the positive feedback and helpful suggestions! Below are our responses.
>
> **In terms of presentation, the paper is relatively notation-heavy; it would be helpful to improve the presentation by, for example, periodically reminding the readers of the key notation to improve readability.**
>
> * Thank you for the suggestion! We will incorporate reviewer feedback to improve readability.
>
> **In the two-step algorithm (Algorithm 2), the same amount of data is collected in both stages. I wonder whether there is any (optimal) tradeoff between the first- and second-stage sample sizes under a fixed total budget.**
>
> * This is a very good point. If we denote the first and second stage sample sizes as $n_1,n_2$, resp., we require $n_1\gtrsim d^3R^2$ and $n_2\gtrsim \frac{dR^2}{\epsilon^2}$ to achieve error $\epsilon$. So the first stage requirement is constant w.r.t. target loss (we only need a "good enough" covering policy), while the second stage mainly determines the final loss. If total budget is fixed and $\epsilon$ is sufficiently small, shifting more data towards $n_2$ will be optimal. This is surprising since Algorithm 1 (ordinary on-policy DPO) is the opposite: the optimal batchsize follows a U-curve, so stage 1 should get more data than stage 2.
>
> **More simulation studies would help justify the advantage of the proposed methodology. / The current simulation study does not seem to include the proposed G-optimal design sampling algorithm (Algorithm 2).**
>
> * While each subproblem $\pi^{cg}(x)$ is a convex optimal-design problem, and approximate solutions can be obtained efficiently with standard methods, our current construction still requires solving the global design $\mu^g$, which is not practical for large-scale prompt space. Nonetheless, the main novelty of Section 5 is the proof of existence: we show that there exists a preferential design whose worst-case conditionally-centered variance is uniformly bounded by $d^2$, which is enough to remove the dependence on coverage and $\lambda$ and obtain two-step convergence with polynomial sample complexity. This existence guarantee is already highly nontrivial and, to us, quite unintuitive, as it does not follow straightforwardly from the "conditional" version of classical G-optimal design, but requires the x-marginal pasting trick.
> * In that sense, our core contribution is to identify the right target object (the design objectives in line 296 and Eq. (6)) and certify that coverage-free designs exist at all. Hence we view the current result as a first step establishing optimal design as a promising approach, rather than claiming that this is already a practical large-scale method. We are currently working on developing practical active learning algorithms for LLM post-training that retain this core principle while being scalable and efficient.

---

> > ### Author Rebuttal · Reviewer_igfJ · 2026-04-04
> >
> > I thank the authors for the clarification. I remain positive about the paper.

---

### Official Review · Reviewer_NnY3 · 2026-03-12

**Soundness:** 4
**Presentation:** 2
**Significance:** 3
**Originality:** 3
**Overall Recommendation:** 4
**Confidence:** 3

**Summary:**

The paper provides a theoretical explanation for the success of on-policy preference learning algorithms over offline counterparts by introducing the coverage improvement principle. In a contextual bandit setting, the authors prove that on-policy DPO achieves rapid exponential convergence, successfully avoiding the slow minimax bottleneck that restricts offline learners. Additionally, the paper contributes a novel hybrid DPO sampler utilizing a preferential G-optimal design and develops on-policy reward distillation algorithms that achieve faster noiseless rates.

**Compliance With Llm Reviewing Policy:**

Affirmed.

**Final Justification:**

Thanks for the response. I will maintain my positive rating.

**Key Questions For Authors:**

1. Regarding the hybrid DPO sampler (Algorithm 2), how computationally scalable is the calculation of the conditional G-optimal design $\pi^{cg}(x)$ for the massive, continuous state and action spaces typical in modern LLM applications?
2. The fast convergence of on-policy reward distillation is theoretically compelling. However, these "noiseless" rates assume the underlying reward model $\tilde{r}$ is highly accurate. How robust are these convergence rates to out-of-distribution hallucinations or severe miscalibration in the reward model?
3. Can you elaborate on the practical tuning of the calibration level $\gamma_{c}$ in the reward-calibration update? Do you have recommended heuristics for selecting this hyperparameter across different datasets?

**Limitations:**

yes

**Strengths And Weaknesses:**

**Strengths:**
- The submission is mathematically robust, expertly demonstrating a sharp, exponential separation in total sample complexity between offline and on-policy DPO.
- The authors take a highly creative approach by analyzing the dynamics of the sampling policy's coverage as it evolves throughout training, rather than treating coverage as a static, fixed object. Furthermore, applying the Kiefer-Wolfowitz equivalence theorem to construct a preferential G-optimal design is a novel and effective use of classical statistics.
- The proposed algorithms offer immediate value to practitioners building alignment pipelines. The variance-reduced reward calibration scheme actively solves the degeneracy issues found in existing distillation methods like REBEL.

**Weaknesses:**
- The theoretical guarantees lean heavily on the assumptions of realizability and access to an exact feedback oracle. While the authors acknowledge that noisy or misaligned oracles introduce irreducible errors, this limits how the theory maps to noisy, real-world human feedback.
- The intuition behind the generalized coverage metric $C_{*}^{p}(r)$ is quite dense. The paper would benefit from a gentler introduction to this concept earlier on for readers who are less entrenched in theoretical reinforcement learning.
- Important discussions regarding the boundaries of the theoretical assumptions, such as the limitations of the Bradley-Terry model, are fragmented across various technical sections rather than consolidated in a dedicated discussion.

---

> ### Author Rebuttal · Authors · 2026-03-27
>
> Thank you for the positive assessment of our contributions and detailed feedback! We are pleased that you found our work mathematically robust and highly creative. Below are our responses.
>
> ### Weaknesses
>
> **1. The theoretical guarantees lean heavily on the assumptions of realizability and access to an exact feedback oracle.**
>
> Actually, our analysis is **easily extended to handle non-realizability & inexact feedback**, as mentioned in lines 163, 210. We omitted these since they only result in additional error terms which do not add much to our understanding of coverage dynamics. We outline how these can be handled below.
> * Non-realizability: suppose $\pi^\star\notin\Pi$ but there exists $\tilde\pi\in\Pi$ close to $\pi^\star$, say with log density ratio bounded as $\epsilon^\star$. Then by Lemma C.7, $|\ln\mathbb{P}\_{\pi^\star,\pi_0} - \ln\mathbb{P}\_{\tilde\pi,\pi_0}|_\infty\le 2\gamma\epsilon^\star$. Plugging this into line 971 (the only place we use realizability) gives an additive $O(\epsilon^\star)$ error bound in the one-step guarantee Eq. (8), equivalently taking the max with $O(\epsilon^*)$. This results in an $O(\epsilon^\star)$ lower threshold on the improvement obtained by the iteration argument. Hence, we only incur an extra $O((\epsilon^\star)^2)$ term in the KL bound in Theorem 3.2.
> * Inexact feedback: suppose the oracle uses a preference distribution $\mathbb{P}'=\mathbb{P}\_{\pi_{\theta'}}\ne\mathbb{P}\_\star$. Then training on labels from $\mathbb{P}'$, we will converge to $\theta'$ instead of $\theta^\star$, exactly as in Theorem 3.2. Then we only incur an additive error $|\theta'-\theta^*|_2$ in the final guarantee; the convergence analysis still applies.
>
> Finally, we mention that realizability is a ubitiquous assumption in RL/contextual bandit formulations of post-training, see e.g. Xiong et al., Xie et al., Foster et al. cited in our paper.
>
> **2. The intuition behind the generalized coverage metric is quite dense.**
> * Thank you for the suggestion! We will include an introduction to the literature on generalized coverage.
>
> **3. Discussions regarding the boundaries of the theoretical assumptions are fragmented across various sections.**
> * Thank you for pointing this out; we will aim to discuss the limitations of each assumption right after each one is presented.
>
> ### Questions
>
> **1. How scalable is the calculation of the conditional G-optimal design?**
>
> * On scalability, the subproblem $\pi^{cg}(x)$ is convex, and approximate solutions can be obtained with standard methods such as Frank--Wolfe in $\widetilde{O}(d)$ time where $d$ is feature dimension, which is extremely efficient (i.e., not depending on the action space at all). The real bottleneck is that our current construction still requires solving the global design $\mu^g$, which is not practical for LLMs. Nonetheless, the main novelty of Section 5 is the proof of existence: we show that there exists a design whose worst-case centered variance is uniformly bounded by $d^2$. This guarantee is already highly nontrivial and, to us, quite unintuitive, as it does not follow straightforwardly from the "conditional" version of classical G-optimal design, but requires the x-marginal pasting trick.
> * In that sense, our core contribution is to identify the right target object (the design objectives in line 296 and Eq. (6)) and certify that coverage-free designs exist at all. Hence we view the current result as a first step establishing optimal design as a promising approach, rather than claiming that this is already a practical large-scale method. We are currently working on developing active learning algorithms for LLM post-training that retain this core principle while being scalable and efficient.
>
> **2. How robust are these convergence rates to out-of-distribution hallucinations or severe miscalibration in the reward model?**
>
> * Our theory regarding reward distillation can also handle non-realizable (out-of-distribution) settings and reward miscalibration, similarly to our response to Weakness 1. The core "iterative improvement" argument still goes through, only resulting in additive error terms in the final bound.
>
> **3. Can you elaborate on the practical tuning of the calibration level? Do you have recommended heuristics for selecting this hyperparameter across different datasets?**
>
> * In practice, we use the following heuristic: $\gamma_c$ is chosen so that $\gamma_c \log \frac{\hat\pi_k}{\pi_0}$ is on a scale comparable to the reward scores/reward differences. Thus, reward models with larger output scales call for a larger $\gamma_c$, while smaller-scale reward models generally require a smaller one. Since ArmoRM used in our main experiments, has a relatively small reward scale (roughly $10^{-2}$ to $10^{-1}$), we use a small fixed $\gamma_c = 3\times 10^{-4}$ across all rounds.
>
> We hope that this answers most questions, and humbly ask the reviewer to consider raising their score if their concerns have been satisfactorily addressed.

---

> > ### Author Rebuttal · Reviewer_NnY3 · 2026-04-04
> >
> > Thanks for the response. I will maintain my positive rating.

---

### Decision · Program_Chairs · 2026-04-30

**Decision:**

Accept (regular)

**Comment:**

This paper provides a theoretical justification for why on-policy DPO improves over offline DPO by analyzing how policy coverage evolves during the online learning process. The main result shows that on-policy DPO achieves exponential convergence, in contrast to the slower minimax rate for offline learning. The paper further proposes a preferential G-optimal design to remove the dependence on coverage. Overall, the reviewers appreciated the paper’s theoretical contributions, technical rigor, and the insights it provides for algorithm design. For the final version, the authors are encouraged to further improve the presentation and discuss model misspecification, experimental support, and the computational limitations of the G-optimal design method.